# THE GEOMETRY OF MEMORYLESS STOCHASTIC POL­ICY OPTIMIZATION IN INFINITE-HORIZON POMDPS

**Johannes Müller**
Max Planck Institute for Mathematics in the Sciences, Leipzig, Germany
`jmueller@mis.mpg.de`

**Guido Montúfar**
Department of Mathematics and Department of Statistics, UCLA, CA, USA
Max Planck Institute for Mathematics in the Sciences, Leipzig, Germany
`montufar@math.ucla.edu`

## ABSTRACT

We consider the problem of finding the best memoryless stochastic policy for an infinite-horizon partially observable Markov decision process (POMDP) with fi­nite state and action spaces with respect to either the discounted or mean reward criterion. We show that the (discounted) state-action frequencies and the expected cumulative reward are rational functions of the policy, whereby the degree is de­termined by the degree of partial observability. We then describe the optimization problem as a linear optimization problem in the space of feasible state-action fre­quencies subject to polynomial constraints that we characterize explicitly. This allows us to address the combinatorial and geometric complexity of the optimiza­tion problem using recent tools from polynomial optimization. In particular, we estimate the number of critical points and use the polynomial programming de­scription of reward maximization to solve a navigation problem in a grid world.

## 1 INTRODUCTION

Markov decision processes (MDPs) were introduced by Bellman (1957) as a model for sequential decision making and optimal planning (see, e.g., Howard, 1960; Derman, 1970; Puterman, 2014). Many algorithms in reinforcement learning rely on the ideas and methods developed in the context of MDPs (see, e.g., Sutton & Barto, 2018). Often in practice, the decisions need to be made based only on incomplete information of the state of the system. This setting is modeled by partially observable Markov decision processes (POMDPs) introduced by Åström (1965) (for a historical discussion see Monahan, 1982), which have become an important model for planning under uncertainty. In this work we pursue a geometric characterization of the policy optimization problem in POMDPs over the class of stochastic memoryless policies and its dependence on the degree of partial observability.

It is well known that acting optimally in POMDPs may require memory (Åström, 1965). A POMDP with unlimited memory policies can be modeled as a belief state MDP, where the states are replaced by probability distributions that serve as sufficient statistics for the previous observations (Kaelbling et al., 1998; Murphy, 2000). Finding an optimal policy in this class is PSPACE-complete for finite horizons (Papadimitriou & Tsitsiklis, 1987) and undecidable for infinite horizons (Madani et al., 2003; Chatterjee et al., 2016). Therefore, it is of interest to consider POMDPs with constrained policy classes. A natural class to consider are memoryless policies, also known as reactive or Markov policies, which select actions based solely on the current observations. In this case, it is useful to allow the actions to be selected stochastically, which not only allows for better solutions but also provides a continuous optimization domain (Singh et al., 1994).

Although they are more restrictive than policies with memory, memoryless policies are attractive as they are easier to optimize and are versatile enough for certain applications (Tesauro, 1995; Loch & Singh, 1998; Williams & Singh, 1999; Kober et al., 2013). In fact, finite-memory policies can be modeled in terms of memoryless policies by supplementing the state of the system with an external memory (Littman, 1993; Peshkin et al., 1999; Icarte et al., 2021). Hence theoretical advances on memoryless policy optimization are also of interest to finite-memory policy optimization. Theoret-

ical aspects and optimization strategies over the class of memoryless policies have been studied in numerous works (see, e.g., Littman, 1994; Singh et al., 1994; Jaakkola et al., 1995; Loch & Singh, 1998; Williams & Singh, 1999; Baxter et al., 2000; Baxter & Bartlett, 2001; Li et al., 2011; Azizzadenesheli et al., 2018). However, finding exact or approximate optimal memoryless stochastic policies for POMDPs is still considered an open problem (Azizzadenesheli et al., 2016), which is NP-hard in general (Vlassis et al., 2012). One reason for the difficulties in optimizing POMDPs is that, even in a tabular setting, the problem is non-convex and can exhibit suboptimal strict local optima (Bhandari & Russo, 2019). For memoryless policies the expected cumulative reward is a linear function of the corresponding (discounted) state-action frequencies. In the case of MDPs the feasible set of state-action frequencies is known to form a polytope, so that the optimization problem can be reduced to a linear program (Manne, 1960; De Ghellinck, 1960; d'Epenoux, 1963; Hordijk & Kallenberg, 1981). On the other hand, to the best of our knowledge, for POMDPs the specific structure of the feasibility constraints and the optimization problem have not been studied, at least not in the same level of detail (see related works below).

**Related works** In MDPs, the (discounted) state-action frequencies form a polytope resp. a compact convex set in the finite resp. countable state-action cases, whereby the extreme points are given by the state-action frequencies of deterministic stationary policies (Derman, 1970; Altman & Shwartz, 1991). Further, Dadashi et al. (2019) showed that the set of state value functions in finite state-action MDPs is a finite union of polytopes. The set of stationary state-action distributions of POMDPs has been studied by Montúfar et al. (2015) highlighting a decomposition into infinitely many convex subsets whose dimensions depend on the degree of observability. Although this decomposition can be used to localize optimal policies to some extent, a description of the pieces in combination is still missing, needed to capture the properties of the optimization problem. We will obtain a detailed description in terms of finitely many polynomial constraints with closed form expressions and bound their degrees in terms of the observation mechanism. This yields a polynomial programming formulation of POMDPs generalizing the linear programming formulation of MDPs. This is different to the formulation as a quadratically constrained problem by Amato et al. (2006), where the number and degree of constraints do not depend on the observability. Finally, Cohen & Parmentier (2018) described finite horizon POMDPs as a mixed integer linear program.

Grinberg & Precup (2013) showed that the expected mean reward is a rational function and obtained bounds on the degree of this function. We generalize this to the setting of discounted rewards and refine the result by relating the rational degree to the degree of observability. For both MDPs and POMDPs the expected cumulative reward is known to be a non-convex function of the policy even for tabular policy models (Bhandari & Russo, 2019). Nonetheless, for MDPs critical points can be shown to be global maxima under mild conditions. In contrast, for POMDPs or MDPs with linearly restricted policy models, it is known that non-global local optimizers can exist (Baxter et al., 2000; Poupart et al., 2011; Bhandari & Russo, 2019). However, nothing is known about the number of local optimizers. We will present bounds on the number of critical points building on our computation of the rational degree and feasibility constraints.

The structure of the expected cumulative reward has been studied in terms of the location of the global optimizers and the existence of local optimizers. Most notably, it is well known that in MDPs there always exist optimal policies which are memoryless and deterministic (see Puterman, 2014). In the case of POMDPs, optimal memoryless policies may need to be stochastic (see Singh et al., 1994). Montúfar et al. (2015); Montúfar & Rauh (2017); Montúfar et al. (2019) obtained upper bounds on the number of actions that need to be randomized by these policies, which in the worst case is equal to the number of states that are compatible with the observation. Although we do not improve these results (which are indeed tight in some cases), our description of the expected cumulative reward function leads to a simpler proof of the bounds obtained by Montúfar et al. (2015).

Neyman (2003) considered stochastic games as semialgebraic problems showing that the minmax and maxmin payoffs in an $n$-player game are semialgebraic functions of the discount factor. Although this is not directly related to our work, we take a similar philosophy. We pursue a semialgebraic description of the feasible set of discounted state-action frequencies in POMDPs, which is closely related to the general spirit of semialgebraic statistics, where this is usually referred to as the implitization problem (Zwiernik, 2016). Based on this we characterize the properties of the optimization problem by its algebraic degree, a concept that has been advanced in recent works on polynomial optimization (Bajaj, 1988; Nie & Ranestad, 2009; Özlüm Çelik et al., 2021).

**Contributions** We obtain results for infinite-horizon POMDPs with memoryless stochastic policies under the mean or discounted reward criteria which can be summarized as follows.

1. We show that the state-action frequencies and the expected cumulative reward can be written as fractions of determinantal polynomials in the entries of the stochastic policy matrix. We show that the degree of these polynomials is directly related to the degree of observability (see Theorem 4).

2. We describe the set of feasible state-action frequencies as a basic semialgebraic set, i.e., as the solution set to a system of polynomial equations and inequalities, for which we also derive closed form expressions (see Theorem 16 and Remark 18).

3. We reformulate the expected cumulative reward optimization problem as the optimization of a linear function subject to polynomial constraints (see Remark 19), which we use to solve a navigation problem in a grid world (see Appendix F). This is a POMDP generalization of the dual linear programming formulation of MDPs (Kallenberg, 1994; Puterman, 2014).

4. We present two methods for computing the number of critical points, which rely, respectively, on the rational degree of the expected cumulative reward function and the geometric description of the feasible set of state-action frequencies (see Theorem 20, Proposition 21 and Appendix D).

## 2 PRELIMINARIES

We denote the simplex of probability distributions over a finite set $\mathcal{X}$ by $\Delta_{\mathcal{X}}$. An element $\mu \in \Delta_{\mathcal{X}}$ is a vector with non-negative entries $\mu_x = \mu(x)$, $x \in \mathcal{X}$ adding to one. We denote the set of Markov kernels from a finite set $\mathcal{X}$ to another finite set $\mathcal{Y}$ by $\Delta_{\mathcal{Y}}^{\mathcal{X}}$. An element $Q \in \Delta_{\mathcal{Y}}^{\mathcal{X}}$ is a $|\mathcal{X}| \times |\mathcal{Y}|$ row stochastic matrix with entries $Q_{xy} = Q(y|x)$, $x \in \mathcal{X}$, $y \in \mathcal{Y}$. Given $Q^{(1)} \in \Delta_{\mathcal{Y}}^{\mathcal{X}}$ and $Q^{(2)} \in \Delta_{\mathcal{Z}}^{\mathcal{Y}}$ we denote their composition into a kernel from $\mathcal{X}$ to $\mathcal{Z}$ by $Q^{(2)} \circ Q^{(1)} \in \Delta_{\mathcal{Z}}^{\mathcal{X}}$. Given $p \in \Delta_{\mathcal{X}}$ and $Q \in \Delta_{\mathcal{Y}}^{\mathcal{X}}$ we denote their composition into a joint probability distribution by $p * Q \in \Delta_{\mathcal{X} \times \mathcal{Y}}$, $(p * Q)(x, y) := p(x)Q(y|x)$. The support of $v \in \mathbb{R}^{\mathcal{X}}$ is the set $\text{supp}(v) = \{x \in \mathcal{X} : v_x \neq 0\}$.

A *partially observable Markov decision process* or shortly *POMDP* is a tuple $(\mathcal{S}, \mathcal{O}, \mathcal{A}, \alpha, \beta, r)$. We assume that $\mathcal{S}, \mathcal{O}$ and $\mathcal{A}$ are finite sets which we call *state*, *observation* and *action space* respectively. We fix a Markov kernel $\alpha \in \Delta_{\mathcal{S}}^{\mathcal{S} \times \mathcal{A}}$ which we call *transition mechanism* and a kernel $\beta \in \Delta_{\mathcal{O}}^{\mathcal{S}}$ which we call *observation mechanism*. Further, we consider an *instantaneous reward vector* $r \in \mathbb{R}^{\mathcal{S} \times \mathcal{A}}$. We call the system *fully observable* if $\beta = \text{id}$[1], in which case the POMDP simplifies to a *Markov decision process* or shortly *MDP*.

As *policies* we consider elements $\pi \in \Delta_{\mathcal{A}}^{\mathcal{O}}$ and call the Markov kernel $\tau = \pi \circ \beta \in \Delta_{\mathcal{A}}^{\mathcal{S}}$ its corresponding *effective policy*. A policy induces transition kernels $P_\pi \in \Delta_{\mathcal{S} \times \mathcal{A}}^{\mathcal{S} \times \mathcal{A}}$ and $p_\pi \in \Delta_{\mathcal{S}}^{\mathcal{S}}$ by

$$P_\pi(s', a'|s, a) := \alpha(s'|s, a)(\pi \circ \beta)(a'|s') \quad \text{and} \quad p_\pi(s'|s) := \sum_{a \in \mathcal{A}} (\pi \circ \beta)(a|s)\alpha(s'|s, a).$$

For any initial state distribution $\mu \in \Delta_{\mathcal{S}}$, a policy $\pi \in \Delta_{\mathcal{A}}^{\mathcal{O}}$ defines a Markov process on $\mathcal{S} \times \mathcal{A}$ with transition kernel $P_\pi$ which we denote by $\mathbb{P}^{\pi,\mu}$. For a *discount rate* $\gamma \in (0, 1)$ and $\gamma = 1$ we define

$$R_\gamma^\mu(\pi) := \mathbb{E}_{\mathbb{P}^{\pi,\mu}}\left[(1 - \gamma)\sum_{t=0}^{\infty} \gamma^t r(s_t, a_t)\right] \quad \text{and} \quad R_1^\mu(\pi) := \lim_{T \to \infty} \mathbb{E}_{\mathbb{P}^{\pi,\mu}}\left[\frac{1}{T}\sum_{t=0}^{T-1} r(s_t, a_t)\right],$$

called the *expected discounted reward* and the *expected mean reward*, respectively. The goal is to maximize this function over the policy polytope $\Delta_{\mathcal{A}}^{\mathcal{O}}$. For a policy $\pi$ we define the *value function* $V_\gamma^\pi \in \mathbb{R}^{\mathcal{S}}$ via $V_\gamma^\pi(s) := R_\gamma^{\delta_s}(\pi)$, $s \in \mathcal{S}$, where $\delta_s$ is the Dirac distribution concentrated at $s$. A short calculation shows that $R_\gamma^\mu(\pi) = \sum_{s,a} r(s,a)\eta_\gamma^{\pi,\mu}(s,a) = \langle r, \eta_\gamma^{\pi,\mu}\rangle_{\mathcal{S} \times \mathcal{A}}$ (Zahavy et al., 2021), where

$$\eta_\gamma^{\pi,\mu}(s,a) := \begin{cases} (1 - \gamma)\sum_{t=0}^{\infty} \gamma^t \mathbb{P}^{\pi,\mu}(s_t = s, a_t = a), & \text{if } \gamma \in (0, 1) \\ \lim_{T \to \infty} \frac{1}{T}\sum_{t=0}^{T-1} \mathbb{P}^{\pi,\mu}(s_t = s, a_t = a), & \text{if } \gamma = 1. \end{cases} \tag{1}$$

Here, $\eta_\gamma^{\pi,\mu}$ is an element of $\Delta_{\mathcal{S} \times \mathcal{A}}$ called *expected (discounted) state-action frequency* (Derman, 1970), (discounted) visitation/occupancy measure or on-policy distribution (Sutton & Barto, 2018). Denoting the state marginal of $\eta_\gamma^{\pi,\mu}$ by $\rho_\gamma^{\pi,\mu} \in \Delta_{\mathcal{S}}$ we have $\eta_\gamma^{\pi,\mu}(s,a) = \rho_\gamma^{\pi,\mu}(s)(\pi \circ \beta)(a|s)$. We recall the following well-known facts.

---

[1] More generally, the system is fully observable if the supports of $\{\beta(\cdot|s)\}_{s \in \mathcal{S}}$ are disjoint subsets of $\mathcal{O}$.

**Proposition 1** (Existence of state-action frequencies and rewards). *Let $(\mathcal{S}, \mathcal{O}, \mathcal{A}, \alpha, \beta, r)$ be a POMDP, $\gamma \in (0, 1]$ and $\mu \in \Delta_{\mathcal{S}}$. Then $\eta_\gamma^{\pi,\mu}, \rho_\gamma^{\pi,\mu}$ and $R_\gamma^\mu(\pi)$ exist for every $\pi \in \Delta_{\mathcal{A}}^{\mathcal{O}}$ and $\mu \in \Delta_{\mathcal{S}}$ and are continuous in $\gamma \in (0, 1]$ for fixed $\pi$ and $\mu$.*

For $\gamma = 1$ we work under the following standard assumption in the (PO)MDP literature[2].

**Assumption 2** (Uniqueness of stationary disitributions). *If $\gamma = 1$, we assume that for any policy $\pi \in \Delta_{\mathcal{A}}^{\mathcal{O}}$ there exists a unique stationary distribution $\eta \in \Delta_{\mathcal{S} \times \mathcal{A}}$ of $P_\pi$.*

The following proposition shows in particular that for any initial distribution $\mu$, the infinite time horizon state-action frequency $\eta_\gamma^{\pi,\mu}$ is the unique discounted stationary distribution of $P_\pi$.

**Proposition 3** (State-action frequencies are discounted stationary). *Let $(\mathcal{S}, \mathcal{O}, \mathcal{A}, \alpha, \beta, r)$ be a POMDP, $\gamma \in (0, 1]$ and $\mu \in \Delta_{\mathcal{S}}$. Then $\eta_\gamma^{\pi,\mu}$ is the unique element in $\Delta_{\mathcal{S} \times \mathcal{A}}$ satisfying the discounted stationarity equation $\eta_\gamma^{\pi,\mu} = \gamma P_\pi^T \eta_\gamma^{\pi,\mu} + (1 - \gamma)(\mu * (\pi \circ \beta))$. Further, $\rho_\gamma^{\pi,\mu}$ is the unique element in $\Delta_{\mathcal{S}}$ satisfying $\rho_\gamma^{\pi,\mu} = \gamma p_\pi^T \rho_\gamma^{\pi,\mu} + (1 - \gamma)\mu$.*

We denote the set of all state-action frequencies in the fully and in the partially observable case by

$$\mathcal{N}_\gamma^\mu := \left\{ \eta_\gamma^{\pi,\mu} \in \Delta_{\mathcal{S} \times \mathcal{A}} \mid \pi \in \Delta_{\mathcal{A}}^{\mathcal{S}} \right\} \quad \text{and} \quad \mathcal{N}_\gamma^{\mu,\beta} := \left\{ \eta_\gamma^{\pi,\mu} \in \Delta_{\mathcal{S} \times \mathcal{A}} \mid \pi \in \Delta_{\mathcal{A}}^{\mathcal{O}} \right\}.$$

We have seen that the expected cumulative reward function $R_\gamma^\mu \colon \Delta_{\mathcal{A}}^{\mathcal{O}} \to \mathbb{R}$ factorises according to

$$\Delta_{\mathcal{A}}^{\mathcal{O}} \xrightarrow{f_\beta} \Delta_{\mathcal{A}}^{\mathcal{S}} \xrightarrow{\Psi_\gamma^\mu} \mathcal{N}_\gamma^{\mu,\beta} \to \mathbb{R}, \quad \pi \mapsto \pi \circ \beta \mapsto \eta_\gamma^{\pi,\mu} \mapsto \langle r, \eta_\gamma^{\pi,\mu} \rangle_{\mathcal{S} \times \mathcal{A}}.$$

This is illustrated in Figure 1. We make use of this decomposition in two different ways. First, in Section 3 we study the algebraic properties of the parametrization $\Psi_\gamma^\mu$ of the set of state-action frequencies $\mathcal{N}_\gamma^{\mu,\beta}$. In Section 4 we derive a description of $\mathcal{N}_\gamma^{\mu,\beta}$ via polynomial inequalities.

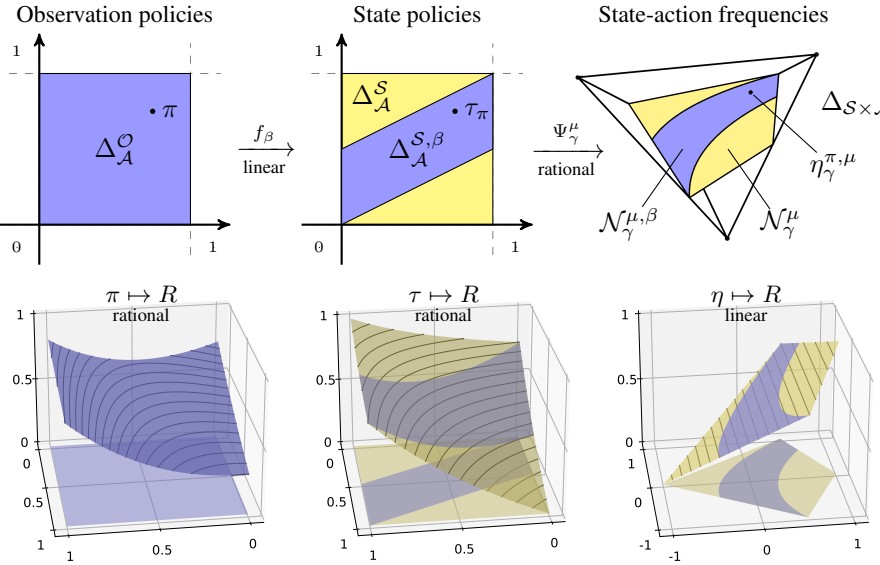

Figure 1: Geometry of a POMDP with two states, two actions and two observations. The top shows the observation policy polytope $\Delta_{\mathcal{A}}^{\mathcal{O}}$; the associated state policy polytope $\Delta_{\mathcal{A}}^{\mathcal{S}}$ (yellow) along with its subset of effective policies $\Delta_{\mathcal{A}}^{\mathcal{S},\beta}$ (blue); and the corresponding sets of discounted state-action frequencies in the simplex $\Delta_{\mathcal{S} \times \mathcal{A}}$ (a tetrahedron in this case). The bottom shows the graph of the expected cumulative discounted reward $R$ as a function of the observation policy $\pi$; the state policy $\tau$; and the discounted state-action frequencies $\eta$. We characterize the parametrization and geometry of these domains and the structure of the expected cumulative reward function.

---

[2] Assumption 2 is weaker than ergodicity, for which well known criteria exist. For $\gamma < 1$ the assumption is not required, since the discounted stationary distributions are always unique.

## 3 THE PARAMETRIZATION OF DISCOUNTED STATE-ACTION FREQUENCIES

In this section we show that the discounted state-action frequencies, the value function and the expected cumulative reward of POMDPs are rational functions and relate their rational degree, which can be interpreted as a measure of their complexity, to the degree of observability. Here, we say that a function is a rational function of degree at most $k$ if it is the fraction of two polynomials of degree at most $k$. By Cramer's rule (see Appendix B.2), it holds that

$$\eta_\gamma^{\pi,\mu}(s,a) = (\pi \circ \beta)(a|s)\rho_\gamma^{\pi,\mu}(s) = (\pi \circ \beta)(a|s) \cdot (1-\gamma) \cdot \frac{\det(I - \gamma p_\pi^T)_s^\mu}{\det(I - \gamma p_\pi^T)},$$

where $(I - \gamma p_\pi^T)_s^\mu$ denotes the matrix obtained by replacing the $s$-row of $I - \gamma p_\pi^T$ with $\mu$. Since $p_\pi$ depends linearly on $\pi$ and the determinant is a polynomial, this is a rational function in the entries of $\pi$. For the degree, we show the following result in Appendix B.1.

**Theorem 4** (Degree of POMDPs). *Let $(\mathcal{S}, \mathcal{O}, \mathcal{A}, \alpha, \beta, r)$ be a POMDP, $\mu \in \Delta_\mathcal{S}$ be an initial distribution and $\gamma \in (0,1)$ a discount factor. The state-action frequencies $\eta_\gamma^{\pi,\mu}$ and $\rho_\gamma^{\pi,\mu}$, the value function $V_\gamma^\pi$ and the expected cumulative reward $R_\gamma^\mu(\pi)$ are rational functions with common denominator in the entries of the policy $\pi$. Further, if they are restricted to the subset $\Pi \subseteq \Delta_\mathcal{A}^\mathcal{O}$ of policies which agree with a fixed policy $\pi_0$ on all states outside of $O \subseteq \mathcal{O}$, they have degree at most*

$$\left| \left\{ s \in \mathcal{S} \mid \beta(o|s) > 0 \text{ for some } o \in O \right\} \right|.$$

Hence, the number of states that are compatible with $o$ determines the algebraic complexity of the discounted state-action frequencies, the value function and the reward function. Various refinements of the theorem are presented in Appendix B.1. For the mean reward case and under an ergodicity assumption, Grinberg & Precup (2013) showed that the stationary distributions are a rational function of degree of most $|\mathcal{S}|$ of the policy. From Theorem 4 we can derive multiple implications (see also Appendix D.5.2 for implications on the optimization landscape):

**Corollary 5** (Feasible state-action frequencies and value functions form semialgebraic sets). *Consider a POMDP $(\mathcal{S}, \mathcal{O}, \mathcal{A}, \alpha, \beta, r)$ and let $\mu \in \Delta_\mathcal{S}$ be an initial distribution and $\gamma \in (0,1)$ a discount factor. The set of discounted state-action frequencies and the set of value functions are semialgebraic sets[3].*

*Proof.* By Theorem 4, both sets possess a rational and thus a semialgebraic parametrization and are semialgebraic by the Tarski-Seidenberg theorem (Neyman, 2003). □

We compute the defining linear and polynomial (in)equalities of the set of feasible state-action frequencies in Section 4 for MDPs and POMDPs respectively, which shows in particular that also in the mean case the state-action frequencies form a semialgebraic set. The special properties of degree-one rational functions, which we elaborate in the Appendix B.3, imply the following results. The first one is a refinement of Dadashi et al. (2019, Lemma 4), stating that linear interpolation between two policies that differ on a single state leads to a linear interpolation of the corresponding value functions. We generalize this to state-action frequencies, explicitly compute the interpolation speed and describe the curves obtained by interpolation between arbitrary policies. Further, our formulation extends to the mean reward case (see Remark 43).

**Proposition 6.** *Let $(\mathcal{S}, \mathcal{A}, \alpha, r)$ be an MDP and $\gamma \in (0,1)$. Further, let $\pi_0, \pi_1 \in \Delta_\mathcal{A}^\mathcal{S}$ be two policies that differ on at most $k$ states. For any $\lambda \in [0,1]$ let $V_\lambda \in \mathbb{R}^\mathcal{S}$ and $\eta_\lambda^\mu \in \Delta_{\mathcal{S} \times \mathcal{A}}$ denote the value function and state-action frequency belonging to the policy $\pi_0 + \lambda(\pi_1 - \pi_0)$ with respect to the discount factor $\gamma$, the initial distribution $\mu$ and the instantaneous reward $r$. Then the rational degrees of $\lambda \mapsto V_\lambda$ and $\lambda \mapsto \eta_\lambda$ are at most $k$. If they differ on at most one state $\tilde{s} \in \mathcal{S}$ then*

$$V_\lambda = V_0 + c(\lambda) \cdot (V_1 - V_0) \quad \text{and} \quad \eta_\lambda^\mu = \eta_0^\mu + c(\lambda) \cdot (\eta_1^\mu - \eta_0^\mu) \quad \text{for all } \lambda \in [0,1],$$

*where*

$$c(\lambda) = \frac{\det(I - \gamma p_1)\lambda}{\det(I - \gamma p_\lambda)} = \frac{\det(I - \gamma p_1)\lambda}{(\det(I - \gamma p_1) - \det(I - \gamma p_0))\lambda + \det(I - \gamma p_0)} = \lambda \cdot \frac{\rho_\lambda^\mu(\tilde{s})}{\rho_1^\mu(\tilde{s})}.$$

[3]A semialgebraic set is a set defined by a number of polynomial inequalities or a finite union of such sets; for details see Appendix A.2.

In particular, for a blind controller with two actions the set of feasible value functions and the set of feasible state-action frequencies are pieces of curves with rational parametrization of degree at most $k = |\mathcal{S}|$. By Theorem 4, the cumulative reward of (PO)MDPs is a degree-one rational function in every row of the (effective) policy. Since degree-one rational functions attain their maximum in a vertex (Corollary 39), we immediately obtain the existence of an optimal policy which is deterministic on every observation from which the state can be reconstructed, which has been shown using other methods by Montúfar et al. (2015).

**Proposition 7** (Determinism of optimal policies). *Let $(\mathcal{S}, \mathcal{O}, \mathcal{A}, \alpha, \beta, r)$ be a POMDP, $\mu \in \Delta_{\mathcal{S}}$ be an initial distribution and $\gamma \in (0, 1)$ a discount factor and let $\pi \in \Delta_{\mathcal{A}}^{\mathcal{O}}$ be an arbitrary policy and denote the set of observations $o$ such that $|\{s \in \mathcal{S} \mid \beta(o|s) > 0\}| \leq 1$ by $O$. Then there is a policy $\tilde{\pi}$, which is deterministic on every $o \in O$ such that $R_{\gamma}^{\mu}(\tilde{\pi}) \geq R_{\gamma}^{\mu}(\pi)$.*

*Proof.* For $o \in O$, the reward function restricted to the $o$-component of the policy is a rational function of degree at most one. By Corollary 39 (see Appendix B.3.2), there is a policy $\tilde{\pi}$, which is deterministic on $o$ and satisfies $R_{\gamma}^{\mu}(\tilde{\pi}) \geq R_{\gamma}^{\mu}(\pi)$. Iterating over $o \in O$ yields the result. □

On observations which can be made from more than one state, bounds on the required stochasticity were established by Montúfar & Rauh (2017); Montúfar et al. (2019).

## 4 THE SET OF FEASIBLE DISCOUNTED STATE-ACTION FREQUENCIES

In Corollary 5, we have seen that the state-action frequencies form a semialgebraic set. Now we aim to describe its defining polynomial inequalities. In the case of full observability, the feasible state-action freqencies are known to form a polytope (Derman, 1970; Altman & Shwartz, 1991) which is closely linked to the dual linear programming formulation of MDPs (Hordijk & Kallenberg, 1981), see also Figure 1. We first describe the combinatorial properties of this polytope (see Appendix C.1.2) and extend the result to the partially observable case, for which we obtain explicit polynomial inequalities induced by the partial observability under a mild assumption. Most proofs are postponed to Appendix C. In Section 5 we discuss how the degree of these defining polynomials allows us to upper bound the number of critical points of the optimization problem. We use the following explicit version of the classic characterization of the state-action frequencies as a polytope (see Appendix C.1).

**Proposition 8** (Characterization of $\mathcal{N}_{\gamma}^{\mu}$). *Let $(\mathcal{S}, \mathcal{A}, \alpha, r)$ be an MDP, $\mu \in \Delta_{\mathcal{S}}$ be an initial distribution and $\gamma \in (0, 1]$. It holds that*

$$\mathcal{N}_{\gamma}^{\mu} = \Delta_{\mathcal{S} \times \mathcal{A}} \cap \left\{ \eta \in \mathbb{R}^{\mathcal{S} \times \mathcal{A}} \mid \langle w_{\gamma}^{s}, \eta \rangle_{\mathcal{S} \times \mathcal{A}} = (1 - \gamma)\mu_{s} \text{ for } s \in \mathcal{S} \right\} \tag{2}$$

*where $w_{\gamma}^{s} := \delta_{s} \otimes \mathbb{1}_{\mathcal{A}} - \gamma\alpha(s|\cdot, \cdot)$. For $\gamma \in (0, 1)$, $\Delta_{\mathcal{S} \times \mathcal{A}}$ can be replaced by $[0, \infty)^{\mathcal{S} \times \mathcal{A}}$ in (2).*

Now we turn towards the partially observable case and introduce the following notation.

**Definition 9** (Effective policy polytope). We call the set of effective policies $\tau = \pi \circ \beta \in \Delta_{\mathcal{A}}^{\mathcal{S}}$ the *effective policy polytope* and denote it by $\Delta_{\mathcal{A}}^{\mathcal{S}, \beta}$.

Note that $\Delta_{\mathcal{A}}^{\mathcal{S}, \beta}$ is indeed a polytope since it is the image of the polytope $\Delta_{\mathcal{A}}^{\mathcal{O}}$ under the linear mapping $\pi \mapsto \pi \circ \beta = \beta\pi$. Hence, we can write it as an intersection $\Delta_{\mathcal{A}}^{\mathcal{S}, \beta} = \Delta_{\mathcal{A}}^{\mathcal{S}} \cap \mathcal{U} \cap \mathcal{C}$, where $\mathcal{U}, \mathcal{C} \subseteq \mathbb{R}^{\mathcal{S} \times \mathcal{A}}$ are an affine subspace and a polyhedral cone and describe a finite set of linear equalities and a finite set of linear inequalities respectively.

**Defining linear inequalities of the effective policy polytope**    Obtaining inequality descriptions of the images of polytopes under linear maps is a fundamental problem that is non-trivial in general. It can be approached algorithmically, e.g., by Fourier-Motzkin elimination, block elimination, vertex approaches, and equality set projection (Jones et al., 2004). In the special case where the linear map is injective, one can give the defining inequalities in closed form as we show in Appendix C.2.1. Hence, for the purpose of obtaining closed-formulas for the effective policy polytope we make the following assumption. However, our subsequent analysis in Section 4 can handle any inequalities.

**Assumption 10.** The matrix $\beta \in \Delta_{\mathcal{O}}^{\mathcal{S}} \subseteq \mathbb{R}^{\mathcal{S} \times \mathcal{O}}$ has linearly independent columns.

**Remark 11.** The assumption above does not imply that the system is fully observable. Recall that if $\beta$ has linearly independent columns, the Moore-Penrose takes the form $\beta^+ = (\beta^T\beta)^{-1}\beta^T$. An interesting special case is when $\beta$ is deterministic but may map several states to the same observation (this is the partially observed setting considered in numerous works). In this case, $\beta^+ = \mathrm{diag}(n_1^{-1}, \dots, n_{|\mathcal{O}|}^{-1})\beta^T$, where $n_o$ denotes the number of states with observation $o$. In this case, $\beta_{so}^+$ agrees with the conditional distribution $\beta(s|o)$ with respect to a uniform prior over the states; however, this is not in general the case since $\beta^+$ can have negative entries.

**Theorem 12** ($H$-description of the effective policy polytope). *Let $(\mathcal{S}, \mathcal{O}, \mathcal{A}, \alpha, \beta, r)$ be a POMDP and let Assumption 10 hold. Then it holds that*

$$\Delta_{\mathcal{A}}^{\mathcal{S},\beta} = \Delta_{\mathcal{A}}^{\mathcal{S}} \cap \mathcal{U} \cap \mathcal{C} = \mathcal{U} \cap \mathcal{C} \cap \mathcal{D}, \tag{3}$$

*where $\mathcal{U} = \{\pi \circ \beta \mid \pi \in \mathbb{R}^{\mathcal{S} \times \mathcal{O}}\} = \ker(\beta^T)^{\perp}$ is a subspace, $\mathcal{C} = \{\tau \in \mathbb{R}^{\mathcal{S} \times \mathcal{A}} \mid \beta^+\tau \geq 0\}$ is a pointed polyhedral cone and $\mathcal{D} = \{\tau \in \mathbb{R}^{\mathcal{S} \times \mathcal{A}} \mid \sum_a (\beta^+\tau)_{oa} = 1 \text{ for all } o \in \mathcal{O}\}$ an affine subspace. Further, the face lattices of $\Delta_{\mathcal{A}}^{\mathcal{O}}$ and $\Delta_{\mathcal{A}}^{\mathcal{S},\beta}$ are isomorphic.*

**Defining polynomial inequalities of the feasible state-action frequencies**   In order to transfer inequalities in $\Delta_{\mathcal{A}}^{\mathcal{S}}$ to inequalities in the set of state-action frequencies $\mathcal{N}_{\gamma}^{\mu}$, we use that the inverse of $\pi \mapsto \eta^{\pi}$ is given through conditioning (see Proposition 46) under the following assumption.

**Assumption 13** (Positivity). *Let $\rho_{\gamma}^{\pi,\mu} > 0$ hold entrywise for all policies $\pi \in \Delta_{\mathcal{A}}^{\mathcal{S}}$.*

This assumption holds in particular, if either $\alpha > 0$ and $\gamma > 0$ or $\gamma < 1$ and $\mu > 0$ entrywise (see Appendix C.1). Assumption 13 is standard in linear programming approaches and necessary for the convergence of policy gradient methods in MDPs (Kallenberg, 1994; Mei et al., 2020). By conditioning, we can translate linear inequalities in $\Delta_{\mathcal{A}}^{\mathcal{S}}$ into polynomial inequalities in $\mathcal{N}_{\gamma}^{\mu}$.

**Proposition 14** (Correspondence of inequalities). *Let $(\mathcal{S}, \mathcal{A}, \alpha, r)$ be an MDP, $\tau \in \Delta_{\mathcal{A}}^{\mathcal{S}}$ and let $\eta \in \Delta_{\mathcal{S} \times \mathcal{A}}$ denote its corresponding discounted state-action frequency for some $\mu \in \Delta_{\mathcal{S}}$ and $\gamma \in (0, 1]$. Let $c \in \mathbb{R}, b \in \mathbb{R}^{\mathcal{S} \times \mathcal{A}}$ and set $S := \{s \in \mathcal{S} \mid b_{sa} \neq 0 \text{ for some } a \in \mathcal{A}\}$. Then*

$$\sum_{s,a} b_{sa}\tau_{sa} \geq c \quad \text{implies} \quad \sum_{s \in S}\sum_{a} b_{sa}\eta_{sa} \prod_{s' \in S \setminus \{s\}}\sum_{a'} \eta_{s'a'} - c \prod_{s' \in S}\sum_{a'} \eta_{s'a'} \geq 0,$$

*where the right is a multi-homogeneous polynomial[4] in the blocks $(\eta_{sa})_{a \in \mathcal{A}} \in \mathbb{R}^{\mathcal{A}}$ with multi-degree $\mathbb{1}_S \in \mathbb{N}^{\mathcal{S}}$. If further Assumption 13 holds, the inverse implication also holds.*

The preceding proposition shows that the state-action frequencies of a linearly constrained policy model, where the constraints only address the policy in individual states form a polytope. However, the effective policy polytope is almost never of this box type (see Remark 55).

**Example 15** (Blind controller). For a blind controller the linear equalities defining the effective policy polytope in $\Delta_{\mathcal{A}}^{\mathcal{S}}$ are $\tau_{s_1 a} - \tau_{s_2 a} = \tau(a|s_1) - \tau(a|s_2) = 0$ for all $a \in \mathcal{A}, s_1, s_2 \in \mathcal{S}$. They translate into the polynomial equalities $\eta_{s_1 a}\rho_{s_2} - \eta_{s_2 a}\rho_{s_1} = 0$ for all $a \in \mathcal{A}, s_1, s_2 \in \mathcal{S}$. In the case that $\mathcal{A} = \{a_1, a_2\}$, we obtain

$$0 = \eta_{s_1 a_1}(\eta_{s_2 a_1} + \eta_{s_2 a_1}) - \eta_{s_2 a_1}(\eta_{s_1 a_1} + \eta_{s_1 a_1}) = \eta_{s_1 a_1}\eta_{s_2 a_2} - \eta_{s_1 a_2}\eta_{s_2 a_1} \quad \text{for all } s_1, s_2 \in \mathcal{S},$$

which is precisely the condition that all $2 \times 2$ minors of $\eta$ vanish. Hence, in this case the set of state-action frequencies $\mathcal{N}_{\gamma}^{\mu,\beta}$ is given as the intersection of $\mathcal{N}_{\gamma}^{\mu}$ of state-action frequencies of the associated MDP and the determinantal variety of rank one matrices.

The following result describes the geometry of the set of feasible state-action frequencies.

**Theorem 16.** *Let $(\mathcal{S}, \mathcal{O}, \mathcal{A}, \alpha, \beta, r)$ be a POMDP, $\mu \in \Delta_{\mathcal{S}}$ and $\gamma \in (0, 1]$ and assume that Assumption 13 holds. Then we have $\mathcal{N}_{\gamma}^{\mu,\beta} = \mathcal{N}_{\gamma}^{\mu} \cap \mathcal{V} \cap \mathcal{B}$, where $\mathcal{V}$ is a variety described by multi-homogeneous polynomial equations and $\mathcal{B}$ is a basic semialgebraic set described by multi-homogeneous polynomial inequalities. Further, the face lattices of $\Delta_{\mathcal{A}}^{\mathcal{S},\beta}$ and $\mathcal{N}_{\gamma}^{\mu,\beta}$ are isomorphic.*

---

[4] A polynomial $p\colon \mathbb{R}^{n_1} \times \cdots \times \mathbb{R}^{n_k} \to \mathbb{R}$ is called *multi-homogeneous* with *multi-degree* $(d_1, \dots, d_k) \in \mathbb{N}^k$, if it is homogeneous of degree $d_j$ in the $j$-th block of variables for $j = 1, \dots, k$.

**Remark 17.** The variety $\mathcal{V}$ corresponds to the subspace $\mathcal{U}$ and the basic semialgebraic set $\mathcal{B}$ to the cone $\mathcal{C}$ from (3). Further, closed form expressions for the defining polynomials can be computed using Proposition 14 (see also Remark 18). The statement about isomorphic face lattices is in the sense that $\Delta_{\mathcal{A}}^{\mathcal{S},\beta}$ and $\mathcal{N}_{\gamma}^{\mu,\beta}$ have the same number of surfaces of a given dimension with the same neighboring properties. This can be seen in Figure 1, where the effective policy polytope and the set of state-action frequencies both have four vertices, four edges, and one two-dimensional face.

**Remark 18.** By Theorem 12 and Proposition 14, the defining polynomials of the basic semialgebraic set $\mathcal{B}$ from Theorem 16 are indexed by $a \in \mathcal{A}, o \in \mathcal{O}$ and are given by

$$p_{ao}(\eta) := \sum_{s \in S_o} \left( \beta_{os}^+ \eta_{sa} \prod_{s' \in S_o \setminus \{s\}} \sum_{a'} \eta_{s'a'} \right) = \sum_{f: S_o \to \mathcal{A}} \left( \sum_{s' \in f^{-1}(\{a\})} \beta_{os'}^+ \right) \prod_{s \in S_o} \eta_{sf(s)} \geq 0, \quad (4)$$

where $S_o := \{s \in \mathcal{S} \mid \beta_{os}^+ \neq 0\}$. The polynomials depend only on $\beta$ and not on $\gamma$, $\mu$ nor $\alpha$, and have $|S_o||\mathcal{A}|^{|S_o|-1}$ monomials of degree $|S_o|$ of the form $\prod_{s \in S_o} \eta_{sf(s)}$ for some $f: S_o \to \mathcal{A}$. In particular, we can read of the multi-degree of $p_{ao}$ with respect to the blocks $(\eta_{sa})_{a \in \mathcal{A}}$ which is given by $\mathbb{1}_{S_o}$ (see also Proposition 14). A complete description of the set $\mathcal{N}_{\gamma}^{\mu,\beta}$ via (in)equalities follows from the description of $\mathcal{N}_{\gamma}^{\mu}$ via linear (in)equalities given in (2). In Section 5 we discuss how the degree of these polynomials controls the complexity of the optimization problem.

**Remark 19** (Planning in POMDPs as a polynomial optimization problem)**.** The semialgebraic description of the set $\mathcal{N}_{\gamma}^{\mu,\beta}$ of feasible state-action distributions allows us to reformulate the reward maximization as a polynomially constrained optimization problem with linear objective (see also Remark 59 and Algorithm 1). This reformulation allows the use of constrained optimization algorithms, which we demonstrate in Appendix F on the toy example of Figure 1 and a grid world. Note that this polynomial program is different to the quadratic program obtained by Amato et al. (2006).

## 5 Number and Location of Critical Points

Although the reward function of MDPs is non convex, it still exhibits desirable properties from a standpoint of optimization. For example, without any assumptions, every policy can be continuously connected to an optimal policy by a path along which the reward is monotone (see Appendix D.5). Under mild conditions, all policies which are critical points of the reward function are globally optimal (Bhandari & Russo, 2019). In partially observable systems, the situation is fundamentally different. In this case, suboptimal local optima of the reward function can exist as can be seen in Figure 1 (see also Poupart et al., 2011; Bhandari & Russo, 2019). In the following we use the geometric description of the discounted state-action frequencies to study the number and location of critical points. These are important properties of the optimization problem and have implications on the required stochasticity of optimal policies. In Appendix D we discuss the mean reward case and an example and describe the sublevelsets as semialgebraic sets.

We regard the reward as a linear function $p_0$ over the set of feasible state-action frequencies $\mathcal{N}_{\gamma}^{\mu,\beta}$. Under Assumption 13 $\pi \mapsto \eta^{\pi}$ is injective and has a full-rank Jacobian everywhere (see Appendix C.1.1). Hence, the critical points in the policy polytope $\Delta_{\mathcal{A}}^{\mathcal{O}}$ correspond to the critical points of $p_0$ on $\mathcal{N}_{\gamma}^{\mu,\beta}$ (see Trager et al., 2019). In general, critical points of this linear function can occur on every face of the semialgebraic set $\mathcal{N}_{\gamma}^{\mu,\beta}$. The optimization problem thus has a combinatorial and a geometric component, corresponding to the number of faces of each dimension and the number of critical points in the relative interior of any given face. We have discussed the combinatorial part in Theorem 16 and focus now on the geometric part. Writing $\mathcal{N}_{\gamma}^{\mu,\beta} = \{\eta \in \mathbb{R}^{\mathcal{S} \times \mathcal{A}} \mid p_i(\eta) \leq 0, i \in I\}$, we are interested in the number of critical points on the interior of a face,

$$\mathrm{int}(F_J) = \{\eta \in \mathcal{N}_{\gamma}^{\mu,\beta} \mid p_j(\eta) = 0 \text{ for } j \in J, p_i(\eta) > 0 \text{ for } i \in I \setminus J\}.$$

Note that a point $\eta$ is critical on $\mathrm{int}(F_J)$, if and only if it is a critical point on the variety $\mathcal{V}_J := \{\eta \in \mathbb{R}^{\mathcal{S} \times \mathcal{A}} \mid p_j(\eta) = 0 \text{ for } j \in J\}$. For the sake of notation we write $J = \{1, \ldots, m\}$. We can bound the number of critical points in the interior of the face by the number of critical points of the polynomial optimization problem of optimizing $p_0(\eta)$ subject to $p_1(\eta) = \cdots = p_m(\eta) = 0$. This number is upper bounded by the algebraic degree of the problem which controls also the (algebraic) complexity of optimal policies (see Appendix D.1 for details). Using Theorem 12, Proposition 14 and an upper bound on the algebraic degree of polynomial optimization by Nie & Ranestad (2009) yields the following result.

**Theorem 20.** *Consider a POMDP $(\mathcal{S}, \mathcal{O}, \mathcal{A}, \alpha, \beta, r)$, $\gamma \in (0,1)$, assume that $r$ is generic, that $\beta \in \mathbb{R}^{\mathcal{S} \times \mathcal{O}}$ is invertible, and that Assumption 13 holds. For any given $I \subseteq \mathcal{A} \times \mathcal{O}$ consider the following set of policies, which is the relative interior of a face of the policy polytope:*

$$\mathrm{int}(F) = \left\{ \pi \in \Delta_{\mathcal{A}}^{\mathcal{O}} \mid \pi(a|o) = 0 \text{ if and only if } (a,o) \in I \right\}.$$

*Let $O := \{o \in \mathcal{O} \mid (a,o) \in I \text{ for some } a\}$ and set $k_o := |\{a \mid (a,o) \in I\}|$ as well as $d_o := |\{s \mid \beta_{os}^{-1} \neq 0\}|$. Then, the number of critical points of the reward function on $\mathrm{int}(F)$ is at most*

$$\left( \prod_{o \in O} d_o^{k_o} \right) \cdot \sum_{\sum_{o \in O} i_o = l} \prod_{o \in O} (d_o - 1)^{i_o}, \tag{5}$$

*where $l = |\mathcal{S}|(|\mathcal{A}| - 1) - |I|$. If $\alpha$ and $\mu$ are generic, this bound can be refined by computing the polar degrees of multi-homogeneous varieties (see Proposition 21 for a special case). The same bound holds in the mean reward case $\gamma = 1$ for $l$ given in Remark 61.*

By results from Montúfar & Rauh (2017) a POMDP has optimal memoryless stochastic policies with $|\operatorname{supp} \pi(\cdot|o)| \leq l_o$, where $l_o = |\operatorname{supp} \beta(o|\cdot)| \geq 1$. Hence, we may restrict attention to optimization over $\mathcal{N}_{\gamma}^{\mu,\beta}$ with $k = \sum_{o \in \mathcal{O}} k_o$ active inequalities (zeros in the policy), where $k_o = \max\{|\mathcal{A}| - l_o, 0\}$. Over these faces of the feasible set, the algebraic degree of the reward maximization problem is upper bounded by $\prod_{o \in \mathcal{O}} d_o^{k_o} \sum_{i_1 + \cdots + i_o = |\mathcal{S}|(|\mathcal{A}| - 1) - k} \prod_{o \in \mathcal{O}} (d_o - 1)^{i_o}$ due to Theorem 20.

In the special case of MDPs the bound shows that for MDPs only deterministic policies can be critical points of the reward function (see Corollary 62). Setting $I := \emptyset$ shows that there are no critical points in the interior of the policy polytope $\Delta_{\mathcal{A}}^{\mathcal{O}}$. This requires the assumption that $\beta$ is invertible (see Appendix D.4). The bound in Theorem 20 neglects the specific algebraic structure of the problem, and can be refined by considering polar degrees of determinantal varieties. This yields the following tighter upper bound for a blind controller with two actions (see Appendix D.3).

**Proposition 21** (Number of critical points in a blind controller)**.** *Let $(\mathcal{S}, \mathcal{O}, \mathcal{A}, \alpha, \beta, r)$ be a POMDP describing a blind controller with two actions, i.e., $\mathcal{O} = \{o\}$ and $\mathcal{A} = \{a_1, a_2\}$ and let $r, \alpha$ and $\mu$ be generic and let $\gamma \in (0,1)$. Then the reward function $R_{\gamma}^{\mu}$ has at most $|\mathcal{S}|$ critical points in the interior $\mathrm{int}(\Delta_{\mathcal{A}}^{\mathcal{O}}) \cong (0,1)$ of the policy polytope and hence at most $|\mathcal{S}| + 2$ critical points.*

In Appendix D.4 we provide examples of blind controllers which have several critical points in the interior $(0,1) \cong \mathrm{int}(\Delta_{\mathcal{A}}^{\mathcal{O}})$ and strict maxima at the two endpoints of the interval $[0,1] \cong \Delta_{\mathcal{A}}^{\mathcal{O}}$ respectively. Such points are called smooth and non-smooth critical points respectively.

# 6 CONCLUSION

We described geometric and algebraic properties of POMDPs and related the rational degree of the discounted state-action frequencies and the expected cumulative reward function to the degree of observability. We described the set of feasible state-action frequencies as a basic semialgebraic set and computed explicit expressions for the defining polynomials. In particular, this yields a polynomial programming formulation of POMDPs extending the linear programming formulation of MDPs. Based on this we use polynomial optimization theory to bound the number of critical points of the reward function over the polytope of memoryless stochastic policies. Our analysis also yields insights into the optimization landscape, such as the number of connected components of superlevel sets of the expected reward. Finally, we use a navigation problem in a grid world to demonstrate that the polynomial programming formulation can offer a computationally feasible approach to the reward maximization problem.

Our analysis focuses on infinite-horizon problems and memoryless policies with finite state, observation, and action spaces. Continuous spaces are interesting avenues, since they occur in real world application like robotics. The general bound on the number of critical points in Theorem 20 does not exploit the special multi-homogeneous structure of the problem, which could allow for tighter bounds as illustrated in Proposition 21 for blind controllers. Computing polar degrees is a challenging problem that remains to be studied using more sophisticated algebraic tools. Possible extensions of our work include the generalization to policies with finite memories as sketched in Appendix E.1. Further, we believe that it is interesting to explore to what extent our results can be used to identify policy classes guaranteed to contain maximizers of the reward in POMDPs.

ACKNOWLEDGMENTS

The authors thank Alex Tong Lin and Thomas Merkh for valuable discussions on POMDPs, Bernd Sturmfels for sharing his expertise on algebraic degrees and Mareike Dressler, Marina Garrote-López and Kemal Rose for their discussions on polynomial optimization. The authors acknowledge support by the ERC under the European Union's Horizon 2020 research and innovation programme (grant agreement no 757983). JM received support from the International Max Planck Research School for Mathematics in the Sciences and the Evangelisches Studienwerk Villigst e.V..

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

## APPENDIX

The Sections A–D of the Appendix correspond to the Sections 2–5 of the main body. We present the postponed proofs and elaborate various remarks in more detail. In Appendix E we discuss possible extensions of our results to memory policies and polynomial POMDPs. In Appendix F we provide details on the example plotted in Figure 1 and provide a plot of a three dimensional state-action frequency set.

## A  DETAILS ON THE PRELIMINARIES

We elaborate the proofs that where ommited or only sketched in the main body.

### A.1  PARTIALLY OBSERVABLE MARKOV DECISION PROCESSES

The statement of Proposition 1 can found in the work by Howard (1960) and we quickly sketch the proof therein. In order to show that the expected state-action frequencies exist without any

assumptions, we recall that for a (row or column) stochastic matrix $P$, the *Cesàro mean* is defined by

$$P^* := \lim_{T \to \infty} \frac{1}{T} \sum_{t=0}^{T-1} P^t$$

and exists without any assumptions. Further, $P^*$ is the projection onto the subspace of stationary distribution (Doob, 1953). For $\gamma \in (0, 1)$, the matrix

$$P_\gamma^* := (1 - \gamma) \sum_{t=0}^{\infty} \gamma^t P^t = (1 - \gamma)(I - \gamma P)^{-1}$$

is known as the *Abel mean* of $P$, where we used the Neumann series. By the Tauberian theorem, it holds that $P_\gamma^* \to P^*$ for $\gamma \nearrow 1$ (Gillette, 1958; Hunter, 1983).

**Proposition 1** (Existence of state-action frequencies and rewards). *Let $(\mathcal{S}, \mathcal{O}, \mathcal{A}, \alpha, \beta, r)$ be a POMDP, $\gamma \in (0, 1]$ and $\mu \in \Delta_\mathcal{S}$. Then $\eta_\gamma^{\pi,\mu}$, $\rho_\gamma^{\pi,\mu}$ and $R_\gamma^\mu(\pi)$ exist for every $\pi \in \Delta_\mathcal{A}^\mathcal{O}$ and $\mu \in \Delta_\mathcal{S}$ and are continuous in $\gamma \in (0, 1]$ for fixed $\pi$ and $\mu$.*

*Proof.* The existence of the state-action frequencies as well as the continuity with respect to the discount parameter follows directly from the general theory since

$$\eta_\gamma^{\pi,\mu} = (P_\pi^T)_\gamma^* (\mu * (\pi \circ \beta))$$

for $\gamma \in (0, 1)$ and $\eta_1^{\pi,\mu} = (P_\pi^T)^* (\mu * (\pi \circ \beta))$. With an analogue argument, the statement follows for the state frequencies and for the reward. $\square$

For $\gamma = 1$ we work under the following standard assumption in the (PO)MDP literature[5].

**Assumption 2** (Uniqueness of stationary disitributions). *If $\gamma = 1$, we assume that for any policy $\pi \in \Delta_\mathcal{A}^\mathcal{O}$ there exists a unique stationary distribution $\eta \in \Delta_{\mathcal{S} \times \mathcal{A}}$ of $P_\pi$.*

The following proposition shows in particular that for any initial distribution $\mu$, the infinite time horizon state-action frequency $\eta_1^{\pi,\mu}$ is the unique stationary distribution of $P_\pi$.

**Proposition 3** (State-action frequencies are discounted stationary). *Let $(\mathcal{S}, \mathcal{O}, \mathcal{A}, \alpha, \beta, r)$ be a POMDP, $\gamma \in (0, 1]$ and $\mu \in \Delta_\mathcal{S}$. Then $\eta_\gamma^{\pi,\mu}$ is the unique element in $\Delta_{\mathcal{S} \times \mathcal{A}}$ satisfying the discounted stationarity equation $\eta_\gamma^{\pi,\mu} = \gamma P_\pi^T \eta_\gamma^{\pi,\mu} + (1 - \gamma)(\mu * (\pi \circ \beta))$. Further, $\rho_\gamma^{\pi,\mu}$ is the unique element in $\Delta_\mathcal{S}$ satisfying $\rho_\gamma^{\pi,\mu} = \gamma p_\pi^T \rho_\gamma^{\pi,\mu} + (1 - \gamma)\mu$.*

*Proof.* By the general theory of Cesàro means, $(P_\pi^T)^*$ projects onto the space of stationary distributions and hence the $\eta_1^{\pi,\mu} = (P_\pi^T)^* (\mu * (\pi \circ \beta))$ is stationary. Hence, by Assumption 2, $\eta_1^{\pi,\mu}$ is the unique stationary distribution. For $\gamma \in (0, 1)$ we have

$$\eta_\gamma^{\pi,\mu} = (P_\pi^T)_\gamma^* (\mu * (\pi \circ \beta)) = (I - \gamma P_\pi^T)^{-1} (\mu * (\pi \circ \beta)),$$

which yields the claim. For the state distributions $\rho_\gamma^{\pi,\mu}$ the claim follows analogously or by marginalisation. $\square$

Since the state-action frequencies satisfy this generalized stationarity condition, we sometimes refer to them as *discounted stationary distributions*.

---

[5]Assumption 2 is weaker than ergodicity and is satisfied whenever the Markov chain with transition kernel $P^\pi$ is irreducible and aperiodic for every policy $\pi$, e.g., when the transition kernel satisfies $\alpha > 0$. For $\gamma \in (0, 1)$ the assumption is not required, since the discounted stationary distributions are always unique since $I - \gamma P_\pi$ is invertible because the spectral norm of $P_\pi$ is one.

## A.2 SEMIALGEBRAIC SETS AND THEIR FACE LATTICES

We recall the definition of semialgebraic sets, which are fundamental objects in real algebraic geometry (Bochnak et al., 2013). A *basic (closed) semialgebraic* set $A$ is a subset of $\mathbb{R}^d$ defined by finitely many polynomial inequalities as

$$A = \{x \in \mathbb{R}^d \mid p_i(x) \geq 0 \text{ for } i \in I\},$$

where $I$ is a finite index set and the $p_i$ are polynomials. A *semialgebraic* set is a finite union of basic (not necessarily closed) semialgebraic sets, and a function is called *semialgebraic* if its graph is semialgebraic. By the Tarski-Seidenberg theorem the range of a semialgebraic function is semialgebraic. A simple algebraic set has a lattice associated to it, induced by the set of active inequalities. More precisely, for a subset $J \subseteq I$ we set $F_J := \{x \in A \mid p_j(x) = 0 \text{ for } j \in J\}$ and endow the set $\mathcal{F} := \{F_J \mid J \subseteq I\}$ with the partial order of inclusion. We call the elements of $\mathcal{F}$ the *faces* of $A$. The faces described above are a generalization of the faces of a classical polytope and a special case of the faces of an abstract polytope and we refer to McMullen & Schulte (2002) for more details. Next, we want to endow this partially ordered set with more structure. A lattice $\mathcal{F}$ carries two operations, the *join* $\wedge$ and the *meet* $\vee$, which satisfy the absortion laws $F \vee (F \wedge G) = F$ and $F \wedge (F \vee G) = F$ for all $F, G \in \mathcal{F}$. In the face lattice of a basic semialgebraic set, the join and meet are given by

$$F \wedge G := F \cap G \quad \text{and} \quad F \vee G := \bigcap_{H \in \mathcal{F}: \ F, G \subseteq H} H.$$

A *morphism* between two lattices $\mathcal{F}$ and $\mathcal{G}$ is a mapping $\varphi \colon \mathcal{F} \to \mathcal{G}$ that respects the join and the meet, i.e., such that $\varphi(F \wedge G) = \varphi(F) \wedge \varphi(G)$ and $\varphi(F \vee G) = \varphi(F) \vee \varphi(G)$ for all $F, G \in \mathcal{F}$. A lattice *isomorphism* is a bijective lattice morphism where the inverse is also a morphism. We say that two basic semialgebraic sets with isomorphic face lattice are *combinatorially equivalent*.

# B DETAILS ON THE PARAMETRIZATON OF DISCOUNTED STATE-ACTION FREQUENCIES

## B.1 THE DEGREE OF DETERMINANTAL POLYNOMIALS

Determinantal representation of polynomials play an important role in convex geometry (see for example Helton & Vinnikov, 2007; Netzer & Thom, 2012), but often the emphasis is put on symmetric matrices. We adapt those arguments to the general case and present them here. We call $p$ a *determinantal polynomial* if it admits a representation

$$p(x) = \det\left(A_0 + \sum_{i=1}^m x_i A_i\right) \quad \text{for all } x \in \mathbb{R}^m, \tag{6}$$

for some $A_0, \dots, A_m \in \mathbb{R}^{n \times n}$. Let us use the notations

$$A(x) := A_0 + \sum_{i=1}^m x_i A_i \quad \text{and} \quad B(x) := \sum_{i=1}^m x_i A_i.$$

**Proposition 22** (Degree of monic univariate determinantal polynomials). *Let $A, B \in \mathbb{R}^{n \times n}$ and $A$ be invertible and let $\lambda_1, \dots, \lambda_n \in \mathbb{C}$ denote the eigenvalues of $A^{-1}B$ if repeated according to their algebraic multiplicity. Then,*

$$p \colon \mathbb{R} \to \mathbb{R}, \quad t \mapsto \det(A + tB)$$

*is a polynomial of degree*

$$\deg(p) = \big|\{j \in \{1, \dots, n\} \mid \lambda_j \neq 0\}\big| \leq \operatorname{rank}(B).$$

*The roots of $p$ are given by $\{-\lambda_j^{-1} \mid j \in J\} \subseteq \mathbb{C}$. If further $A^{-1}B$ is symmetric, then we have $\deg(p) = \operatorname{rank}(B)$.*

*Proof.* Let $J \subseteq \{1, \ldots, n\}$ denote the set of indices $j$ such that $\lambda_j \neq 0$. For $x \neq 0$ we have[6]

$$p(t) = \det(A) \det(I + tA^{-1}B) = x^n \det(A) \det(A^{-1}B + t^{-1}I) = x^n \det(A) \chi_{A^{-1}B}(-t^{-1})$$

$$= t^n \prod_{i=1}^{n} (-t^{-1} - \lambda_i) = (-1)^{n-|J|} \cdot \prod_{j \in J} (-\lambda_j) \cdot \prod_{j \in J} \left(t + \lambda_j^{-1}\right),$$

which is a polynomial of degree $|J|$. Note that $|J|$ is upper bounded by the complex rank of $A^{-1}B$. Since the rank over $\mathbb{C}$ and $\mathbb{R}$ agree for a real matrix, we have $\deg(p) \leq \operatorname{rank}(A^{-1}B) = \operatorname{rank}(B)$. Assume now that $A^{-1}B$ is symmetric, then the rank of $A^{-1}B$ coincides with the number $|J|$ of non zero eigenvalues. Further, the rank of $B$ and $A^{-1}B$ is the same. $\qquad\square$

**Remark 23.** Note that the degree of $p$ can be lower than $\operatorname{rank}(B)$, for example if

$$A = I \quad \text{and } B = \begin{pmatrix} 1 & -1 \\ 1 & -1 \end{pmatrix} = \begin{pmatrix} 1 \\ 1 \end{pmatrix} \begin{pmatrix} 1 & -1 \end{pmatrix}.$$

Then we have $\operatorname{rank}(B) = 1$, but

$$p(\lambda) = \det \begin{pmatrix} 1 + \lambda & -\lambda \\ \lambda & 1 - \lambda \end{pmatrix} = (1 + \lambda)(1 - \lambda) + \lambda^2 = 1$$

and therefore $\deg(p) = 0$. Note that in this case $A^{-1}B = B$ has no non-zero eigenvalues.

Now we show that the degree of $p$ is still bounded by $\operatorname{rank}(B)$ even if $A$ is not invertible. However, we loose an explicit description of the degree in this case.

**Proposition 24** (Degree of univariate determinantal polynomials). *Let $A, B \in \mathbb{R}^{n \times n}$ and consider the polynomial*

$$p \colon \mathbb{R} \to \mathbb{R}, \quad t \mapsto \det(A + tB).$$

*Then either $p = 0$ or if $p(t_0) \neq 0$ and $\lambda_1, \ldots, \lambda_n \in \mathbb{C}$ denote the eigenvalues of $(A + t_0 B)^{-1}B$ repeated according to their algebraic multiplicity, then $p$ has degree*

$$\deg(p) = \left| \{ j \in \{1, \ldots, n\} \mid \lambda_j \neq 0 \} \right| \leq \operatorname{rank}(B).$$

*In particular, it always holds that $\deg(p) \leq \operatorname{rank}(B)$.*

*Proof.* Let without loss of generality $p(t_0) \neq 0$, then $C = A + t_0 B$ is invertible. By Proposition 22, the degree of $q(t) = \det(C + tB)$ is precisely $|\{ j \mid \lambda_j \neq 0 \}|$. Noting that $p(t) = q(t - t_0)$ yields the claim. $\qquad\square$

The following result generalizes Proposition 24 to multivariate determinantal polynomials.

**Proposition 25** (Degree of determinantal polynomials). *Let $p \colon \mathbb{R}^m \to \mathbb{R}$ be a determinantal polynomial with the representation (6). Then*

$$\deg(p) \leq \max \left\{ \operatorname{rank}(B(x)) \mid x \in \mathbb{R}^m \right\}.$$

*Proof.* Let us fix $x \in \mathbb{R}^m$ and for $t \in \mathbb{R}$ set $f(t) := p(tx) = \det(A_0 + tB(x))$. By the next proposition it suffices to show that $\deg(f) \leq \operatorname{rank}(B(x))$. However, this is precisely the statement of Proposition 24. $\qquad\square$

**Proposition 26** (Degree of polynomials). *Let $p \colon \mathbb{R}^n \to \mathbb{R}$ be a polynomial. Then there is a direction $x \in \mathbb{R}^n$ such that $t \mapsto p(tx)$ is a polynomial of degree $\deg(p)$. Moreover, for any $x \in \mathbb{R}^n$, the univariate polynomial $t \mapsto p(tx)$ has degree at most $\deg(p)$.*

*Proof.* Let without loss of generality $p$ be non trivial. Decompose $p$ into its leading and lower order terms $p = p_1 + p_2$ and choose $x \in \mathbb{R}^n$ such that $p_1(x) \neq 0$. Let $k := \deg(p)$, then we have $p_1(tx) = t^k p_1(x)$ for all $\mu \in \mathbb{R}$. Since the degree of $t \mapsto p_2(tx)$ is at most $k - 1$, the degree of $t \mapsto p(tx) = p_1(tx) + p_2(tx)$ is $k$. $\qquad\square$

---

[6]Here, $\chi_C(\lambda) = \det(C - \lambda I)$ denotes the characteriztic polynomial of a matrix $C$.

**Remark 27.** Analogue to the univariate case, it is possible to give a precise statement on the degree, which is the following. If $p$ is not vanishing and $x_0 \in \mathbb{R}^m$ is such that $p(x_0) \neq 0$, then $A(x_0) \in \mathbb{R}^{n \times n}$ is invertible. Writing $\lambda_1(x), \ldots, \lambda_n(x) \in \mathbb{C}$ for the eigenvalues of $A(x_0)^{-1}B(x)$ and $J(x)$ for the indices $j$ such that $\lambda_j(x) \neq 0$ we obtain

$$\deg(p) = \max\left\{|J(x)| \mid x \in \mathbb{R}^m\right\}.$$

### B.2 The degree of POMDPs

The general bounds on the degree of determinantal polynomials directly implies Theorem 4, which we state again for the sake of convenience here.

**Theorem 4** (Degree of POMDPs). *Let $(\mathcal{S}, \mathcal{O}, \mathcal{A}, \alpha, \beta, r)$ be a POMDP, $\mu \in \Delta_\mathcal{S}$ be an initial distribution and $\gamma \in (0, 1)$ a discount factor. The state-action frequencies $\eta_\gamma^{\pi,\mu}$ and $\rho_\gamma^{\pi,\mu}$, the value function $V_\gamma^\pi$ and the expected cumulative reward $R_\gamma^\mu(\pi)$ are rational functions with common denominator in the entries of the policy $\pi$. Further, if they are restricted to the subset $\Pi \subseteq \Delta_\mathcal{A}^\mathcal{O}$ of policies which agree with a fixed policy $\pi_0$ on all states outside of $O \subseteq \mathcal{O}$, they have degree at most*

$$\left|\left\{s \in \mathcal{S} \mid \beta(o|s) > 0 \text{ for some } o \in O\right\}\right|.$$

In fact, the results from the preceding pararaph imply the following sharper versions.

**Theorem 28** (Degree of discounted state-action frequencies of POMDPs). *Let $(\mathcal{S}, \mathcal{O}, \mathcal{A}, \alpha, \beta, r)$ be a POMDP, $\mu \in \Delta_\mathcal{S}$ be an initial distribution and $\gamma \in (0, 1)$ a discount factor. Then the discounted state-action distributions can be expressed as*

$$\eta_\gamma^{\pi,\mu}(s, a) = \frac{q_{s,a}(\pi)}{q(\pi)} \quad \text{for every } \pi \in \Delta_\mathcal{A}^\mathcal{O} \text{ and } s \in \mathcal{S}, \tag{7}$$

*where*

$$q_{s,a}(\pi) := (\pi \circ \beta)(a|s) \cdot (1 - \gamma) \cdot \det(I - \gamma p_\pi^T)_s^\mu \quad \text{and} \quad q(\pi) := \det(I - \gamma p_\pi^T)$$

*are polynomials in the entries of the policy. Further, if $q_{s,a}$ and $q$ are restricted to the subset $\Pi \subseteq \Delta_\mathcal{A}^\mathcal{O}$ of policies which agree with a fixed policy $\pi_0$ on all states outside of $O \subseteq \mathcal{O}$ and if we set $S := \{s \in \mathcal{S} \mid \beta(o|s) > 0 \text{ for some } o \in O\}$, then they have degree at most*

$$\deg(q_{s,a}) \leq \max_{\pi \in \Pi} \operatorname{rank}(p_\pi^T - p_{\pi_0}^T)_s^0 + \mathbb{1}_S(s) \leq |S| \tag{8}$$

*and*

$$\deg(q) \leq \max_{\pi \in \Pi} \operatorname{rank}(p_\pi^T - p_{\pi_0}^T) \leq |S|. \tag{9}$$

*Proof.* Recall that we have

$$\eta_\gamma^{\pi,\mu}(s, a) = (\pi \circ \beta)(a|s)\rho_\gamma^{\pi,\mu}(s).$$

Further, by Proposition 3, the state distribution is given by $\rho_\gamma^{\pi,\mu} = (1 - \gamma)(1 - \gamma p_\pi^T)^{-1}\mu$. Applying Cramer's rule yields

$$\rho_\gamma^{\pi,\mu}(s) = \frac{\det(I - \gamma p_\pi^T)_s^\mu}{\det(I - \gamma p_\pi^T)},$$

where $(I - \gamma p_\pi^T)_s^\mu$ denotes the matrix obtained by replacing the $s$-th row of $I - \gamma p_\pi^T$ with $\mu$, which shows (7). For the estimates on the degree, we note that

$$\deg(q_{s,a}) \leq \deg(I - \gamma p_\pi^T)_s^\mu + \mathbb{1}_S(s).$$

Further, we can use Proposition 25 to estimate the degree over a subset $\Pi \subseteq \Delta_\mathcal{A}^\mathcal{S}$ of policies which agree with a fixed policy $\pi_0$ on all states outside of $O \subseteq \mathcal{O}$. We obtain

$$\deg(I - \gamma p_\pi^T)_s^\mu \leq \max_{\pi \in \Pi} \operatorname{rank}((I - \gamma p_{\pi_0}^T)_s^\mu - (I - \gamma p_\pi^T)_s^\mu) = \max_{\pi \in \Pi} \operatorname{rank} \gamma(p_\pi^T - p_{\pi_0}^T)_s^0,$$

which shows the first estimate in (8). To see the second inequality, we first assume that $s \in S$. Then $(p_\pi^T - p_{\pi_0}^T)_s^0$ has at most $|S| - 1$ non zero columns and hence rank at most $|S| - 1$. If $s \notin S$, then with the same argument, the rank of $(p_\pi^T - p_{\pi_0}^T)_s^0 = p_\pi^T - p_{\pi_0}^T$ is at most $|S|$ and the second estimate in (8) holds in both cases. The estimates in (9) follow with completely analoguous arguments. $\square$

**Remark 29.** The polynomial $q$ is independent of the initial distribution $\mu$, whereas the polynomials $q_{s,a}$ and therefore also their degrees depend on $\mu$. Further, Proposition 24 contains an exact expressions for the degree of the polynomials $q_{s,a}$ and $q$ depending on the eigenvalues of certain matrices.

**Corollary 30** (Degree of the reward and value function). *Theorem 28 also yields the rational degree of the reward and value function. Indeed, it holds that[7]*

$$R_\gamma^\mu(\pi) = \frac{\sum_{s,a} r(s,a) q_{s,a}(\pi)}{q(\pi)} = (1 - \gamma) \cdot \frac{\det(I - \gamma p_\pi + r_\pi \otimes \mu)}{\det(I - \gamma p_\pi)} - 1 + \gamma,$$

*where we used the matrix determinant lemma (Vrabel, 2016), and*

$$V_\gamma^\pi(s) = \frac{\det(I - \gamma p_\pi + r_\pi \otimes \delta_s)}{\det(I - \gamma p_\pi)} - 1 + \gamma.$$

*The degree of their denominator is bounded by (9). The degree of the numerator of $V_\gamma^\pi(s)$ is bounded by (8). Finally, the degree of the numerator of the reward $R_\gamma^\mu$ is bounded by the maximum degree of the numerators of $V_\gamma^\pi(s)$ over the support of $\mu$. An explicit formula for the rational degrees can be deduced from Remark 27.*

**Corollary 31** (Degree of curves). *Let $(\mathcal{S}, \mathcal{A}, \alpha, r)$ be an MDP, $\mu \in \Delta_\mathcal{S}$ be an initial distribution, $\gamma \in (0, 1)$ a discount factor and $r \in \mathbb{R}^{\mathcal{S} \times \mathcal{A}}$. Further, let $\pi_0, \pi_1 \in \Delta_\mathcal{A}^\mathcal{S}$ be two policies that disagree on $k$ states. Let $\eta_\lambda$ and $V_\lambda$ denote the discounted state-action frequencies and the value function belonging to the policy $\pi_\lambda := \pi_0 + \lambda(\pi_1 - \pi_0)$. Then both $\lambda \mapsto \eta_\lambda$ and $\lambda \mapsto V^\lambda$ are rational functions of degree at most $k$.*

### B.3 PROPERTIES OF DEGREE-ONE RATIONAL FUNCTIONS

#### B.3.1 A LINE THEOREM FOR DEGREE-ONE RATIONAL FUNCTIONS

First, we notice that certain degree-one rational functions map lines to lines which implies that they map polytopes to polytopes. Further, the extreme points of the range lie in the image of the extreme points which implies that degree-one rational functions are maximized in extreme points – just like linear functions.

**Definition 32.** We say that a function $f \colon \Omega \to \mathbb{R}^m$ is a *rational function of degree at most $k$ with common denominator* if it admits a representation of the form $f_i = p_i/q$ for polynomials $p_i$ and $q$ of degree at most $k$.

**Remark 33.** We have seen that the state-action frequencies, the reward function and the value function of POMDPs are rational functions of degree at most $|\mathcal{S}|$ with common denominator. In the case of MDPs and if a policy is fixed on all but $k$ states, it is a rational function with common denominator of degree at most $k$.

**Proposition 34.** *Let $\Omega \subseteq \mathbb{R}^d$ be convex and $f \colon \Omega \to \mathbb{R}^m$ be a rational function of degree at most one with common denominator and the representation $f_i(x) = p_i(x)/q(x)$ for affine linear functions $p_i, q$. Then, $f$ maps lines to lines. More precisely, if $x_0, x_1 \in \Omega$, then*

$$c \colon [0, 1] \to [0, 1], \quad \lambda \mapsto \frac{q(x_1)\lambda}{q(x_\lambda)} = \frac{q(x_1)\lambda}{(q(x_1) - q(x_0))\lambda + q(x_0)}$$

*is strictly increasing and satisfies*

$$f((1 - \lambda)x_0 + \lambda x_1) = (1 - c(\lambda))f(x_0) + c(\lambda)f(x_1) = f(x_0) + c(\lambda)(f(x_1) - f(x_0)). \quad (10)$$

*Further, $c$ is strictly convex if $|q(x_1)| < |q(x_0)|$, strictly concave if $|q(x_1)| > |q(x_0)|$ and linear if $|q(x_0)| = |q(x_1)|$.*

*Proof.* We set $x_\lambda := (1 - \lambda)x_0 + \lambda x_1$ and by explicit computation we obtain

$$f(x_\lambda) = \frac{p(x_\lambda)}{q(x_\lambda)} = \frac{(1 - \lambda)p(x_0) + \lambda p(x_1)}{q(x_\lambda)} = \frac{(1 - \lambda)q(x_0)}{q(x_\lambda)} \cdot f(x_0) + \frac{\lambda q(x_1)}{q(x_\lambda)} \cdot f(x_1).$$

---

[7]Here, $u \otimes v := uv^T$ denotes the Kronecker product.

Noting that

$$\frac{\lambda q(x_1)}{q(x_\lambda)} = \frac{\lambda q(x_1)}{(1-\lambda)q(x_0) + \lambda q(x_1)} = c(\lambda)$$

and

$$\frac{(1-\lambda)q(x_0)}{q(x_\lambda)} + \frac{\lambda q(x_1)}{q(x_\lambda)} = \frac{(1-\lambda)q(x_0) + \lambda q(x_1)}{q(x_\lambda)} = 1$$

yields (10). Finally, we differentiate and obtain

$$c'(\lambda) = \frac{q(x_0)q(x_1)}{q(x_\lambda)^2}. \tag{11}$$

Since $q$ has no root in $\Omega$ it follows that $q(x_0)$ and $q(x_1)$ have the same sign and hence $c'(\lambda) > 0$. Differentiating a second time yields

$$c''(\lambda) = -2q(x_0)q(x_1)(q(x_1) - q(x_0)) \cdot q(x_\lambda)^{-3}.$$

Using that $\mathrm{sgn}(q(x_\lambda)) = \mathrm{sgn}(q(x_0)) = \mathrm{sgn}(q(x_1))$ yields the assertion. $\square$

**Remark 35.** The formula (10) holds for all $\lambda \in \mathbb{R}$ for which $x_\lambda = \lambda x_0 + (1-\lambda)x_1 \in \Omega$.

**Proposition 36** (Level sets of degree one rational functions). *Let $\Omega \subseteq \mathbb{R}^d$ be convex and $f\colon \Omega \to \mathbb{R}$ be a rational function of degree at most one. Then, $L_\alpha := \{x \in \Omega \mid f(x) = \alpha\}$ is the intersection of an affine space with $\Omega$.*

*Proof.* For $x, y \in L_\alpha$ the ray $\{x + t(y-x) \mid t \in \mathbb{R}\} \cap \Omega$ is contained in $L_\alpha$ by the line theorem. $\square$

### B.3.2 EXTREME POINTS OF DEGREE-ONE RATIONAL FUNCTIONS

It is well known that linear functions obtain their maxima on extreme points. We show that this is also the case for rational functions of degree at most one.

**Definition 37.** Let $\Omega \subseteq \mathbb{R}^d$. Then we call $x \in \Omega$ an *extreme point* of $\Omega$ if $x$ is not the strict convex combination of two other points in $\Omega$, i.e., if $x = (1-\lambda)x_0 + \lambda x_1$ for $x_0, x_1 \in \Omega$ and $\lambda \in (0,1)$ implies $x_0 = x_1 = x$. We denote the set of extreme points of $\Omega$ by $\mathrm{extr}(\Omega)$.

**Proposition 38.** *Let $\Omega \subseteq \mathbb{R}^d$ be convex and $f\colon \Omega \to \mathbb{R}^m$ be a rational function of degree at most one with common denominator. Then $f(\Omega)$ is convex and we have $\mathrm{extr}(f(\Omega)) \subseteq f(\mathrm{extr}(\Omega))$.*

*Proof.* Let $y_0 = f(x_0), y_1 = f(x_1) \in f(\Omega)$. Then by the line theorem, the line connecting $y_0$ and $y_1$ agrees with the image of the line connecting $x_0$ and $x_1$ under $f$, in particular, it is contained in $f(\Omega)$ which shows the convexity of $f(\Omega)$. Pick now an extreme point $y = f(x) \in \mathrm{extr}(f(\Omega))$. If $x \in \mathrm{extr}(\Omega)$, there is nothing to show, so let $x \notin \mathrm{extr}(\Omega)$. Then by the Carathéodory theorem we can write $x$ as a strict convex combination $\sum_{i=1}^n \lambda_i x_i, \lambda_i > 0, n \geq 2$ for some extreme points $x_i \in \mathrm{extr}(\Omega)$. In particular, it is possible to write $x$ as the strict convex combination $x = (1-\lambda)x_0 + \lambda x_1$ by setting $x_0 := \sum_{i=2}^n \lambda_i x_i$. Now, by the line theorem we have

$$y = (1 - c(\lambda))f(x_0) + c(\lambda)f(x_1),$$

where $c(\lambda) \in (0,1)$. Since $y$ is an extreme point, this implies $f(x_0) = f(x_1) = y$. In particular, this shows that $y = f(x_1) \in f(\mathrm{extr}(\Omega))$. $\square$

**Corollary 39** (Maximizers of degree-one rational functions). *Let $\Omega \subseteq \mathbb{R}^d$ be a convex and compact set and let $f\colon \Omega \to \mathbb{R}$ be a rational function of degree at most one with common denominator. Then $f$ is maximized in at least one extreme point of $\Omega$. In particular, if $\Omega$ is a polytope, $f$ is maximized in at least one vertex.*

*Proof.* Since $\Omega$ is compact and $f$ is continuous, $f(\Omega)$ is a compact interval, lets say $f(\Omega) = [\alpha, \beta]$. By the preceding proposition we have $\{\alpha, \beta\} = \mathrm{extr}(f(\Omega)) \subseteq f(\mathrm{extr}(\Omega))$, which shows that $f$ is maximized in at least one extreme point. $\square$

**Corollary 40.** *Let $P \subseteq \mathbb{R}^d$ be a polytope and $f\colon \Omega \to \mathbb{R}^m$ be a rational function of degree at most one with common denominator. Then $f(P)$ is a polytope and we have $\mathrm{vert}(f(P)) \subseteq f(\mathrm{vert}(P))$.*

*Proof.* By the preceding proposition, $f(P)$ is convex. Further, $f(P)$ has finitely many extreme points since $\mathrm{extr}(f(P)) \subseteq f(\mathrm{extr}(P)) = f(\mathrm{vert}(P))$, which implies the assertion. □

**Proposition 41.** *Let* $f \colon P \to \mathbb{R}^m$ *be defined on the Cartesian product* $P = P_1 \times \cdots \times P_k$ *of polytopes, which is a degree-one rational function with common denominator whenever all but one components are fixed. Then* $f(P)$ *has finitely many extreme points and it holds that*

$$\mathrm{extr}(f(P)) \subseteq f(\mathrm{vert}(P)) = f(\mathrm{vert}(P_1) \times \cdots \times \mathrm{vert}(P_k)).$$

*In particular, if* $m = 1$ *this shows that* $f$ *is maximized in at least one vertex of* $P$.

*Proof.* Let now $x = (x^{(1)}, \ldots, x^{(k)}) \in P_1 \times \cdots \times P_k$ be such that $f(x) \in \mathrm{extr}(f(P))$. If $x^{(i)} \in \mathrm{vert}(P_i)$, there is nothing to show. Hence, we assume that $x^{(i)} \notin \mathrm{vert}(P_i)$. Let us denote the restriction of $f$ onto $P_i$ by $g$, where we keep the other components fixed to be $x^{(j)}$. Then we have $g(x^{(i)}) \in \mathrm{extr}(g(P_i))$ and hence by Proposition 38 there is $\tilde{x}^{(i)} \in \mathrm{vert}(P_i)$ such that $g(\tilde{x}^{(i)}) = g(x^{(i)}) = f(x)$. Replacing $x^{(i)}$ by $\tilde{x}^{(i)}$ and iterating over $i$ yields the claim. □

**Remark 42.** We have seen that both the value function as well as the discounted state-action frequencies are degree-one rational functions in the rows of the policy in the case of full observability. Hence, the extreme points of the set of all value functions and of the set of discounted state-action frequencies are described by the proposition above. In fact we will see later that the discounted state-action frequencies form a polytope; further, one can show that the set of value functions is a finite union of polytopes (see Dadashi et al., 2019).

### B.3.3 Implications for POMDPs

**Proposition 6.** *Let* $(\mathcal{S}, \mathcal{A}, \alpha, r)$ *be an MDP and* $\gamma \in (0, 1)$. *Further, let* $\pi_0, \pi_1 \in \Delta_{\mathcal{A}}^{\mathcal{S}}$ *be two policies that differ on at most* $k$ *states. For any* $\lambda \in [0, 1]$ *let* $V_\lambda \in \mathbb{R}^{\mathcal{S}}$ *and* $\eta_\lambda^\mu \in \Delta_{\mathcal{S} \times \mathcal{A}}$ *denote the value function and state-action frequency belonging to the policy* $\pi_0 + \lambda(\pi_1 - \pi_0)$ *with respect to the discount factor* $\gamma$, *the initial distribution* $\mu$ *and the instantaneous reward* $r$. *Then the rational degrees of* $\lambda \mapsto V_\lambda$ *and* $\lambda \mapsto \eta_\lambda$ *are at most* $k$. *If they differ on at most one state* $\tilde{s} \in \mathcal{S}$ *then*

$$V_\lambda = V_0 + c(\lambda) \cdot (V_1 - V_0) \quad and \quad \eta_\lambda^\mu = \eta_0^\mu + c(\lambda) \cdot (\eta_1^\mu - \eta_0^\mu) \quad for\ all\ \lambda \in [0, 1],$$

*where*

$$c(\lambda) = \frac{\det(I - \gamma p_1)\lambda}{\det(I - \gamma p_\lambda)} = \frac{\det(I - \gamma p_1)\lambda}{(\det(I - \gamma p_1) - \det(I - \gamma p_0))\lambda + \det(I - \gamma p_0)} = \lambda \cdot \frac{\rho_\lambda^\mu(\tilde{s})}{\rho_1^\mu(\tilde{s})}.$$

*Proof.* This is a direct consequence of Proposition 34 and Theorem 4. □

**Remark 43.** The proposition above describes the *interpolation speed* $\lambda \cdot \rho_\lambda^\mu(\tilde{s})/\rho_1^\mu(\tilde{s})$ in terms of the discounted state distribution in $\tilde{s}$. This expressions extends to the case of mean rewards – note that the determinants vanish – and the theorem can be shown to hold in this case as well, if we set $0/0 := 0$. Note that the interpolation speed does not depend on the initial condition $\mu$.

**Remark 44.** Refinements on the upper bound of the rational degree of $\lambda \mapsto V_\lambda$ and $\lambda \mapsto \eta_\lambda$ can be obtained using Proposition 24. Indeed, if we write $\eta_\lambda(s, a) = q_{sa}(\lambda)/q(\lambda)$ like in Theorem 28 those degrees can be upper bounded by

$$\deg(q_{sa}) \le \mathrm{rank}(p_1 - p_0)_s^0 + \mathbb{1}_S(s) \le \mathrm{rank}(p_1 - p_0) \quad and \quad \deg(q) \le \mathrm{rank}(p_1 - p_0),$$

where $S \subseteq \mathcal{S}$ is the set of states on which the two policies differ; see also the proof of Theorem 28 for more details on an analogue argument. Hence, the degree of the two curves $\lambda \mapsto V_\lambda$ and $\lambda \mapsto \eta_\lambda$ is upper bounded by $\mathrm{rank}(p_1 - p_0)$.

## C Details on the Geometry of State-Action Frequencies

The set of all state-action frequencies is known to be a polytope in the fully observable case (Derman, 1970) and we show that it is combinatorially equivalent to the conditional probability polytope $\Delta_{\mathcal{A}}^{\mathcal{S}}$. We show that in the partially observable case the set of feasible state-action frequencies is cut out from this polytope by a finite set of polynomial inequalities. We discuss the special structure of those polynomials and give closed form expressions for them.

## C.1 THE FULLY OBSERVABLE CASE

Let $\nu_\gamma^{\pi,\mu} \in \Delta_{\mathcal{S}\times\mathcal{S}}$ denote the expected number of transitions from $s$ to $s'$ given by

$$(1-\gamma)\sum_{t=0}^{\infty}\gamma^t\mathbb{P}^{\pi,\mu}(s_t = s, s_{t+1} = s') \quad \text{and} \quad \lim_{T\to\infty}\frac{1}{T}\sum_{t=0}^{T-1}\mathbb{P}^{\pi,\mu}(s_t = s, s_{t+1} = s')$$

respectively. Note that we have

$$\nu_\gamma^{\pi,\mu}(s,s') = \sum_{a\in\mathcal{A}}\eta_\gamma^{\pi,\mu}(s,a)\alpha(s'|s,a), \tag{12}$$

hence $\nu_\gamma^{\pi,\mu}$ is the image of $\eta_\gamma^{\pi,\mu}$ under the linear transformation

$$f_\alpha\colon \Delta_{\mathcal{S}\times\mathcal{A}} \to \Delta_{\mathcal{S}\times\mathcal{S}}, \quad \eta \mapsto \left(\sum_{a\in\mathcal{A}}\eta(s,a)\alpha(s'|s,a)\right)_{s,s'\in\mathcal{S}}. \tag{13}$$

Therefore, we can hope to obtain a characterization of $\mathcal{N}_\gamma^\mu$ using this mapping. In order to do so, we would like to understand the structural properties of $\nu_\gamma^{\pi,\mu}$. For $\gamma = 1$ those distributions have equal marginals since we can compute

$$\sum_{s'\in\mathcal{S}}\nu_1^{\pi,\mu}(s,s') - \sum_{s'\in\mathcal{S}}\nu_1^{\pi,\mu}(s',s) = \lim_{T\to\infty}\frac{1}{T}\big(\mathbb{P}^{\pi,\mu}(s_0 = s) - \mathbb{P}^{\pi,\mu}(s_{T+1} = s)\big) = 0. \tag{14}$$

In the discounted case, we compute similarly

$$\sum_{s'\in\mathcal{S}}\nu_\gamma^{\pi,\mu}(s,s') - \gamma\sum_{s'\in\mathcal{S}}\nu_\gamma^{\pi,\mu}(s',s) = (1-\gamma)\left(\sum_{t=0}^{\infty}\gamma^t\mathbb{P}^{\pi,\mu}(s_t = s) - \sum_{t=0}^{\infty}\gamma^{t+1}\mathbb{P}^{\pi,\mu}(s_{t+1} = s)\right)$$

$$= (1-\gamma)\mu(s).$$

If we perceive $\nu_\gamma^{\pi,\mu} \in \Delta_{\mathcal{S}\times\mathcal{S}}$ as a matrix, we have shown that

$$(\nu_\gamma^{\pi,\mu})^T\mathbb{1}_\mathcal{S} = \gamma(\nu_\gamma^{\pi,\mu})\mathbb{1}_\mathcal{S} + (1-\gamma)\mu,$$

which motivates the following definition.

We will see that the set of state-action frequencies is the pre-image of the following polytope under a linear map.

**Definition 45** (Discounted Kirchhoff polytopes). For a distribution $\mu \in \Delta_\mathcal{S}$ and $\gamma \in (0,1]$ we define the *discounted Kirchhoff polytope* (this is a generalization of a definition by Weis 2010)

$$\Xi_\gamma^\mu := \big\{\nu \in \Delta_{\mathcal{S}\times\mathcal{S}} \subseteq \mathbb{R}^{\mathcal{S}\times\mathcal{S}} \mid \nu^T\mathbb{1}_\mathcal{S} = \gamma\nu\mathbb{1}_\mathcal{S} + (1-\gamma)\mu\big\},$$

where $\mathbb{1}_\mathcal{S} \in \mathbb{R}^\mathcal{S}$ is the all one vector.

So far, we have observed that $f_\alpha(\eta_\gamma^{\pi,\mu}) \in \Xi_\gamma^\mu$, i.e., that

$$\Psi_\gamma^\mu\colon \Delta_\mathcal{A}^\mathcal{S} \to \Delta_{\mathcal{S}\times\mathcal{A}}, \quad \pi \mapsto \eta_\gamma^{\pi,\mu}$$

maps to $f_\alpha^{-1}(\Xi_\gamma^\mu)$. In order to see that this mapping is surjective on $f_\alpha^{-1}(\Xi_\gamma^\mu)$ we show that its right inverse is given through conditioning. The following proposition uses the ergodicity assumption.

**Proposition 46.** *Let $\gamma \in (0,1]$ and $\eta \in f_\alpha^{-1}(\Xi_\gamma^\mu)$ and let $\rho \in \Delta_\mathcal{S}$ denote the state marginal of $\eta$. Set*

$$\pi(\cdot|s) := \begin{cases} \eta(\cdot|s) = \eta(s,\cdot)/\rho(s) & \text{if } \rho(s) > 0 \\ \text{arbitrary element in } \Delta_\mathcal{A} & \text{if } \rho(s) = 0, \end{cases}$$

*then we have $\eta_\gamma^{\pi,\mu} = \eta$.*

*Proof.* We calculate

$$\gamma(P^\pi)^T\eta(s,a) = \gamma\sum_{s',a'}\alpha(s|s',a')\pi(a|s)\eta(s',a') = \gamma\pi(a|s)\sum_{s',a'}\alpha(s|s',a')\eta(s',a')$$

$$= \gamma\pi(a|s)\sum_{s'}\nu(s',s) = \pi(a|s)\left(\sum_{s'}\nu(s,s') - (1-\gamma)\mu(s)\right)$$

$$= \pi(a|s)\rho(s) - (1-\gamma)\pi(a|s)\mu(s) = \eta(s,a) - (1-\gamma)(\mu * \pi)(s,a).$$

The unique characterization from Proposition 3 of $\eta_\gamma^{\pi,\mu}$ yields the assertion. $\qquad\square$

The proposition states that we can reconstruct the policy from the state-action frequencies by conditioning and is well known in the context of the dual linear programming formulation of MDPs (Kallenberg, 1994). Hence, it will be convenient later to work under the following assumption in which ensures that policies in $\Delta_{\mathcal{A}}^{\mathcal{S}}$ are one-to-one with state-action frequencies.

**Assumption 13** (Positivity). Let $\rho_{\gamma}^{\pi,\mu} > 0$ hold entrywise for all policies $\pi \in \Delta_{\mathcal{A}}^{\mathcal{S}}$.

Note that this positivity assumption holds in particular, if either $\alpha > 0$ and $\gamma > 0$ or $\gamma < 1$ and $\mu > 0$ or entrywise. Indeed, if $\alpha > 0$, then the transition kernel $p_{\pi}$ is strictly positive for any policy since

$$p_{\pi}(s'|s) = \sum_a (\pi \circ \beta)(a|s)\alpha(s'|s,a) > 0,$$

since $(\pi \circ \beta)(a|s) > 0$ for some $a \in \mathcal{A}$. Since $\rho_{\gamma}^{\pi,\mu}$ is discounted stationary with respect to $p_{\pi}$ (Proposition 3), it holds that

$$\rho_{\gamma}^{\pi,\mu}(s) = \gamma \sum_{s'} \rho_{\gamma}^{\pi,\mu}(s')p_{\pi}(s|s') + (1-\gamma)\mu(s) > 0$$

since $\rho_{\gamma}^{\pi,\mu}(s') > 0$ for some $s' \in \mathcal{S}$. If $\mu > 0$ and $\gamma < 1$, then $\rho_{\gamma}^{\pi,\mu}(s) \geq (1-\gamma)\mu(s) > 0$. As a consequence of Proposition 46, we obtain the following characterization of $\mathcal{N}_{\gamma}^{\mu}$.

**Proposition 8** (Characterization of $\mathcal{N}_{\gamma}^{\mu}$). *Let $(\mathcal{S}, \mathcal{A}, \alpha, r)$ be an MDP, $\mu \in \Delta_{\mathcal{S}}$ be an initial distribution and $\gamma \in (0,1]$. It holds that*

$$\mathcal{N}_{\gamma}^{\mu} = \Delta_{\mathcal{S}\times\mathcal{A}} \cap \left\{\eta \in \mathbb{R}^{\mathcal{S}\times\mathcal{A}} \mid \langle w_{\gamma}^s, \eta\rangle_{\mathcal{S}\times\mathcal{A}} = (1-\gamma)\mu_s \text{ for } s \in \mathcal{S}\right\} \tag{2}$$

*where $w_{\gamma}^s := \delta_s \otimes \mathbb{1}_{\mathcal{A}} - \gamma\alpha(s|\cdot,\cdot)$. For $\gamma \in (0,1)$, $\Delta_{\mathcal{S}\times\mathcal{A}}$ can be replaced by $[0,\infty)^{\mathcal{S}\times\mathcal{A}}$ in (2).*

Instead of proving this proposition directly, we first present the following version of it.

**Proposition 47.** *Let $(\mathcal{A}, \mathcal{S}, \alpha, r)$ be an MDP and $\gamma \in (0,1]$. It holds that $\mathcal{N}_{\gamma}^{\mu} = f_{\alpha}^{-1}(\Xi_{\gamma}^{\mu})$.*

*Proof.* By (13) and (14), it holds that $f_{\alpha}(\mathcal{N}_{\gamma}^{\mu}) \subseteq \Xi_{\gamma}^{\mu}$ and thus $\mathcal{N}_{\gamma}^{\mu} \subseteq f_{\alpha}^{-1}(\Xi_{\gamma}^{\mu})$. However, by Proposition 46, for every $\eta \in f_{\alpha}(\Xi_{\gamma}^{\mu})$ there is a policy $\pi \in \Delta_{\mathcal{A}}^{\mathcal{S}}$ such that $\eta_{\gamma}^{\pi,\mu} = \eta$ and hence it holds that $f_{\alpha}^{-1}(\Xi_{\gamma}^{\mu}) \subseteq f_{\alpha}^{-1}(\Xi_{\gamma}^{\mu})$. $\square$

*Proof of Proposition 8.* By the preceding proposition $\eta \in \mathcal{N}_{\gamma}^{\mu}$ is equivalent to $\eta \in \Delta_{\mathcal{S}\times\mathcal{A}}$ and $\nu := f_{\alpha}(\eta) \in \Xi_{\gamma}^{\mu}$. Using the definition of $\Xi_{\gamma}^{\mu}$ this equivalent to

$$\sum_{s'} \nu(s,s') = \gamma\sum_{s'}\nu(s',s) + (1-\gamma)\mu(s)$$

for all $s \in \mathcal{S}$. Plugging in the definition of $f_{\alpha}$ we see that the term on the left hand side is equivalent to

$$\sum_{s'}\sum_a \eta(s,a)\alpha(s'|s,a) = \sum_a \eta(s,a) = \langle\delta_s \otimes \mathbb{1}_{\mathcal{A}}, \eta\rangle_{\mathcal{S}\times\mathcal{A}}.$$

The first term of the right hand side is precisely

$$\gamma\sum_{s'}\sum_a \eta(s',a)\alpha(s|s',a) = \langle\gamma\alpha(s|\cdot,\cdot), \eta\rangle_{\mathcal{S}\times\mathcal{A}}.$$

Hence, we have seen that $f_{\alpha}(\eta) \in \Xi_{\gamma}^{\mu}$ is equivalent to the condition

$$\langle w_{\gamma}^s, \eta\rangle_{\mathcal{S}\times\mathcal{A}} = (1-\gamma)\mu(s) \quad \text{for all } s \in \mathcal{S}. \tag{15}$$

Note that

$$\{\eta \mid \langle w_{\gamma}^s, \eta\rangle_{\mathcal{S}\times\mathcal{A}} = (1-\gamma)\mu(s) \text{ for all } s \in \mathcal{S}\} = \eta_0 + \{w_{\gamma}^s \mid s \in \mathcal{S}\}^{\perp}$$

for an arbitrary element $\eta_0$ satisfying (15). This shows the first equation in (2). The second equation follows from the observation that $\sum_s w_{\gamma}^s = (1-\gamma)\mathbb{1}_{\mathcal{S}}$. Hence, for $\gamma < 1$ it holds that

$$\left(\eta_0 + \{w_{\gamma}^s \mid s \in \mathcal{S}\}^{\perp}\right) \cap \Delta_{\mathcal{S}\times\mathcal{A}} = \left(\eta_0 + \left(\{w_{\gamma}^s \mid s \in \mathcal{S}\} \cup \{\mathbb{1}_{\mathcal{S}}\}^{\perp}\right) \cap [0,\infty)^{\mathcal{S}\times\mathcal{A}}\right)$$

$$= \left(\eta_0 + \{w_{\gamma}^s \mid s \in \mathcal{S}\}^{\perp}\right) \cap [0,\infty)^{\mathcal{S}\times\mathcal{A}}.$$

$\square$

C.1.1 DERIVATIVE OF THE DISCOUNTED STATE-ACTION FREQUENCIES

In this section we discuss the Jacobian of the parametrization $\pi \mapsto \eta^\pi$ of the discounted state-action frequencies. One motivation for this is that this Jacobian plays an important role in the relation of critical points in the policy space and the space of discounted state-action frequencies. Note that $\Psi_\gamma^\mu(\pi) = (1-\gamma)(1-\gamma P_\pi^T)^{-1}(\mu * \pi)$ is well defined, whenever $\|P_\pi\|_2 < \gamma^{-1}$. Hence, we can extend $\Psi_\gamma^\mu$ onto the neighborhood $\{\pi \in \mathbb{R}^{\mathcal{S} \times \mathcal{A}} \mid \|P_\pi\|_2 < \gamma^{-1}\}$ of $\Delta_{\mathcal{A}}^{\mathcal{S}}$, which enables us to compute the Jacobian of $\Psi_\gamma^\mu$.

**Proposition 48** (Jacobian of $\Psi_\gamma^\mu$). *For any policy $\pi \in \Delta_{\mathcal{A}}^{\mathcal{S}}$ and $s \in \mathcal{S}, a \in \mathcal{A}$ it holds that*

$$\partial_{(s,a)} \Psi_\gamma^\mu(\pi) = (I - \gamma P_\pi^T)^{-1}(\rho_\gamma^{\pi,\mu} * \partial_{(s,a)} \pi) = \rho_\gamma^{\pi,\mu}(s)(I - \gamma P_\pi^T)^{-1}(\delta_s \otimes \delta_a), \qquad (16)$$

*where*

$$(\rho_\gamma^{\pi,\mu} * \partial_{(s,a)} \pi)(s', a') = \rho_\gamma^{\pi,\mu}(s')\partial_{(s,a)} \pi(a'|s') = \rho_\gamma^{\pi,\mu}(s)(\delta_s \otimes \delta_a)(s', a').$$

*Hence, $\partial_{(s,a)} \Psi_\gamma^\mu(\pi)$ is identical to the $(s, a)$-th column of $(I - \gamma P_\pi^T)^{-1}$ up to the scaling factor of $\rho_\gamma^{\pi,\mu}(s)$. In particular, if $\rho_\gamma^{\pi,\mu}(s) > 0$ for all $s \in \mathcal{S}$, the Jacobian $D\Psi_\gamma^\mu$ has full rank.*

*Proof.* Recall that for invertible matrices $A(t)$, it holds that $\partial_t A(t)^{-1} = -A(t)^{-1}(\partial_t A(t))A(t)^{-1}$. We compute

$$
\begin{aligned}
(1 - \gamma)^{-1} \partial_{(s,a)} \Psi_\gamma^\mu(\pi) &= \partial_{(s,a)}(I - \gamma P_\pi^T)^{-1}(\mu * \pi) \\
&= (\partial_{(s,a)}(I - \gamma P_\pi^T)^{-1})(\mu * \pi) + (I - \gamma P_\pi^T)^{-1}\partial_{(s,a)}(\mu * \pi) \\
&= -(1 - \gamma)^{-1}(I - \gamma P_\pi^T)^{-1}\partial_{(s,a)}(I - \gamma P_\pi^T)\eta_\gamma^{\pi,\mu} \\
&\quad + (I - \gamma P_\pi^T)^{-1}(\mu * \partial_{(s,a)} \pi) \\
&= (I - \gamma P_\pi^T)^{-1}\left((1 - \gamma)^{-1}\gamma(\partial_{(s,a)} P_\pi^T)\eta_\gamma^{\pi,\mu} + \mu * \partial_{(s,a)} \pi\right).
\end{aligned}
$$

Further, direct computation shows

$$
\begin{aligned}
((\partial_{(s,a)} P_\pi^T)\eta_\gamma^{\pi,\mu})(s, a) &= \partial_{(s,a)} \pi(a|s) \sum_{s',a'} \alpha(s|s', a')\pi(a'|s')\rho_\gamma^{\pi,\mu}(s') \\
&= (p_\pi^T \rho_\gamma^{\pi,\mu} * \partial_{(s,a)} \pi)(s, a).
\end{aligned}
$$

Using the fact that $\rho_\gamma^{\pi,\mu}$ is the discounted stationary distribution, yields

$$
\begin{aligned}
(1 - \gamma)^{-1}\gamma(\partial_{(s,a)} P_\pi^T)\eta_\gamma^{\pi,\mu} + \mu * \partial_{(s,a)} \pi &= ((1 - \gamma)^{-1}\gamma p_\pi^T \rho_\gamma^{\pi,\mu} + \mu) * \partial_{(s,a)} \pi \\
&= (1 - \gamma)^{-1}\rho_\gamma^{\pi,\mu} * \partial_{(s,a)} \pi,
\end{aligned}
$$

which shows (16). Note that $(I - \gamma P_\pi^T)^{-1}(\delta_s \otimes \delta_a)$ is precisely the $(s_0, a_0)$-th column of the matrix $(I - \gamma P_\pi^T)^{-1}$. Those columns are linearly independent, and so are the partial derivatives $\partial_{(s,a)} \Psi_\gamma^\mu(\pi)$, given that the discounted stationary distribution $\rho_\gamma^{\pi,\mu}$ vanishes nowhere. □

**Corollary 49** (Dimension of $\mathcal{N}_\gamma^\mu$). *Assume that $\rho_\gamma^{\pi,\mu} > 0$ entrywise for some policy $\pi \in \mathrm{int}(\Delta_{\mathcal{A}}^{\mathcal{S}})$. Then we have*

$$\dim(\mathcal{N}_\gamma^\mu) = \dim(\Delta_{\mathcal{A}}^{\mathcal{S}}) = |\mathcal{S}|(|\mathcal{A}| - 1).$$

*Proof.* By Proposition 48 the mapping $\Psi_\gamma^\mu$ is full rank in a neighborhood of $\pi$ and hence, we have

$$\dim(\mathcal{N}_\gamma^\mu) = \dim(\Psi_\gamma^\mu(\Delta_{\mathcal{A}}^{\mathcal{S}})) = \dim(\Delta_{\mathcal{A}}^{\mathcal{S}}).$$

□

Let us consider a parametrized policy model $\Pi_\Theta = \{\pi_\theta \mid \theta \in \Theta\}$ with differentiable parametrization $\theta \mapsto \pi_\theta$.

**Proposition 50** (Parameter derivatives of discounted state-action frequencies). *It holds that*

$$\partial_{\theta_i} \eta_\gamma^{\pi_\theta,\mu} = (I - \gamma P_{\pi_\theta}^T)^{-1}(\rho_\gamma^{\pi_\theta,\mu} * \partial_{\theta_i} \pi_\theta), \quad \text{where } (\rho_\gamma^{\pi_\theta,\mu} * \partial_{\theta_i} \pi_\theta)(s, a) = \rho_\gamma^{\pi_\theta,\mu}(s)\partial_{\theta_i} \pi_\theta(a|s).$$

*Proof.* This follows directly from the application of the chain rule and (16). □

Using this expression, we can compute the parameter gradient with respect to the discounted reward $F(\theta) \coloneqq R_\gamma^\mu(\pi_\theta)$ and recover the well known policy gradient theorem, see Sutton et al. (1999).

**Definition 51** (state-action value function). We call $Q^\pi \coloneqq (I - \gamma P_\pi)^{-1}r \in \mathbb{R}^{\mathcal{S} \times \mathcal{A}}$ the *state-action value function* or the *Q-value function* of the policy $\pi$.

**Corollary 52** (Policy gradient theorem). *It holds that*

$$\partial_{\theta_i} F(\theta) = \sum_s \rho_\gamma^{\pi_\theta,\mu}(s) \sum_a \partial_{\theta_i} \pi_\theta(a|s) Q^{\pi_\theta}(s,a) = \sum_{s,a} \eta_\gamma^{\pi_\theta,\mu}(s,a) \partial_{\theta_i} \log(\pi_\theta(a|s)) Q^{\pi_\theta}(s,a).$$

*Proof.* Using the preceding proposition, we compute

$$\partial_{\theta_i} F(\theta) = \langle \rho_\gamma^{\pi_\theta,\mu} * \partial_{\theta_i} \pi_\theta, Q^{\pi_\theta} \rangle_{\mathcal{S} \times \mathcal{A}} = \sum_s \rho_\gamma^{\pi_\theta,\mu}(s) \sum_a \partial_{\theta_i} \pi_\theta(a|s) Q^{\pi_\theta}(s,a)$$

$$= \sum_{s,a} \eta_\gamma^{\pi_\theta,\mu}(s,a) \partial_{\theta_i} \log(\pi_\theta(a|s)) Q^{\pi_\theta}(s,a).$$

□

**Remark 53** (POMDPs as parametrized policy models). The case of partial observability can sometimes be regarded as a special case of parametrized policies. In fact the observation mechanism $\beta$ induces a linear map $\pi \mapsto \pi \circ \beta$. This interpretation together with the preceding proposition can be used to calculate policy gradients in partially observable systems.

### C.1.2 THE FACE LATTICE IN THE FULLY OBSERVABLE CASE

So far, we have seen that the set of state-action frequencies form a polytope in the fully observable case. However, not all polytopes are equally complex and thus we aim to describe the *face lattice* of $\mathcal{N}_\gamma^\mu$, which describes the combinatorial properties of a polytope, see Ziegler (2012).

**Theorem 54** (Combinatorial equivalence of $\mathcal{N}_\gamma^\mu$ and $\Delta_\mathcal{A}^\mathcal{S}$). *Let $(\mathcal{A}, \mathcal{S}, \alpha, r)$ be an MDP and $\gamma \in (0,1]$. Then $\pi \mapsto \eta_\gamma^{\pi,\mu}$ induces an order preserving surjective morphism between the face lattices of $\Delta_\mathcal{A}^\mathcal{S}$ and $\mathcal{N}_\gamma^\mu$, such that for every $I \subseteq \mathcal{S} \times \mathcal{A}$ it holds that*

$$\left\{ \pi \in \Delta_\mathcal{A}^\mathcal{S} \mid \pi(a|s) = 0 \text{ for all } (s,a) \in I \right\} \mapsto \left\{ \eta \in \mathcal{N}_\gamma^\mu \mid \eta(s,a) = 0 \text{ for all } (s,a) \in I \right\}.$$

*If additionally Assumption 13 holds, this is an isomorphism and preserves the dimension of the faces.*

*Proof.* First, we note that the faces of both $\Delta_\mathcal{A}^\mathcal{S}$ and $\mathcal{N}_\gamma^\mu$ have the structure of the left and right hand side of (54) respectively, which follows from (2). Denote now the left and right hand side in (54) by $F$ and $G$ respectively, then we need to show that $\Psi_\gamma^\mu(F) = G$. For $\pi \in F$ it holds that

$$\eta_\gamma^{\pi,\mu}(s,a) = \rho_\gamma^{\pi,\mu}(s)\pi(a|s) = 0 \quad \text{for all } (s,a) \in I$$

and hence $\eta_\gamma^{\pi,\mu} \in G$. On the other hand for $\eta \in G$ we can set $\pi(\cdot|s) \coloneqq \eta(\cdot|s)$ whenever defined and any other element such that $\pi(a|s) = 0$ for all $(s,a) \in I$ otherwise. Then we surely have $\pi \in F$ and by Proposition 46 also $\eta_\gamma^{\pi,\mu} = \eta$. In the case that $\rho_\gamma^{\pi,\mu} > 0$ holds entrywise for all policies $\pi \in \text{int}(\Delta_\mathcal{A}^\mathcal{S})$, the mapping $\eta \mapsto \eta(\cdot|\cdot)$ defines an inverse to $\Psi_\gamma^\mu$, which shows that the mapping defined in (54) is injective. The assertion on the dimension follows from basic dimension counting, from the fact that the rank is preserved by a lattice isomorphism or by virtue of Proposition 48. □

### C.2 THE PARTIALLY OBSERVABLE CASE

In Corollary 5, we have seen that the discounted state-action frequencies form a semialgebraic set. Now we aim to describe its defining polynomial inequalities. In Section 5 we will discuss how the degree of these polynomials allows us to upper bound the number of critical points of the optimization problem.

**Definition 9** (Effective policy polytope). We call the set of effective policies $\tau = \pi \circ \beta \in \Delta_{\mathcal{A}}^{\mathcal{S}}$ the *effective policy polytope* and denote it by $\Delta_{\mathcal{A}}^{\mathcal{S},\beta}$.

Note that $\Delta_{\mathcal{A}}^{\mathcal{S},\beta}$ is indeed a polytope since it is the image of the polytope $\Delta_{\mathcal{A}}^{\mathcal{O}}$ under the linear mapping $\pi \mapsto \pi \circ \beta$. Hence, we can write it as an intersection

$$\Delta_{\mathcal{A}}^{\mathcal{S},\beta} = \Delta_{\mathcal{A}}^{\mathcal{S}} \cap \mathcal{U} \cap \mathcal{C}, \tag{17}$$

where $\mathcal{U}, \mathcal{C} \subseteq \mathbb{R}^{\mathcal{S} \times \mathcal{A}}$ are an affine subspace and a polyhedral cone and describe a finite set of linear equalities and a finite set of linear inequalities respectively. In the following we will compute those sets explicitly under mild conditions and see that they do not carry an affine part.

### C.2.1 DEFINING LINEAR INEQUALITIES OF THE EFFECTIVE POLICY POLYTOPE

Obtaining inequality descriptions of the images of polytopes under linear maps is a fundamental problem that is non-trivial in general. It can be approached algorithmically, e.g., by Fourier-Motzkin elimination, block elimination, vertex approaches, and equality set projection (Jones et al., 2004). We discuss the special case where the linear map is injective, corresponding to the case where the associated matrix $B$ has linearly independent columns. As a polytope is a finite intersection of closed half spaces $H^+ = \{x \mid n^T x \geq \alpha\}$, it suffices to characterize the image $BH^+$. It holds that

$$BH^+ = \{y \in \operatorname{range} B \mid n^T B^+ y \geq \alpha\} = \{y \mid ((B^+)^T n)^T y \geq \alpha\} \cap \ker(B^T)^\perp, \tag{18}$$

where $B^+$ is a pseudoinverse and where we have used that $B^+ y$ consists of at most one element by the injectivity of $B$. Let us now come back to the mapping $\pi \mapsto \pi \circ \beta = \beta\pi$. By the "vec-trick", this map corresponds to $\operatorname{vec}(\beta\pi I) = (I^T \otimes \beta)\operatorname{vec}(\pi)$. Hence the linear map is represented by the matrix $B = I \otimes \beta$. We observe that $(I \otimes \beta)^+ = I \otimes \beta^+$ (see Langville & Stewart, 2004, Section 2.6.3). Notice that $B = I \otimes \beta$ has linearly independent columns if and only if $\beta$ does. By the above discussion, if $\beta$ has linearly independent columns, then an inequality $\langle \pi, n \rangle \geq 0$ in the policy polytope $\Delta_{\mathcal{A}}^{\mathcal{O}}$ corresponds to an inequality $\langle \tau, (\beta^+)^T n \rangle \geq 0$ in the polytope $\Delta_{\mathcal{A}}^{\mathcal{S}}$.

**Assumption 10.** The matrix $\beta \in \Delta_{\mathcal{O}}^{\mathcal{S}} \subseteq \mathbb{R}^{\mathcal{S} \times \mathcal{O}}$ has linearly independent columns.

**Remark 11.** The assumption above does not imply that the system is fully observable. Recall that if $\beta$ has linearly independent columns, the Moore-Penrose takes the form $\beta^+ = (\beta^T \beta)^{-1}\beta^T$. An interesting special case is when $\beta$ is deterministic but may map several states to the same observation (this is the partially observed setting considered in numerous works). In this case, $\beta^+ = \operatorname{diag}(n_1^{-1}, \ldots, n_{|\mathcal{O}|}^{-1})\beta^T$, where $n_o$ denotes the number of states with observation $o$. In this case, $\beta_{so}^+$ agrees with the conditional distribution $\beta(s|o)$ with respect to a uniform prior over the states; however, this is not in general the case since $\beta^+$ can have negative entries.

**Theorem 12** ($H$-description of the effective policy polytope). *Let $(\mathcal{S}, \mathcal{O}, \mathcal{A}, \alpha, \beta, r)$ be a POMDP and let Assumption 10 hold. Then it holds that*

$$\Delta_{\mathcal{A}}^{\mathcal{S},\beta} = \Delta_{\mathcal{A}}^{\mathcal{S}} \cap \mathcal{U} \cap \mathcal{C} = \mathcal{U} \cap \mathcal{C} \cap \mathcal{D}, \tag{3}$$

*where $\mathcal{U} = \{\pi \circ \beta \mid \pi \in \mathbb{R}^{\mathcal{S} \times \mathcal{O}}\} = \ker(\beta^T)^\perp$ is a subspace, $\mathcal{C} = \{\tau \in \mathbb{R}^{\mathcal{S} \times \mathcal{A}} \mid \beta^+ \tau \geq 0\}$ is a pointed polyhedral cone and $\mathcal{D} = \{\tau \in \mathbb{R}^{\mathcal{S} \times \mathcal{A}} \mid \sum_a (\beta^+ \tau)_{oa} = 1 \text{ for all } o \in \mathcal{O}\}$ an affine subspace. Further, the face lattices of $\Delta_{\mathcal{A}}^{\mathcal{O}}$ and $\Delta_{\mathcal{A}}^{\mathcal{S},\beta}$ are isomorphic.*

*Proof.* First, we recall the defining linear (in)equalities of the policy polytope $\Delta_{\mathcal{A}}^{\mathcal{O}}$, which are given by

$$\pi(a|o) = \langle \delta_o \otimes \delta_a, \pi \rangle_{\mathcal{O} \times \mathcal{A}} \geq 0 \quad \text{for all } a \in \mathcal{A}, o \in \mathcal{O} \quad \text{and}$$

$$\sum_a \pi(a|o) = \langle \delta_o \otimes \mathbb{1}_{\mathcal{A}}, \pi \rangle_{\mathcal{O} \times \mathcal{A}} = 1 \quad \text{for all } o \in \mathcal{O}.$$

Hence, by the general discussion from above, namely by (18), it holds that

$$\Delta_{\mathcal{A}}^{\mathcal{S},\beta} = \ker(\beta^T)^\perp \cap \{\tau \mid \beta^+ \tau \geq 0\} \cap \Big\{\tau \mid \sum_a (\beta^+ \tau)_{oa} = 1 \text{ for all } o \in \mathcal{O}\Big\}.$$

Note that the linear inequalities $\sum_a (\beta^+ \tau)_{oa} = 1$ are redundant in $\Delta_{\mathcal{A}}^{\mathcal{S}}$. To see this, we note that $\beta^+ \mathbb{1}_{\mathcal{S}} = \mathbb{1}_{\mathcal{O}}$ by the injectivity of $\beta$ and $\beta \mathbb{1}_{\mathcal{O}} = \mathbb{1}_{\mathcal{S}}$. Now we can check that

$$\sum_a (\beta^+ \tau)_{oa} = \sum_a \sum_s \beta_{os}^+ \tau_{sa} = \sum_s \beta_{os}^+ \sum_a \tau_{sa} = \sum_s \beta_{os}^+ = 1.$$

This together with $\beta(\Delta_{\mathcal{A}}^{\mathcal{O}}) \subseteq \Delta_{\mathcal{A}}^{\mathcal{S}}$ shows that

$$\Delta_{\mathcal{A}}^{\mathcal{S},\beta} = \Delta_{\mathcal{A}}^{\mathcal{S}} \cap \ker(\beta^T)^\perp \cap \{\tau \mid \beta^+ \tau \geq 0\}.$$

The reformulation of the sets $\mathcal{C}$ and $\mathcal{D}$ for deterministic observation mechanisms $\beta$ follows from the preceding remark. $\qquad\square$

### C.2.2 Defining polynomial inequalities of the feasible state-action frequencies

Using that the inverse of $\pi \mapsto \eta^\pi$ is given through conditioning (see Proposition 46), we can translate linear inequalities in $\Delta_{\mathcal{A}}^{\mathcal{S}}$ into polynomial inequalities in $\mathcal{N}_\gamma^\mu$. More precisely, we have the following result, which can easily be extended to more general inequalities.

**Proposition 14** (Correspondence of inequalities). *Let $(\mathcal{S}, \mathcal{A}, \alpha, r)$ be an MDP, $\tau \in \Delta_{\mathcal{A}}^{\mathcal{S}}$ and let $\eta \in \Delta_{\mathcal{S} \times \mathcal{A}}$ denote its corresponding discounted state-action frequency for some $\mu \in \Delta_{\mathcal{S}}$ and $\gamma \in (0,1]$. Let $c \in \mathbb{R}, b \in \mathbb{R}^{\mathcal{S} \times \mathcal{A}}$ and set $S := \{s \in \mathcal{S} \mid b_{sa} \neq 0 \text{ for some } a \in \mathcal{A}\}$. Then*

$$\sum_{s,a} b_{sa} \tau_{sa} \geq c \quad \text{implies} \quad \sum_{s \in S} \sum_a b_{sa} \eta_{sa} \prod_{s' \in S \setminus \{s\}} \sum_{a'} \eta_{s'a'} - c \prod_{s' \in S} \sum_{a'} \eta_{s'a'} \geq 0,$$

*where the right is a multi-homogeneous polynomial[8] in the blocks $(\eta_{sa})_{a \in \mathcal{A}} \in \mathbb{R}^{\mathcal{A}}$ with multi-degree $\mathbb{1}_S \in \mathbb{N}^{\mathcal{S}}$. If further Assumption 13 holds, the inverse implication also holds.*

*Proof.* Let $\tau \in \Delta_{\mathcal{A}}^{\mathcal{S}}$ and let $\eta$ denote its corresponding discounted stationary distribution and $\rho$ the state marginal. Assuming that the left hand side holds, we compute

$$\sum_{s \in S} \sum_a b_{sa} \eta_{sa} \prod_{s' \in S \setminus \{s\}} \sum_{a'} \eta_{s'a'} = \sum_{s \in S} \sum_a b_{sa} \tau_{sa} \rho_s \prod_{s' \in S \setminus \{s\}} \rho_{s'}$$

$$= \left( \sum_{s,a} b_{sa} \tau_{sa} \right) \cdot \prod_{s' \in S} \rho_{s'} \geq c \prod_{s' \in S} \rho_{s'},$$

which shows the first implication. If further Assumption 13 holds, the product over the marginals is strictly positive, which shows the other implication. $\qquad\square$

**Remark 55.** According to the preceding proposition, a linear inequality in the state policy polytope $\Delta_{\mathcal{A}}^{\mathcal{S}}$ involving actions of $k$ different states yields a polynomial inequality of degree $k$ in the set of state-action frequencies $\mathcal{N}_\gamma^\mu$. In particular, for a linearly constrained policy model $\Pi \subseteq \Delta_{\mathcal{A},\gamma}^{\mathcal{S}}$, where every constraint only addresses a single state, the set of state-action frequencies induced by these policies will still form a polytope. This shows that this type of box constraints are well aligned with the algebraic geometric structure of the problem. The linear constraints arising from partial observability never exhibit this box type structure – unless the system is equivalent to its fully observable version. This is because the projection of the effective policy polytope $\Delta_{\mathcal{A}}^{\mathcal{S},\beta}$ onto a single state always gives the entire probability simplex $\Delta_{\mathcal{A}}$, which is never the case, if there is a non trivial linear constraint concerning only this state.

**Theorem 16.** *Let $(\mathcal{S}, \mathcal{O}, \mathcal{A}, \alpha, \beta, r)$ be a POMDP, $\mu \in \Delta_{\mathcal{S}}$ and $\gamma \in (0,1]$ and assume that Assumption 13 holds. Then we have $\mathcal{N}_\gamma^{\mu,\beta} = \mathcal{N}_\gamma^\mu \cap \mathcal{V} \cap \mathcal{B}$, where $\mathcal{V}$ is a variety described by multi-homogeneous polynomial equations and $\mathcal{B}$ is a basic semialgebraic set described by multi-homogeneous polynomial inequalities. Further, the face lattices of $\Delta_{\mathcal{A}}^{\mathcal{S},\beta}$ and $\mathcal{N}_\gamma^{\mu,\beta}$ are isomorphic.*

---

[8]A polynomial $p \colon \mathbb{R}^{n_1} \times \cdots \times \mathbb{R}^{n_k} \to \mathbb{R}$ is called *multi-homogeneous* with *multi-degree* $(d_1, \ldots, d_k) \in \mathbb{N}^k$, if it is homogeneous of degree $d_j$ in the $j$-th block of variables for $j = 1, \ldots, k$.

| | (In)equalities of state policies | (In)equalities of state-action frequencies |
|---|---|---|
| MDPs | $\Delta_{\mathcal{A}}^{\mathcal{S}}$ is described by | $\mathcal{N}_{\gamma}^{\mu}$ is described by |
| | $\tau(a\|s) \geq 0$ 
 Row normalization: 
 $\sum_a \tau(a\|s) - 1 = 0$ | $\eta(s,a) \geq 0$ 

 – |
| | – | Discounted stationarity: 
 $\langle w_{\gamma}^s, \eta \rangle - (1-\gamma)\mu(s) = 0$ 
 For $\gamma = 1$: 
 $\sum_{s,a} \eta_{sa} - 1 = 0$ |
| | – | |
| POMDPs | $\Delta_{\mathcal{A}}^{\mathcal{S},\beta}$ is described in $\Delta_{\mathcal{A}}^{\mathcal{S}}$ by | $\mathcal{N}_{\gamma}^{\mu,\beta}$ is described in $\mathcal{N}_{\gamma}^{\mu}$ by |
| | Linear (in)equalities 
 See Section 4 

 Closed form under Assumption 10: 
 See Theorem 12 

 Closed form for deterministic observ.: 
 See Remark 57 | Polynomial (in)equalities 
 See Section 4, Proposition 14 

 Closed form under Assumption 10: 
 See Remark 18 for inequalities 
 See Remark 56 for equalities 

 Closed form for deterministic observ.: 
 See Remark 57 |

Table 1: Correspondence of the defining linear and polynomial inequalities of the (effective) state policies and the (feasible) state-action frequencies for MDPs and POMDPs respectively.

*Proof.* The equation $\mathcal{N}_{\gamma}^{\mu,\beta} = \mathcal{N}_{\gamma}^{\mu} \cap \mathcal{V} \cap \mathcal{B}$ is a direct consequence of (3) and Proposition 14. Further, it is clear from Proposition 14 that the mapping $\Psi \colon \Delta_{\mathcal{A}}^{\mathcal{S}} \to \mathcal{N}_{\gamma}^{\mu}, \pi \mapsto \eta_{\gamma}^{\pi,\mu}$ induces a bijection of the face lattices of $\Delta_{\mathcal{A}}^{\mathcal{S},\beta}$ and $\mathcal{N}_{\gamma}^{\mu,\beta}$. In order to see that the join and meet are respected, we note that for $F, G \in \mathcal{F}(\Delta_{\mathcal{A}}^{\mathcal{S},\beta})$ it holds that $\Psi(F \wedge G) = \Psi(F \cap G) = \Psi(F) \cap \Psi(G) = \Psi(F) \wedge \Psi(G)$. Further, $\Psi(F \vee G)$ is a face of $\mathcal{N}_{\gamma}^{\mu,\beta}$ containing $\Psi(F)$ and $\Psi(G)$ and hence by definition $\Psi(F) \vee \Psi(G) \subseteq \Psi(F \vee G)$. Further, for any face $I$ of $\mathcal{N}_{\gamma}^{\mu,\beta}$ containing $\Psi(F)$ and $\Psi(G)$ it holds that $\Psi^{-1}(I)$ is a face of $\Delta_{\mathcal{A}}^{\mathcal{S},\beta}$ containing $F$ and $G$ and hence $\Psi^{-1}(I) \supseteq F \vee G$ or equivalently $\Psi(F \vee G) \subseteq I$. □

Comparing (17) and Theorem 16 we see that the linear space $\mathcal{U}$ corresponds to the variety $\mathcal{V}$, where the cone $\mathcal{C}$ corresponds to the basic semialgebraic set $\mathcal{B}$. In general, every linear (in)equality cutting out the effective policy polytope $\Delta_{\mathcal{A}}^{\mathcal{S},\beta}$ from the state policy polytope $\Delta_{\mathcal{A}}^{\mathcal{S}}$ of the associated MDP corresponds to a polyomial (in)equality cutting out the feasible state-action frequencies $\mathcal{N}_{\gamma}^{\mu,\beta}$ from all state-action frequencies $\mathcal{N}_{\gamma}^{\mu}$ of the corresponding MDP, see also Table 1. This correspondence arises by relating state-action frequencies to state policies via conditioning. Hence, the problem of computing the defining polynomial inequalities of the feasible state-action frequencies reduces to computing the defining linear inequalities of the effective policy polytope. This can be done in closed form if $\beta$ has linearly independent columns or if it deterministic, see Remark 18, 56 and 57.

**Remark 18.** By Theorem 12 and Proposition 14, the defining polynomials of the basic semialgebraic set $\mathcal{B}$ from Theorem 16 are indexed by $a \in \mathcal{A}, o \in \mathcal{O}$ and are given by

$$p_{ao}(\eta) := \sum_{s \in S_o} \left( \beta_{os}^+ \eta_{sa} \prod_{s' \in S_o \setminus \{s\}} \sum_{a'} \eta_{s'a'} \right) = \sum_{f \colon S_o \to \mathcal{A}} \left( \sum_{s' \in f^{-1}(\{a\})} \beta_{os'}^+ \right) \prod_{s \in S_o} \eta_{sf(s)} \geq 0, \quad (4)$$

where $S_o := \{ s \in \mathcal{S} \mid \beta_{os}^+ \neq 0 \}$. The polynomials depend only on $\beta$ and not on $\gamma$, $\mu$ nor $\alpha$, and have $|S_o||\mathcal{A}|^{|S_o|-1}$ monomials of degree $|S_o|$ of the form $\prod_{s \in S_o} \eta_{sf(s)}$ for some $f \colon S_o \to \mathcal{A}$. In particular, we can read of the multi-degree of $p_{ao}$ with respect to the blocks $(\eta_{sa})_{a \in \mathcal{A}}$ which is given by $\mathbb{1}_{S_o}$ (see also Proposition 14). A complete description of the set $\mathcal{N}_{\gamma}^{\mu,\beta}$ via (in)equalities follows from the description of $\mathcal{N}_{\gamma}^{\mu}$ via linear (in)equalities given in (2). In Section 5 we discuss how the degree of these polynomials controls the complexity of the optimization problem.

**Remark 56** (Defining polynomial equalities). Analogously to the defining inequalities, we can compute the defining polynomial equalities in the following way. First, we need to compute a basis $\{b^j\}_{j \in J}$ of $\{\beta\pi \mid \pi \in \mathbb{R}^{\mathcal{O} \times \mathcal{A}}\}^{\perp} = \ker(\beta^T) \subseteq \mathbb{R}^{\mathcal{S} \times \mathcal{A}}$, which can easily be done using the Gram-Schmidt process. Note that the defining linear equalities of the effective policy polytope (in the policy polytope) are given by $\langle b^j, \tau \rangle_{\mathcal{S} \times \mathcal{A}} = 0$. Hence, by Proposition 14 the corresponding polynomial equality is given by

$$q_j(\eta) := \sum_{s \in S_j} \sum_{a \in \mathcal{A}} b^j_{sa} \eta_{sa} \prod_{s' \in S_j \setminus \{s\}} \sum_{a' \in \mathcal{A}} \eta_{s'a'} = 0, \tag{19}$$

where $S_j := \{s \in \mathcal{S} \mid b^j_{sa} \neq 0 \text{ for some } a \in \mathcal{A}\}$.

**Remark 57** (Polynomial constraints for deterministic observations). In the case, where $\beta$ corresponds to a determinstic mapping we can compute all polynomial constraints in closed form. Let us assume that $\beta(o|s) = \delta_{og(s)}$ for some mapping $g \colon \mathcal{S} \to \mathcal{O}$ and write $S_o := g^{-1}(\{o\}) \subseteq \mathcal{S}$, then $\tau \in \Delta^{\mathcal{S}}_{\mathcal{A}}$ belongs to the effective policy polytope $\Delta^{\mathcal{S},\beta}_{\mathcal{A}}$ if and only if

$$\tau(a|s_1) = \tau(a|s_2) \quad \text{for all } s_1, s_2 \in S_o, a \in \mathcal{A}, o \in \mathcal{O}. \tag{20}$$

Note that this can be encoded in $\sum_o |\mathcal{A}|(|S_o| - 1) = |\mathcal{A}|(|\mathcal{S}| - |\mathcal{O}|)$ linear equations; indeed if we fix $s_o \in S_o$, then (20) is equivalent to

$$\tau(a|s) - \tau(a|s_o) = 0 \quad \text{for all } s \in S_o \setminus \{s_o\}, a \in \mathcal{A}, o \in \mathcal{O}. \tag{21}$$

Another way to derive these linear equalities is by noticing that $e_{sa} - e_{s_o a}$ form a basis of $\ker(\beta^T)$, compare also Remark 56. By Proposition 14 for $\eta \in \mathcal{N}^{\mu}_{\gamma}$ it is equivalent to lie in $\mathcal{N}^{\mu,\beta}_{\gamma}$ or to satisfy

$$\eta_{sa} \sum_{a'} \eta_{s_o a'} - \eta_{s_o a} \sum_{a'} \eta_{sa'} = 0 \quad \text{for all } s \in S_o \setminus \{s_o\}, a \in \mathcal{A}, o \in \mathcal{O}. \tag{22}$$

Note that in this case, there are no polynomial inequalities; this can also be seen from Remark 11 and Remark 18. Indeed, it holds that $\beta^+ = \beta^T \operatorname{diag}(n_1, \ldots, n_{|\mathcal{O}|}) \geq 0$, where $n_o := |S_o|$. Hence, the polynomial inequalities $p_{ao}(\eta) \geq 0$ are redundant on the cone $[0, \infty)^{\mathcal{S} \times \mathcal{A}}$.

**Remark 58.** In the fully observable case we have $|S_o| = 1$ for each $o$. Hence, each of the polynomial inequalities has a single term of degree 1. Indeed, in this case the inequalities are simply $\eta_{sa} \geq 0$, for each $a$, for each $s$. In the case of a deterministic $\beta$, we have $\beta^+_{os} = \mathbb{1}_{s \in S_o}/|S_o|$. For each $o, a$, there is an inequality $\sum_{f \colon S_o \to \mathcal{A}} |f^{-1}(a)| \prod_{s \in S_o} \eta_{sf(s)} \geq 0$ of degree $|S_o|$ equal to the number of states that are compatible with $o$.

**Remark 59** (Reformulation of reward maximization as a polynomial program). By the theorem above and Proposition 14, reward maximization is equivalent to the maximization of a linear function subject to polynomial constraints. This enables the use of any (approximate) solution technique of polynomial optimization problems in order to solve POMDPs. Such methods have been developed for a long time and have been applied to a variety of problems (Anjos & Lasserre, 2011; Lasserre, 2015). As meta algorithm, this is presented in Algorithm 1. Once, a solution $\eta^*$ is obtained, the corresponding state policy $\tau^* \in \Delta^{\mathcal{S}}_{\mathcal{A}}$ can be computed by conditioning, i.e. $\tau(a|s) := \eta_{sa}/(\sum_{a'} \eta_{sa'})$. Then, every $\pi^* \in \Delta^{\mathcal{O}}_{\mathcal{A}}$ with $\beta\pi^* = \tau^*$ is an optimal policy. Such a policy can be computed by solving a system of linear equations, which are $\beta\pi = \tau$ and $\pi \in \Delta^{\mathcal{O}}_{\mathcal{A}}$, which is standard. In particular, if $\beta$ has linearly independent columns, it holds that $\pi^* := \beta^+ \tau$. We demonstrate that this offers a computationally feasible approach to planning of POMPDs in Section F on the toy example used for Figure 1 and a grid world.

## D  DETAILS ON THE OPTIMIZATION

Let us quickly recall how we can reformulate the reward maximization problem as a polynomial optimization problem, which then leads us to the mighty tools of algebraic degrees. We perceive the reward maximization problem again as the maximization of a linear function $p_0$ over the set of feasible state-action frequencies $\mathcal{N}^{\mu,\beta}_{\gamma}$. Since under Assumption 13 the parametrization $\pi \mapsto \eta^{\pi}$ is injective and has a full-rank Jacobian everywhere (see Appendix C.1.1), the critical points in the policy polytope $\Delta^{\mathcal{O}}_{\mathcal{A}}$ correspond to the critical points of $p_0$ on $\mathcal{N}^{\mu,\beta}_{\gamma}$ (Trager et al., 2019). In general,

---

**Algorithm 1** Polynomial programming for POMDPs

---

**Require:** $\alpha, \beta, \gamma, \mu$
   **for** $s \in \mathcal{S}$ **do**
      $w^s \leftarrow \delta_s \otimes \mathbb{1}_{\mathcal{A}} - \gamma\alpha(s|\cdot,\cdot)$
   **end for**
   **for** $a \in \mathcal{A}, o \in \mathcal{O}$ **do**
      Define $p_{ao}$ according to Equation (4)
   **end for**
   Compute a basis $\{b^j\}_{j \in J}$ of $\{\beta\pi \mid \pi \in \mathbb{R}^{\mathcal{O} \times \mathcal{A}}\}^{\perp} \subseteq \mathbb{R}^{\mathcal{S} \times \mathcal{A}}$
   **for** $j \in J$ **do**
      Define $q_j$ according to Equation (19)
   **end for**
   $\eta^* \leftarrow \arg\max\langle r, \eta\rangle$ sbj to $\eta \geq 0, \langle w^s, \eta\rangle = (1-\gamma)\mu_s, \langle\mathbb{1}_{\mathcal{S} \times \mathcal{A}}, \eta\rangle = 1, p_{ao}(\eta) \geq 0, q_j(\eta) = 0$
   $R^* \leftarrow \langle r, \eta^*\rangle$
   $\tau^* \leftarrow \eta^*(\cdot|\cdot) \in \Delta_{\mathcal{A}}^{\mathcal{S}}$
   $\pi^* \leftarrow$ solution of $\beta\pi = \tau^*$
      **return** maximizer $\eta^*$, optimal value $R^*$, optimal policy $\pi^*$

---

critical points of this linear function can occur on every face of the semialgebraic set $\mathcal{N}_{\gamma}^{\mu,\beta}$. The optimization problem thus has a combinatorial and a geometric component, corresponding to the number of faces of each dimension and the number of critical points in the interior of any given face. We have discussed the combinatorial part in Theorem 16 and focus now on the geometric part. Writing $\mathcal{N}_{\gamma}^{\mu,\beta} = \{\eta \in \mathbb{R}^{\mathcal{S} \times \mathcal{A}} : p_i(\eta) \leq 0, i \in I\}$, we are interested in the number of critical points on the interior of a face

$$\text{int}(F_J) = \{\eta \in \mathcal{N}_{\gamma}^{\mu,\beta} \mid p_j(\eta) = 0 \text{ for } j \in J, p_i(\eta) > 0 \text{ for } i \in I \setminus J\}.$$

Note that a point $\eta \in \text{int}(F_J)$ is critical, if and only if it is a critical point on the variety

$$\mathcal{V}_J := \{\eta \in \mathbb{R}^{\mathcal{S} \times \mathcal{A}} \mid p_j(\eta) = 0 \text{ for } j \in J\}.$$

For the sake of notation, let us assume that $J = \{1, \ldots, m\}$ from now on. We can upper bound the number of critical points in the interior of the face by the number of critical points of the polynomial optimization problem

$$\text{maximize } p_0(\eta) \quad \text{subject to} \quad p_j(\eta) = 0 \text{ for } j = 1, \ldots, m, \tag{23}$$

where the polynomials have $n$ variables. The number of critical points of this problems is upper bounded by the algebraic degree of the problem as we discuss now.

### D.1 INTRODUCTION TO ALGEBRAIC DEGREES

We try to present the results from the mighty theory of algebraic degrees that we use here and refer the interested reader to the excellent low level introduction by Breiding et al. (2021) and to the references therein. Let us consider the polynomial optimization problem (23), where we do not require $p_0$ to be linear. Further, denote the number of variables by $n$ (in the case of state-action frequencies $n = |\mathcal{S}||\mathcal{A}|$) and denote the degrees of $p_0, \ldots, p_m$ by $d_0, \ldots, d_m$. We call a point *critical*, if it satisfies the KKT conditions ($\nabla p_0(x) + \sum_{i=1}^{m} \lambda_i \nabla p_i(x) = 0, p_1(x) = \cdots = p_m(x) = 0$), which can be phrased as a system of polynomial equations (see Nie & Ranestad, 2009). The number of complex solutions to those criticality equations, when finite, is called the *algebraic degree* of the problem. The algebraic degree is determined by the nature of the polynomials $p_0, \ldots, p_m$ and captures the computational complexity of the optimization problem (Kung, 1973; Bajaj, 1988).[9] A special case of (23) is when $m = n$ and the polynomials $p_1, \ldots, p_m$ are generic. Then by Bézout's theorem there are exactly $d_1 \cdots d_n$ isolated points satisfying the polynomial constraints and all of them are critical and hence the algebraic degree is precisely $d_1 \cdots d_n$ (Timme, 2021). If the polynomials $p_0, \ldots, p_m$ define a complete intersection, i.e., the co-dimension of their induced

---

[9]The coordinates of critical points can be shown to be roots of some univariate polynomials whose degree equals the algebraic degree and whose coefficients are rational functions of the coefficients of $p_0, \ldots, p_m$.

variety is $m + 1$, the algebraic degree of (23) is upper bounded by

$$d_1 \cdots d_m \sum_{i_0 + \cdots + i_m = n - m} (d_0 - 1)^{i_0} \cdots (d_m - 1)^{i_m}, \tag{24}$$

and this bound is attained for generic polynomials (Nie & Ranestad, 2009; Breiding et al., 2021). For non-complete intersections, the expression (24) does not need to yield an upper bound if some constraints are redundant. However, we can modify the expression to obtain a valid upper bound. Indeed, if $l$ and $c = n - l$ denote the dimension and co-dimension of

$$\mathcal{V} := \{x \mid p_1(x) = \cdots = p_m(x) = 0\}$$

and if $p_0$ is generic and if the degrees are ordered, i.e., $d_1 \geq \cdots \geq d_m$, then the algebraic degree is upper bounded by

$$d_1 \cdots d_c \sum_{i_0 + \cdots + i_c = l} (d_0 - 1)^{i_0} \cdots (d_c - 1)^{i_c}. \tag{25}$$

To see this, fix a subset $J \subseteq \{1, \ldots, m\}$ of cardinality $c$, such that $\mathcal{V} = \{x \mid p_j(x) = 0 \text{ for } j \in J\}$. Then we can apply the bound from (24) and evaluate it to be

$$\prod_{j \in J} d_j \sum_{i_0 + \sum_{j \in J} i_j = n - c} (d_0 - 1)^{i_0} \cdot \prod_{j \in J} (d_j - 1)^{i_j},$$

which is clearly upper bounded by (25). If $p_0$ is linear, then $d_0 = 1$ and the expression simplifies to

$$d_1 \cdots d_c \sum_{i_1 + \cdots + i_c = l} (d_1 - 1)^{i_0} \cdots (d_c - 1)^{i_c}.$$

If further $d_i = 1$ for $i \geq k$ for some $k \leq c$, then we obtain

$$d_1 \cdots d_k \sum_{i_1 + \cdots + i_k = l} (d_1 - 1)^{i_1} \cdots (d_k - 1)^{i_k}. \tag{26}$$

If $p_{k+1}, \ldots, p_m$ are affine linear (and in general position relative to $p_1, \ldots, p_k$, the algebraic degree of (23) is given by the $(m - k)$-th *polar degree* $\delta_{m-k}(\mathcal{V})$ of the variety

$$\mathcal{V} := \{\eta \mid p_{k+1}(\eta) = \cdots = p_m(\eta) = 0\},$$

see Draisma et al. (2016); Özlüm Çelik et al. (2021). This relation is particularly useful, since for state-action frequencies there are always active linear equations as described in (2). The polar degrees of certain interesting cases (Segre-Veronese varieties) have been recently computed by Sodomaco (2020, Section 5) and our proof of Proposition 21 builds on those formulas and their presentation by Özlüm Çelik et al. (2021).

**Remark 60** (Genericity assumptions). In the case, where the polynomials $p_0, \ldots, p_m$ are not generic, there might be infinitely many critical points. Indeed, even for a linear program, i.e., when all polynomials are linear, there might be infinitely many and even a non-trivial face of global optima. This is however not the case if $p_0$ is generic. Hence, the genericity assumptions on the reward vector $r$ and also other elements of the POMDP are not surprising. For example, they prevent the reward vector to be identical to zero or to be perpendicular on all vectors $\delta_s \otimes \mathbb{1}_{\mathcal{A}} - \gamma \alpha(s|\cdot, \cdot)$ in which cases the reward function would be constant and every policy would be a global optimum.

## D.2 GENERAL UPPER BOUND ON THE NUMBER OF CRITICAL POINTS

**Theorem 20.** *Consider a POMDP $(\mathcal{S}, \mathcal{O}, \mathcal{A}, \alpha, \beta, r)$, $\gamma \in (0, 1)$, assume that $r$ is generic, that $\beta \in \mathbb{R}^{\mathcal{S} \times \mathcal{O}}$ is invertible, and that Assumption 13 holds. For any given $I \subseteq \mathcal{A} \times \mathcal{O}$ consider the following set of policies, which is the relative interior of a face of the policy polytope:*

$$\mathrm{int}(F) = \left\{ \pi \in \Delta_{\mathcal{A}}^{\mathcal{O}} \mid \pi(a|o) = 0 \text{ if and only if } (a, o) \in I \right\}.$$

*Let $O := \{o \in \mathcal{O} \mid (a, o) \in I \text{ for some } a\}$ and set $k_o := |\{a \mid (a, o) \in I\}|$ as well as $d_o := |\{s \mid \beta_{os}^{-1} \neq 0\}|$. Then, the number of critical points of the reward function on $\mathrm{int}(F)$ is at most*

$$\left( \prod_{o \in O} d_o^{k_o} \right) \cdot \sum_{\sum_{o \in O} i_o = l} \prod_{o \in O} (d_o - 1)^{i_o}, \tag{5}$$

*where $l = |\mathcal{S}|(|\mathcal{A}| - 1) - |I|$. If $\alpha$ and $\mu$ are generic, this bound can be refined by computing the polar degrees of multi-homogeneous varieties (see Proposition 21 for a special case). The same bound holds in the mean reward case $\gamma = 1$ for $l$ given in Remark 61.*

*Proof.* The face $G$ of the effective policy polytope corresponding to $F$ is given by

$$\text{int}(G) = \left\{ \tau \in \Delta_{\mathcal{A}}^{\mathcal{S},\beta} \mid (\beta^+ \tau)_{oa} = 0 \Leftrightarrow (a, o) \in I \right\}.$$

In order to describe the corresponding set of discounted state-action frequencies, we use the notation

$$p_{ao}(\eta) := \sum_{s \in S_o} \left( \beta_{os}^+ \eta_{sa} \prod_{s' \in S_o \setminus \{s\}} \sum_{a'} \eta_{s'a'} \right),$$

then it holds that

$$\mathcal{N}_\gamma^{\mu,\beta} = \{ \eta \in \mathcal{N}_\gamma^\mu \mid p_{ao}(\eta) \geq 0 \text{ for all } a \in \mathcal{A}, o \in \mathcal{O} \}.$$

Then, $F$ and $G$ correspond to the face

$$\begin{aligned}
\text{int}(H) &= \left\{ \eta \in \mathcal{N}_\gamma^{\mu,\beta} \mid p_{ao}(\eta) = 0 \Leftrightarrow (a, o) \in I \right\} \\
&= \left\{ \eta \in \mathcal{N}_\gamma^\mu \mid p_{ao}(\eta) \geq 0 \text{ and equality if and only if } (a, o) \in I \right\}.
\end{aligned}$$

In order to use the explicit description of $\mathcal{N}_\gamma^\mu$ given in (2), we remind the reader that $w_\gamma^s := \delta_s \otimes \mathbb{1}_{\mathcal{A}} - \gamma \alpha(s|\cdot, \cdot)$. Then, it holds that

$$\begin{aligned}
\text{int}(H) = \{ \eta \in [0, \infty)^{\mathcal{S} \times \mathcal{A}} \mid{} & p_{ao}(\eta) \geq 0 \text{ and equality if and only if } (a, o) \in I \\
& \langle w_\gamma^s, \eta \rangle_{\mathcal{S} \times \mathcal{A}} = (1 - \gamma)\mu_s \text{ for } s \in \mathcal{S} \},
\end{aligned}$$

where we used Proposition 8. Since the discounted state distributions are all positive by assumption, for $\eta \in \text{int}(H)$ it holds $\eta_{sa} = 0$ if and only if $\tau(a|s) := \eta(a|s) = 0$. Note that $\tau = \pi \circ \beta$ for some $\pi \in \Delta_{\mathcal{A}}^{\mathcal{O}}$ by assumption and thus for $\eta \in \text{int}(H)$ it holds that $\eta_{sa} = 0$ if and only if

$$0 = \tau(a|s) = \sum_o \beta(o|s)\pi(a|o),$$

which holds if and only if $(a, o) \in I$ for every $o \in \mathcal{O}$ with $\beta(o|s) > 0$. Hence, if we write $J := \{(s, a) \mid (a, o) \in I \text{ for all } o \in \mathcal{O} \text{ with } \beta(o|s) > 0\}$, we obtain

$$\begin{aligned}
\text{int}(H) = \{ \eta \in \mathbb{R}^{\mathcal{S} \times \mathcal{A}} \mid{} & \eta_{sa} \geq 0 \text{ and equality if and only if } (s, a) \in J, \\
& \langle w_\gamma^s, \eta \rangle_{\mathcal{S} \times \mathcal{A}} = (1 - \gamma)\mu_s \text{ for } s \in \mathcal{S}, \\
& p_{ao}(\eta) \geq 0 \text{ and equality if and only if } (a, o) \in I \}.
\end{aligned}$$

The number of critical points over this surface is upper bounded by the number of critical points over

$$\begin{aligned}
\mathcal{V} = \{ \eta \in \mathbb{R}^{\mathcal{S} \times \mathcal{A}} \mid{} & \eta_{sa} = 0 \text{ for } (s, a) \in J, \\
& \langle w_\gamma^s, \eta \rangle_{\mathcal{S} \times \mathcal{A}} = (1 - \gamma)\mu_s \text{ for } s \in \mathcal{S}, p_{ao}(\eta) = 0 \text{ for } (a, o) \in I \}.
\end{aligned}$$

Now we want to apply (26) and note that the objective $p_0 = r$ is generic. Further, we see that there are $|I|$ non-linear constraints and hence in the notation of (26) have $k = |I|$. Further, we can calculate to dimension and co-dimension of $\mathcal{V}$ as follows. Note that $F \to \mathcal{V}, \pi \mapsto \eta^\pi$ is a local parametrization of $\mathcal{V}$ (meaning it parametrizes a full dimensional subset of $\mathcal{V}$), which is injective and has full rank Jacobian everywhere. Hence, we have

$$l = \dim(\mathcal{V}) = \dim(F) = |\mathcal{S}|(|\mathcal{A}| - 1) - |I| = |\mathcal{S}||\mathcal{A}| - |\mathcal{S}| - |I|.$$

The co-dimension of $\mathcal{V}$ is given by $|\mathcal{S}||\mathcal{A}| - \dim(\mathcal{V}) = |\mathcal{S}| + |I|$ and with the notation from (26), we have $c = |\mathcal{S}| + |I| \geq k$. Further, it holds that $\deg(p_{ao}) \leq d_o$ and using (26) yields an upper bound of

$$\prod_{(s,o) \in I} d_o \cdot \sum_{\sum_{(a,o) \in I} j_{ao} = m} \prod_{(a,o) \in I} (d_o - 1)^{j_{ao}} = \prod_{o \in O} d_o^{k_o} \cdot \sum_{\sum_{o \in O} i_o = m} \prod_{o \in O} (d_o - 1)^{i_o}.$$

$\square$

**Remark 61** (The mean reward case). Theorem 20 can be generalized to the mean reward case, i.e., to the case of $\gamma = 1$ with some adjustments. Indeed, the proof can be carried out analogously, however, the characterization of $\mathcal{N}_1^\mu$ has the extra linear condition that $\sum_{sa} \eta_{sa} = 1$, see also Proposition 8. Indeed, in the mean reward case we have with the notation from the proof above

$$\mathrm{int}(H) = \Big\{ \eta \in \mathbb{R}^{\mathcal{S} \times \mathcal{A}} \mid \eta_{sa} \geq 0 \text{ and equality if and only if } (s,a) \in J,$$
$$\langle w_\gamma^s, \eta \rangle_{\mathcal{S} \times \mathcal{A}} = 0 \text{ for } s \in \mathcal{S}, \sum_{sa} \eta_{sa} = 1,$$
$$p_{ao}(\eta) \geq 0 \text{ and equality if and only if } (a,o) \in I \Big\}.$$

Hence, the upper bound in (5) remains valid if we set

$$l := \dim \Big\{ \eta \in \mathbb{R}^{\mathcal{S} \times \mathcal{A}} \mid \eta_{sa} = 0 \text{ for } (s,a) \in J, \langle w_\gamma^s, \eta \rangle_{\mathcal{S} \times \mathcal{A}} = 0 \text{ for } s \in \mathcal{S},$$
$$\sum_{sa} \eta_{sa} = 1, p_{ao}(\eta) = 0 \text{ for } (a,o) \in I \Big\}. \tag{27}$$

In the discounted case we obtained an explicit formulation for $l$. In the mean case the value obeys a case distinction depending, in particular, on whether the all ones vector $\mathbb{1}_\mathcal{S}$ lies in the span of the vectors $w_\gamma^s$. However, the value can be computed from the above expression (27) in any given specific case.

**Corollary 62** (Critical points of MDPs). *Consider an MDP $(\mathcal{S}, \mathcal{A}, \alpha, r)$, $\gamma \in (0,1)$, assume that $r$ is generic, that $\beta \in \mathbb{R}^{\mathcal{S} \times \mathcal{O}}$ is invertible, and that Assumption 13 holds. Then, every critical point $\pi \in \Delta_\mathcal{A}^\mathcal{O}$ of the discounted expected reward function is deterministic.*

*Proof.* We evaluate the bound of Equation (5). If the face is not a vertex, then the corresponding index set $I \subseteq \mathcal{A} \times \mathcal{O}$ satisfies $|I| < |\mathcal{O}|(|\mathcal{A}| - 1)$ and thus in the notation from Theorem 20 it holds that $l > 0$. Note that $d_o = 1$ for every $o \in \mathcal{O}$ and hence there is at least one factor in the product in (5) that vanishes and so does the whole expression in (5). $\qquad\square$

**Remark 63** (Geometry around the critical points). The key argument in the proof of Theorem 20 is that a critical point $\pi \in \Delta_\mathcal{A}^\mathcal{O}$ of the reward function corresponds to a critical point $\eta$ of a linear function over a multi-homogeneous variety $\mathcal{V}$, where the defining polynomials can be computed by the means of Proposition 8 and Proposition 14. A closer study of this variety would shed light into the geometry of the loss landscape around the critical points, which has important implications for gradient based methods.

**Remark 64** (Efficient design of observation mechanisms). The bound (5) could be used to design observation mechanisms in such a way that the reward function has the least critical points, which would potentially make the system more approachable for gradient based methods. Rauh et al. (2021) showed that planning in POMDPs is stable under perturbations of the observation kernel $\beta$. More precisely, consider two observation kernels $\beta, \beta' \in \Delta_\mathcal{O}^\mathcal{S}$ satisfying $\|\beta(\cdot|s) - \beta'(\cdot|s)\|_{TV} = \sum_o |\beta(o|s) - \beta'(o|s)|/2 \leq \varepsilon$ for every $s \in \mathcal{S}$. Then if $\pi \in \Delta_\mathcal{A}^\mathcal{O}$ is an optimal policy of $(\mathcal{S}, \mathcal{A}, \mathcal{O}, \alpha, \beta', r)$, then it is a $2\varepsilon\gamma\|r\|_\infty/(1-\gamma)$-optimal policy of $(\mathcal{S}, \mathcal{A}, \mathcal{O}, \alpha, \beta, r)$. Hence, if $\beta$ does not fulfill the invertability assumption made in Theorem 20 an arbitrary small perturbation of it does (given that $\beta$ is a square matrix) and hence Theorem 20 provides an upper bound on the number of critical points of an approximate problem. Further, note that the faces, which are guaranteed to contain an optimal policy by Montúfar & Rauh (2017) might be considerably fewer for the POMDP $(\mathcal{S}, \mathcal{A}, \mathcal{O}, \alpha, \beta', r)$. The bound (5) could be used to identify the best perturbations of a given magnitude to obtain a problem with a minimal number of critical points.

**Remark 65** (Design of policy models). Knowledge about the location of critical points of the reward function can be used to design policy models, which provably include those critical points and therefore also the optimal policy.

### D.3 NUMBER OF CRITICAL POINTS IN A TWO-ACTION BLIND CONTROLLER

This subsection is devoted to the proof of Proposition 21 that we restate here for convenience.

**Proposition 21** (Number of critical points in a blind controller). *Let $(\mathcal{S}, \mathcal{O}, \mathcal{A}, \alpha, \beta, r)$ be a POMDP describing a blind controller with two actions, i.e., $\mathcal{O} = \{o\}$ and $\mathcal{A} = \{a_1, a_2\}$ and let $r, \alpha$ and $\mu$ be generic and let $\gamma \in (0, 1)$. Then the reward function $R_\gamma^\mu$ has at most $|\mathcal{S}|$ critical points in the interior $\mathrm{int}(\Delta_{\mathcal{A}}^{\mathcal{O}}) \cong (0, 1)$ of the policy polytope and hence at most $|\mathcal{S}| + 2$ critical points.*

Before we present the proof of this result, we discuss how the bound on the rational degree of the reward function leads to am upper bound on the number of critical points. We consider a blind controller and restrict ourselves to the discounted case $\gamma \in (0, 1)$. We associate the policy polytope $\Delta_{\mathcal{A}}^{\mathcal{O}}$ with $[0, 1]$ and for $p \in [0, 1]$ we write $\pi_p$ and $\eta^p$ for the associated policy and the state-action frequency. From Theorem 4 we know that the reward function $R = f/g \colon [0, 1] \to \mathbb{R}$ is a rational function of degree at most $k := |\mathcal{S}|$, which is well known to possess at most $2k - 2$ critical points. Hence, there are at most this many critical points in the interior $(0, 1)$ if the reward function is not constant. Now we use the geometric description of the set of state-action frequencies and yields a refined bound.

*Proof of Proposition 21.* First, we note that since $\mu$ is generic and $\gamma < 1$ Assumption 13 is satisfied. In this case, the combinatorial part is simple, since there are only two zero-dimensional faces of the state-action frequencies (corresponding to the endpoints of the unit interval) and one one-dimensional face (corresponding to the interior of the unit interval). Let us set

$$\mathcal{U} = \{\eta \in \mathbb{R}^{\mathcal{S} \times \mathcal{A}} \mid \langle w_\gamma^s, \eta \rangle_{\mathcal{S} \times \mathcal{A}} = (1 - \gamma)\mu_s \text{ for all } s \in \mathcal{S}\},$$

where $w_\gamma^s := \delta_s \otimes \mathbb{1}_{\mathcal{A}} - \gamma\alpha(s|\cdot, \cdot)$. By Proposition 8 and Example 15 the set of discounted state-action frequencies is given by

$$\mathcal{N}_\gamma^{\mu,\beta} = \mathcal{N}_\gamma^\mu \cap \mathcal{D}_1 = [0, \infty)^{\mathcal{S} \times \mathcal{A}} \cap \mathcal{U} \cap \mathcal{D}_1.$$

Like above, we associate the policy polytope $\Delta_{\mathcal{A}}^{\mathcal{O}}$ with $[0, 1]$ and for $p \in [0, 1]$ we write $\pi_p$ and $\eta^p$ for the associated policy and the state-action frequency. We aim to bound the number of critical points of the reward function over $(0, 1)$ or equivalently the number of critical over $\{\eta^p \mid p \in (0, 1)\}$ where we used that Assumption 13 holds. Further, recall that $\eta^p(a|s) = \eta_{sa}^p/\rho_s^p$, we have that

$$\begin{aligned}
\{\eta^p \mid p \in (0, 1)\} &= \{\eta \in \mathcal{N}_\gamma^{\mu,\beta} \mid \eta(a|s) > 0 \text{ for all } s \in \mathcal{S}, a \in \mathcal{A}\} \\
&= \{\eta \in \mathcal{N}_\gamma^{\mu,\beta} \mid \eta_{sa} > 0 \text{ for all } s \in \mathcal{S}, a \in \mathcal{A}\} \\
&= (0, \infty)^{\mathcal{S} \times \mathcal{A}} \cap \mathcal{U} \cap \mathcal{D}_1.
\end{aligned}$$

Thus the number of critical points over $\{\eta^p \mid p \in (0, 1)\}$ are upper bounded by the number of critical points on $\mathcal{U} \cap \mathcal{D}_1$. Note that if $\alpha$ and $\mu$ are generic, the subspace $\mathcal{U}$ is in general position. Further, its dimension is $|\mathcal{S}||\mathcal{A}| - |\mathcal{S}| = |\mathcal{S}|$, where we used $|\mathcal{A}| = 2$. Hence, the number of complex solutions to the KKT conditions over $\mathcal{U} \cap \mathcal{D}_1$ are given by the $k$-th polar degree $\delta_k(\mathcal{D}_1)$, where $k := |\mathcal{S}|$, where we also used the genericity of the reward vector. We can compute the polar degree using the formula presented by Özlüm Çelik et al. (2021) to obtain

$$\begin{aligned}
\delta_k(\mathcal{D}_1) &= \sum_{s=0}^{k-2k+k+2} (-1)^s \binom{k-s+1}{2k-(k+1)} (k-s)! \left( \sum_{i+j=s} \frac{\binom{k}{i}}{(k-1-i)!} \cdot \frac{\binom{2}{j}}{(2-1-j)!} \right) \\
&= \sum_{s=0}^{2} (-1)^s \binom{k-s+1}{k-1} (k-s)! \left( \sum_{i+j=s} \frac{\binom{k}{i}}{(k-1-i)!} \cdot \frac{\binom{2}{j}}{(2-1-j)!} \right).
\end{aligned}$$

We calculate the three individual terms to be

$$\binom{k+1}{k-1} k! \left( \sum_{i+j=0} \frac{\binom{k}{i}}{(k-1-i)!} \cdot \frac{\binom{2}{j}}{(2-1-j)!} \right) = \frac{(k+1)k}{2} \cdot k! \cdot \frac{\binom{k}{0}}{(k-1)!} \cdot \frac{\binom{2}{0}}{1!} = \frac{(k+1)k^2}{2},$$

and

$$-\binom{k}{k-1} (k-1)! \left( \frac{\binom{k}{1}}{(k-2)!} + \frac{\binom{2}{1}}{(k-1)!} \right) = -k! \left( \frac{k}{(k-2)!} + \frac{2}{(k-1)!} \right) = -k^2(k-1) - 2k,$$

and

$$\binom{k-1}{k-1}(k-2)!\left(\frac{\binom{k}{1}\binom{2}{1}}{(k-2)!}+\frac{\binom{k}{2}}{(k-3)!}\right)=(k-2)!\left(\frac{2k}{(k-2)!}+\frac{k(k-1)}{2(k-3)!}\right)$$

$$=2k+\frac{k(k-1)(k-2)}{2}.$$

Adding those three summands we obtain

$$\delta_k(\mathcal{D}_1)=\frac{k^3+k^2+k^3-3k^2+2k}{2}-k^3+k^2-2k+2k=k.$$

Note that there is also a more structural argument to obtain this polar degree. In fact, the polar degree $\delta_l(\mathcal{D}_1)=0$ for $l>\dim(\mathcal{D}_1^*)-1$, where $\mathcal{D}_1^*$ denotes the dual variety of $\mathcal{D}_1$ (Özlüm Çelik et al., 2021). Note that in the case of $k\times 2$ matrices $\mathcal{D}_1^*=\mathcal{D}_1$ (Draisma et al., 2016) and hence it holds that $\delta_l(\mathcal{D}_1)=0$ for $l>\dim(\mathcal{D}_1)-1=k$ (Spaenlehauer, 2012). The largest non-zero polar degree is equal to the degree of the dual variety (Draisma et al., 2016) and hence we obtain $\delta_k(\mathcal{D}_1)=\mathrm{degree}(\mathcal{D}_1^*)=\mathrm{degree}(\mathcal{D}_1)=k$ (Spaenlehauer, 2012). □

Note that this bound is not necessarily sharp, since it is exactly the number of complex solutions of the criticality equations over $\mathcal{U}\cap\mathcal{D}_1$. Overall, we have seen that the study of the algebraic properties of the reward function provided an upper bound on the number of critical points of the problem, which can be improved using the description of the state-action frequencies as a basic semialgebraic set and employing tools from algebraic geometry.

### D.4 EXAMPLES WITH MULTIPLE SMOOTH AND NON-SMOOTH CRITICAL POINTS

It is the goal of this example to demonstrate that for a blind controller multiple critical points can occur in the interior $(0,1)\cong\mathrm{int}(\Delta_{\mathcal{A}}^{\mathcal{O}})$ as well as at the two endpoints of $[0,1]\cong\Delta_{\mathcal{A}}^{\mathcal{O}}$ of the policy polytope. We refer to such points as smooth and non-smooth critical points. We consider a blind controller with one observation, two actions $a_1,a_2$ and three states $s_1,s_2,s_3$ and a deterministic transition kernel $\alpha$ and reward described by the graph shown in Figure 2.

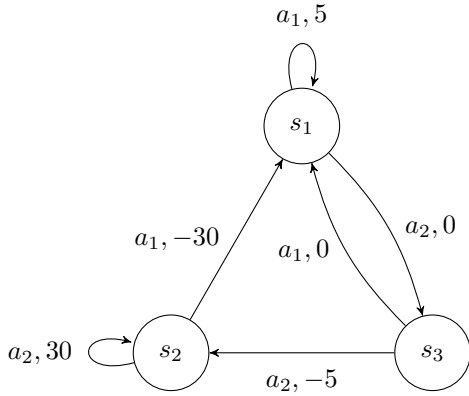

Figure 2: Graph describing the deterministic transition kernel $\alpha$ and the associated instantaneous rewards.

We make the usual identification $[0,1]\cong\Delta_{\mathcal{A}}^{\mathcal{O}}$, where we associate $p$ with $\pi(a_1|o)$. In Figure 3, the reward function is plotted on the left for the three initial conditions $\mu=\delta_{s_1},\delta_{s_2},\delta_{s_3}$. It is apparent that the reward has two critical points in the interior of the policy polytope $\Delta_{\mathcal{A}}^{\mathcal{O}}\cong[0,1]$ for the two initial conditions $\mu=\delta_{s_1}=\delta_{s_3}$. For $\mu=\delta_{s_2}$, there are two strict local maxima on the two endpoints of the interval. In this example, the bound from Proposition 21 ensures that there are at most $|\mathcal{S}|=3$ critical points in the interior and at most $|\mathcal{S}|+2=5$ critical points in the whole policy polytope. We see that those bounds are not sharp in this specific setting. Note that this example is stable under small perturbations of the transition kernel and reward vector and hence can occur for

generic $\alpha$ and $r$. The right hand side of Figure 3 shows a three dimensional random projection of the set of feasible discounted state-action frequencies. By Theorem 4 they are a curve in $\mathbb{R}^{\mathcal{S} \times \mathcal{A}} \cong \mathbb{R}^6$ with an injective rational parametrization of degree at most $|\mathcal{S}| = 3$.

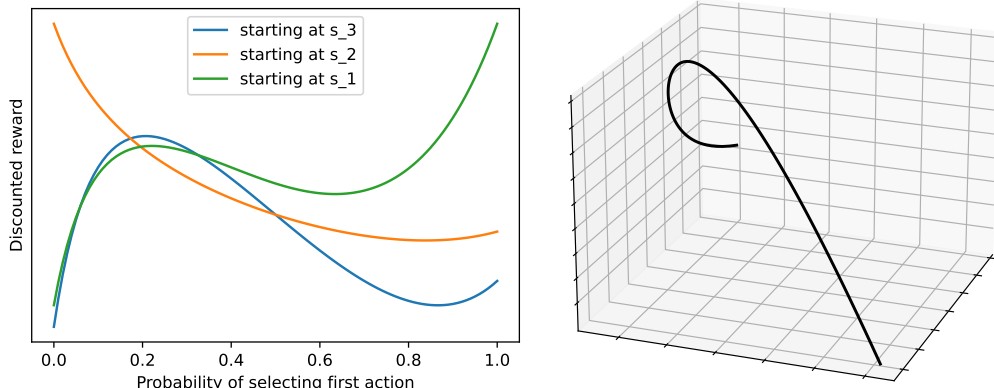

Figure 3: Plot of the reward function for initial distributions $\delta_s$ on the left and of a three-dimensional random projection of the set of feasible discounted state-action frequencies on the right.

## D.5 (Super)level sets of (PO)MDPs

### D.5.1 Connectedness of superlevel sets in MDPs

**Theorem 66** (Existence of improvement paths in MDPs). *For every policy $\pi \in \Delta_{\mathcal{A}}^{\mathcal{S}}$, there is a continuous path connecting $\pi$ to an optimal policy along which the reward is monotone. If further $\pi \mapsto \eta^{\pi}$ is injective, the reward is strictly monotone along this path, if $\pi$ is suboptimal. In particular, the superlevel sets of MDPs are connected.*

*Proof.* Let us fix $\pi \in \Delta_{\mathcal{A}}^{\mathcal{S}}$ and set $\eta_0 := \eta^{\pi}$ and $\eta_1$ be a global optimum and $\eta_t$ be the linear interpolation and $\rho_t$ be the corresponding state marginal. Note that for $s \in \mathcal{S}$ it holds that either $\rho_t(s) > 0$ for all $t \in (0, 1)$ or $\rho_t(s) = 0$ for all $t \in [0, 1]$. In the latter case, we can set $\pi_t(\cdot|s)$ to be an arbitrary element in $\Delta_{\mathcal{A}}$. For the other states and $t \in (0, 1)$ we can define the policy through conditioning by $\pi_t(a|s) := \eta_t(s, a)/\rho_t(s)$ and will continuously extend the definition to $t \in \{0, 1\}$ in the following. If $\rho_0(s) > 0$ or $\rho_1(s) > 0$, then the definition extends naturally. Suppose that $\rho_0(s) = 0$, then we now that $\rho_1(s) > 0$ since otherwise $\rho_t(s) = 0$ for all $t \in [0, 1]$. Now for $t > 0$ it holds that

$$\pi_t(s, a) = \frac{\eta_t(s, a)}{\rho_t(s)} = \frac{(1-t)\eta_0(s, a) + t\eta_1(s, a)}{(1-t)\rho_0(s) + t\rho_1(s)} = \frac{t\eta_1(s, a)}{t\rho_1(s)} = \frac{\eta_1(s, a)}{\rho_1(s)},$$

which extends continuously to $t = 0$. If $\rho_1(s) = 0$, then like before, $\pi_t(\cdot|s)$ does not depend on $t$ and we can extend it to $t = 1$. Now we have constructed a continuous path $\pi_t$, such that $\eta^{\pi_t} = \eta_t$ and thus

$$R(\pi_t) = \langle r, \eta_t \rangle = (1-t)\langle r, \eta_0 \rangle + t\langle r, \eta_1 \rangle = R(\pi_0) + t(R^* - R(\pi_0)),$$

which is strictly increasing if $\pi_0$ is suboptimal. It remains to construct a continuous path between $\pi_0$ and $\pi$. Note that if $\rho_0(s) > 0$, the policies $\pi_0$ and $\pi$ agree on the state $s$ and so does the linear interpolation between the two policies. Now, by Proposition 46 we see that every linear interpolation between $\pi_0$ and $\pi$ has the state-action distribution $\eta_0$. Gluing the two paths, we obtain a path that first leaves the state-action distribution unchanged and then increases the reward strictly up to optimality. $\qquad\square$

### D.5.2 The semialgebraic structure of level and superlevel sets for POMDPs

Consider a POMDP $(\mathcal{S}, \mathcal{A}, \mathcal{O}, \alpha, \beta, r)$ and fix a discount rate $\gamma \in (0, 1)$ as well as an initial condition $\mu \in \Delta_{\mathcal{S}}$. The levelset

$$L_a := \{\pi \in \Delta_{\mathcal{A}}^{\mathcal{O}} \mid R_{\gamma}^{\mu}(\pi) = a\}$$

of the reward function is the intersection of a variety generated by one determinantal polynomial of degree at most $|\mathcal{S}|$ with the policy polytope $\Delta_{\mathcal{A}}^{\mathcal{O}}$. Indeed, by Theorem 4 the reward function $R_\gamma^\mu$ is the fraction $f/g$ of two determinantal polynomials $f$ and $g$ of degree at most $|\mathcal{S}|$. The level set consists of all policies, such that $f(\pi) = ag(\pi)$. Thus, the levelset is given by

$$L_a = \Delta_{\mathcal{A}}^{\mathcal{O}} \cap \left\{ x \in \mathbb{R}^{\mathcal{O} \times \mathcal{A}} \mid f(x) - ag(x) = 0 \right\}.$$

Analogously, a superlevel set is the intersection

$$\Delta_{\mathcal{A}}^{\mathcal{O}} \cap \left\{ x \in \mathbb{R}^{\mathcal{O} \times \mathcal{A}} \mid f(x) - ag(x) \geq 0 \right\}$$

of a basic semialgebraic generated by one determinantal polynomial of degree at most $|\mathcal{S}|$ with the policy polytope $\Delta_{\mathcal{A}}^{\mathcal{O}}$. In particular, both the levelset and superlevel sets of POMDPs are semialgebraic sets defined by linear inequalities and equations (corresponding to the conditional probability polytope $\Delta_{\mathcal{A}}^{\mathcal{O}}$) and a determinantal (in)equality of degree at most $|\mathcal{S}|$. This description can be used to bounds the number of connected components, which captures important properties of the loss landscape of an optimization problem (Barannikov et al., 2019; Catanzaro et al., 2020). By a theorem due to Łojasiewicz, level and superlevel sets possess finitely many connected (semialgebraic) components (Ruiz, 1991; Basu et al., 2006) and there exist algorithmic approaches to computing the number of connected components (Grigor'ev & Vorobjov, 1992) as well as explicits upper bounds, which involve the dimension, the number of defining polynomials as well as their degrees (Basu, 2003; 2014). Those results are generalizations of the classic result due to Milnor and Thom which bounds the sum of all Betti numbers of a variety. If we apply the Milnor-Thom theorem to the variety $\mathcal{V}$ we obtain that there are at most $|\mathcal{S}|(2|\mathcal{S}| - 1)^{|\mathcal{O}||\mathcal{A}|-1}$ many connected components of $\mathcal{V}$. This bound neglects the determinantal nature of the defining polynomial and might therefore be coarse. Using an analogue approach, we can also study the level and superlevel sets of the reward function in the space of feasible state-action frequencies. Indeed, they are the intersections of the hyperplane $\{\eta \in \mathbb{R}^{\mathcal{S} \times \mathcal{A}} \mid \langle r, \eta \rangle_{\mathcal{S} \times \mathcal{A}} = a\}$ and halfspace $\{\eta \in \mathbb{R}^{\mathcal{S} \times \mathcal{A}} \mid \langle r, \eta \rangle_{\mathcal{S} \times \mathcal{A}} \geq a\}$ with the semialgebraic set $\mathcal{N}_\gamma^{\mu,\beta}$ of state-action frequencies.

# E    POSSIBLE EXTENSIONS

## E.1    APPLICATION TO FINITE MEMORY POLICIES

In general, it is possible to reduce POMDPs with finite memory policies to a POMDP with memoryless policies by augmenting the state and observation space with the memory. Say we consider policies with a memory that stores the last $k$ observations that were made. Then we could set $\tilde{\mathcal{S}} := \mathcal{S} \times \mathcal{O}^{k-1}$ and $\tilde{\mathcal{O}} := \mathcal{O}^k$. If the first state is $s_0$ and the first observation that is being made is $o_0$, then we will associate it with $\tilde{s}_0 := (s_0, o_0, \ldots, o_0) \in \tilde{\mathcal{S}}$ and $\tilde{o}_0 := (o_0, \ldots, o_0) \in \tilde{\mathcal{O}}$ respectively. If after $t$ steps, the current state is $\tilde{s}_t = (s_t, o_{t-k}, \ldots, o_{t-1})$ and the next observation is $o_t$, then we set $\tilde{o}_t := (o_{t-k}, \ldots, o_t)$. An analogue strategy can be taken when the memory does consist of more than the history of observations and for example includes the history of decision. It remains open to explore the implications of the translation of our results to policies with internal memory with this identification.

## E.2    POLYNOMIAL POMDPS

Zahavy et al. (2021) consider MDPs, where the objective is a convex function of the state-action frequency, i.e., where $R_\gamma^\mu(\pi) = f(\eta_\gamma^{\pi,\mu})$ for some convex function $f \colon \mathbb{R}^{\mathcal{S} \times \mathcal{A}} \to \mathbb{R}$ and coin the name of *convex MDPs*. In analogy, we refer to the case where $f$ is a polyomial function as *polynomial (PO)MDPs*. In polynomial POMDPs, the problem of reward maximization is by definition an optimization problem of a polynomial function over the set $\mathcal{N}_\gamma^{\mu,\beta}$ of feasible state-action frequencies. Since the feasible state-action frequencies form a basic semialgebraic set, the problem of reward maximization in polynomials is a polynomial optimization problem. Hence, the method of bounding the number of critical points as discussed in Section 5 generalizes to the case of polynomial reward criteria. If $f$ is a polynomial of degree $d$, the upper bound (5) from Theorem 20 takes the form

$$\prod_{o \in O} d_o^{k_o} \cdot \sum_{i + \sum_{o \in O} i_o = m} (d-1)^i \prod_{o \in O} (d_o - 1)^{i_o}.$$

The use of polar degrees does not extend in general to the case of polynomial POMDPs, since they require a linear objective function, but can still be related to the algebraic degree for a quadratic objective as it is the case for the Euclidean distance function (Draisma et al., 2016).

## F EXAMPLES

Here, we provide examples, which illustrate our findings. In particular, we compute the defining polynomial inequalities of the set of feasible state-action frequencies for the example from Figure 1 and a navigation problem in a grid world. We use an interior point method to solve the constrained optimization problem corresponding to the polynomial programming formulation of the respective POMPDs and see that in this offers a computationally feasible approach to the reward maximization problem.

### F.1 TOY EXAMPLE OF FIGURE 1

We discuss in detail a toy POMDP which we used to generate the plots in Figure 1. We consider state, observation, and action spaces with two elements each, as well as following deterministic transition mechanism $\alpha$, observation mechanism $\beta$, and instantaneous reward $r$:

$$
\begin{aligned}
\mathcal{S} &= \{s_1, s_2\}, \\
\mathcal{O} &= \{o_1, o_2\}, \\
\mathcal{A} &= \{a_1, a_2\}, \qquad \alpha(s_i|s_j, a_k) = \delta_{ik},
\end{aligned}
\qquad
\beta = \begin{pmatrix} 1 & 0 \\ 1/2 & 1/2 \end{pmatrix}, \qquad
\begin{aligned}
r(s,a) &= \delta_{s_1,s}\delta_{a_1,a} + \delta_{s_2,s}\delta_{a_2,a}, \\
\gamma &= 0.5.
\end{aligned}
$$

The transitions, instantaneous rewards, and observations are shown in Figure 4. As an initial distribution we take the uniform distribution $\mu = (\delta_{s_1} + \delta_{s_2})/2$ over the states.

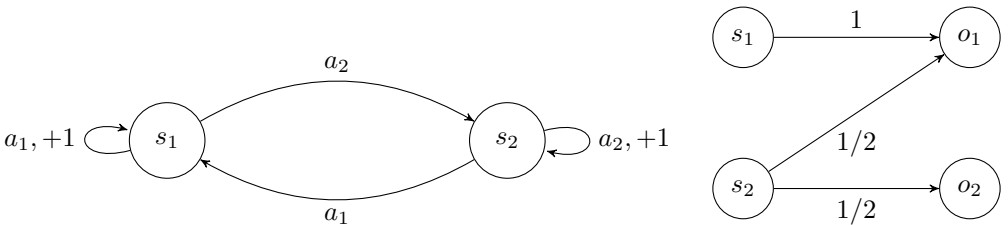

Figure 4: The left shows the transition graph of the toy example. The right shows the observation mechanism; the numbers on the edges indicate the observation probabilities.

**Polynomial programming formulation** To illustrate Theorem 16 (and Proposition 14), we derive step-by-step the explicit polynomial program for the reward maximization in this toy example. For this, we first compute the defining inequalities of the set of feasible state-action frequencies. We begin with the linear constraints that define the set $\mathcal{N}_\gamma^\mu$ of state-action frequencies of the associated MDP, given in general form in Proposition 8. In the remainder, we denote the state-action frequencies as matrices

$$
\eta = \begin{pmatrix} \eta(s_1, a_1) & \eta(s_1, a_2) \\ \eta(s_2, a_1) & \eta(s_2, a_2) \end{pmatrix} = \begin{pmatrix} \eta_{11} & \eta_{12} \\ \eta_{21} & \eta_{22} \end{pmatrix} \in \mathbb{R}^{\mathcal{S} \times \mathcal{A}}.
$$

Following Proposition 8, the linear inequalities are $\eta_{ij} \geq 0$ for all $i, j \in \{1, 2\}$, and the linear equations are $\langle w_\gamma^i, \eta \rangle_{\mathcal{S} \times \mathcal{A}} = (1 - \gamma)\mu_i = 1/4$ for $i \in \{1, 2\}$, whereby here

$$
w_\gamma^1 = \delta_1 \otimes \mathbb{1}_{\mathcal{A}} - \gamma\alpha(1|\cdot, \cdot) = \begin{pmatrix} 1 & 1 \\ 0 & 0 \end{pmatrix} - \frac{1}{2}\begin{pmatrix} 1 & 0 \\ 1 & 0 \end{pmatrix} = \frac{1}{2}\begin{pmatrix} 1 & 2 \\ -1 & 0 \end{pmatrix}
$$

and

$$
w_\gamma^2 = \delta_2 \otimes \mathbb{1}_{\mathcal{A}} - \gamma\alpha(2|\cdot, \cdot) = \begin{pmatrix} 0 & 0 \\ 1 & 1 \end{pmatrix} - \frac{1}{2}\begin{pmatrix} 0 & 1 \\ 0 & 1 \end{pmatrix} = \frac{1}{2}\begin{pmatrix} 0 & -1 \\ 2 & 1 \end{pmatrix}.
$$

Thus the two linear equations are

$$
\begin{aligned}
2\eta_{11} + 4\eta_{12} - 2\eta_{21} &= 1 \\
-2\eta_{12} + 4\eta_{21} + 2\eta_{22} &= 1.
\end{aligned}
\tag{28}
$$

It remains to compute the polynomial inequalities, which can be done using Remark 18. We invert the matrix $\beta$ and obtain

$$\beta^+ = \beta^{-1} = \begin{pmatrix} 1 & 0 \\ -1 & 2 \end{pmatrix} \in \mathbb{R}^{\mathcal{S} \times \mathcal{O}}.$$

Using the notation from Remark 18 we have $S_{o_1} = \{s_1\}$ and $S_{o_2} = \{s_1, s_2\}$, and thus the polynomial inequalities are

$$\eta_{11} \geq 0$$
$$\eta_{12} \geq 0$$
$$-\eta_{11}(\eta_{21} + \eta_{22}) + 2\eta_{21}(\eta_{11} + \eta_{12}) \geq 0$$
$$-\eta_{12}(\eta_{21} + \eta_{22}) + 2\eta_{22}(\eta_{11} + \eta_{12}) \geq 0.$$

The first two inequalities can be seen to be redundant and can be discarded. Finally, note that the objective function is given by

$$\langle r, \eta \rangle_{\mathcal{S} \times \mathcal{A}} = \eta_{11} + \eta_{22}.$$

Hence, we have obtained the following explicit formulation of the reward maximization problem as a polynomial optimization problem:

$$\text{maximize } \eta_{11} + \eta_{22} \quad \text{subject to} \quad \begin{cases} 2\eta_{11} + 4\eta_{12} - 2\eta_{21} - 1 = 0 \\ -2\eta_{12} + 4\eta_{21} + 2\eta_{22} - 1 = 0 \\ \eta_{11}, \eta_{12}, \eta_{21}, \eta_{22} \geq 0 \\ \eta_{11}\eta_{21} + 2\eta_{21}\eta_{12} - \eta_{11}\eta_{22} \geq 0 \\ \eta_{12}\eta_{22} + 2\eta_{11}\eta_{22} - \eta_{12}\eta_{21} \geq 0. \end{cases} \tag{29}$$

**Solution with constrained and polynomial optimization tools** The formulation (29) allows us to use polynomial optimization algorithms, semi-definite programming (SDP) solvers, or relaxation hierarchies such as the popular Sum Of Squares (SOS). Using the modeling language `JuMP` and the interior point solver `Ipopt` we directly obtained the globally optimal[10] solution to problem (29) (rounded to three digits)

$$\eta^* = \begin{pmatrix} 0.667 & 0 \\ 0.167 & 0.167 \end{pmatrix}.$$

The corresponding optimal state policy $\tau^*$ is obtained simply by conditioning on states, and any pre-image under the observation kernel is an optimal observation policy, in this case simply $\pi^* = \beta^{-1}\tau^*$,

$$\pi^* = \begin{pmatrix} 1 & 0 \\ 0 & 1 \end{pmatrix} \in \Delta_{\mathcal{A}}^{\mathcal{O}}.$$

This policy achieves a reward of $R_\gamma^\mu(\pi^*) = \langle r, \eta^* \rangle_{\mathcal{S} \times \mathcal{A}} = 0.833$ (rounded to three digits). The computations took $0.01$s (on a 2 GHz Quad-Core Intel Core i5 processor). The command in `JuMP` to call the optimizer `Ipopt` is simply:

```
model = Model(optimizer_with_attributes(Ipopt.Optimizer)
@variable(model, \eta[1:2, 1:2]>=0)
@constraint(model, 2\eta[1, 1] + 4\eta[1, 2] - 2\eta[2, 1] == 1)
@constraint(model, -2\eta[1, 2] + 4\eta[2, 1] + 2\eta[2, 2] == 1)
@constraint(model, \eta[1, 1]\eta[2, 1] + 2\eta[2, 1]\eta[1, 2]
    - \eta[1, 1]\eta[2, 2] >= 0)
@constraint(model, \eta[1, 2]\eta[2, 2] + 2\eta[1, 1]\eta[2, 2]
    - \eta[1, 2]\eta[2, 1] >= 0)
@NLobjective(model, Max, \eta[1, 1] + \eta[2, 2])
optimize!(model)
```

For completeness, we also provide the command to solve a relaxation in Python `SumOfSquares`, which is the following, although we found this to run a bit slower depending on the selected degree. Here we negate the objective in order to obtain a minimization problem and square the search variables (which are required to be non-negative) in order to obtain polynomials of even degree:

---

[10]The SOS relaxation provides a certificate for global optimality in this case.

```
e11, e12, e21, e22 = sp.symbols('e11 e12 e21 e22')
prob = poly_opt_prob([e11, e12, e21, e22], - e11**2 - e22**2,
    eqs=[+ 2 * e11**2 + 4 * e12**2 - 2 * e21**2 - 1,
    - 2 * e12**2 + 4 * e21**2 + 2 * e22**2 - 1,
    + e11**2 + e12**2 + e21**2 + e22**2 - 1],
    ineqs=[e11**2 * e21**2 + 2 * e21**2 * e12**2 - e11**2 * e22**2,
    e12**2 * e22**2 + 2 * e11**2 * e22**2 - e12**2 * e21**2], deg=2)
prob.solve()
print(prob.value)
```

**Policy gradient methods may not find a global optimum**    We want to demonstrate an important problem of policy gradient methods, which is the well known possibility to get stuck in local optima, in the case of the toy example. For this, we used a tabular softmax policy model to represent the interior of the policy polytope $\Delta_{\mathcal{A}}^{\mathcal{O}}$, i.e. used the following parametric policy model

$$\pi_\theta(a|o) := \frac{\exp(\theta_{oa})}{\sum_{a'} \exp(\theta_{oa'})} \quad \text{for } \theta \in \mathbb{R}^{\mathcal{O} \times \mathcal{A}}.$$

We computed 15 policy gradient trajectories, where we used the policy gradient theorem (see Corollary 52) to compute the update directions. The starting positions where generated randomly, such that the initial conditions in the policy polytope $\Delta_{\mathcal{A}}^{\mathcal{O}}$ are uniformly random. The trajectories in the policy polytope $\Delta_{\mathcal{A}}^{\mathcal{O}} \cong [0, 1]^2$ are shown in Figure 5, which also shows a heat map of the reward function. We observe that 5 of the trajectories converge to a suboptimal strict local minimum. Note that this is not artefact of the parametrization, but of the fact that there is a strict local minimum and hence every naive local optimization method will suffer from this problem. The reward of the suboptimal local minimum

$$\pi = \begin{pmatrix} 1 & 0 \\ 1 & 0 \end{pmatrix} \in \Delta_{\mathcal{A}}^{\mathcal{O}}$$

is 0.747 if rounded to 3 digits.

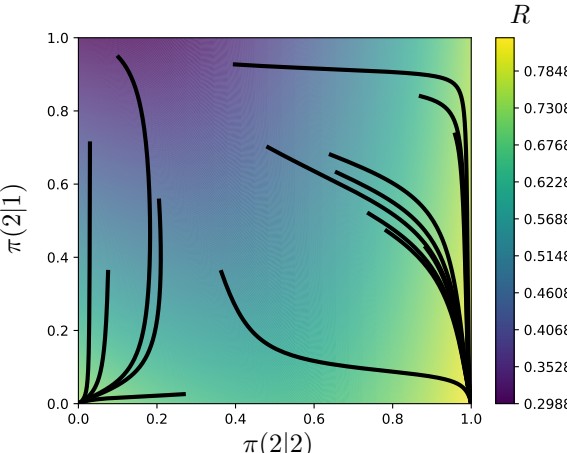

Figure 5: Policy gradient optimization trajectories (shown as black curves) in the observation policy polytope. As expected, since the problem is non-convex and has several distinct local optimizers, the trajectories converge to different local optimizers depending on the initial policy.

**Number of critical points**    We evaluate the bound of Theorem 20 for this toy problem. First note that in this example the observation matrix $\beta$ is invertible with $\beta^{-1} = \begin{pmatrix} 1 & 0 \\ -1 & 2 \end{pmatrix}$. Further, Assumption 13 is satisfied for initial distributions $\mu$ with full support. Hence, we can apply Theorem 20. Here, we have $|\mathcal{S}| = |\mathcal{A}| = |\mathcal{O}| = 2$ and in the notation of Theorem 20 we have $d_{o_1} = 1$ and $d_{o_2} = 2$. As discussed in the main body, the bound evaluates to zero if we consider the interior of the policy polytope, which corresponds to $I = \emptyset$. This means that there are no critical

points in the interior of the policy polytope, in other words, all optimal policies lie at the boundary and hence have one or more zero entries. The one-dimensional faces correspond to the index sets $\{(a_1, o_1)\}, \{(a_2, o_1)\}, \{(a_1, o_2)\}, \{(a_2, o_2)\}$. The choices $I = \{(a_1, o_1)\}, \{(a_2, o_1)\}$ correspond to the two edges on the left and right of the policy polytope $\Delta_{\mathcal{A}}^{\mathcal{O}}$ as shown in the top left corner of Figure 1 or alternatively to the two straight faces of the set $\mathcal{N}_\gamma^{\mu,\beta}$ of state-action distributions shown in the top right corner. The bound (5) evaluates to zero for those choices. This can also be seen in the bottom row in Figure 1, where it is apparent that there are no critical points on the respective faces. For the choices $I = \{(a_1, o_2)\}, \{(a_2, o_2)\}$ the bound (5) evaluates to two. Indeed, these faces contain critical points. The bound is not sharp in this case since the actual number of critical points in any of the two faces of the policy polytope $\Delta_{\mathcal{A}}^{\mathcal{O}}$, which correspond to the two non-linear faces of $\mathcal{N}_\gamma^{\mu,\beta}$ is one. Nonetheless, this illustrates how the theorem allows us to discard most faces of the polytope and focus the search for an optimal policy on just two faces.

## F.2 Navigation in a grid world

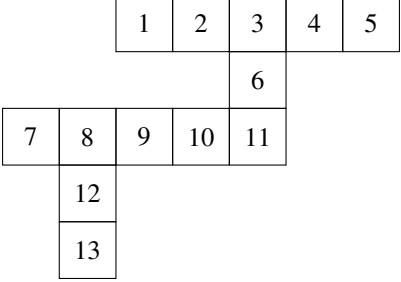

Figure 6: Depiction of a grid world; the reward of $R$ is obtained in state 1, the actions are $\{R, L, U, D\}$ corresponding to movements to the right, left, up and down; observed are the possible directions that the agent can move in. Once the agent transitions to state 1, she transfers to state 7 and 13 uniformly.

We consider the grid world depicted in Figure 6 with 13 states and 7 observations, where it is the goal to reach state 1. The four actions are $\{R, L, U, D\}$ corresponding to the directions right, left, up and down on the grid. The transitions are deterministic and lead to the cell right, left, above or below the current cell, if this cell is admissible; from the goal state 1 one transitions uniformly to the states 7 and 13 independently of the chosen action. Further, we consider deterministic observations, which correspond to the agent being able to observe its immediate four neighboring positions. This observation mechanism partitions the state space into the seven subsets $\{1, 7\}, \{2, 4, 9, 10\}, \{3, 8\}, \{5\}, \{6, 12\}, \{11\}, \{13\}$, which lead to the observations $o_1, o_2, o_3, o_4, o_5, o_6$ and $o_7$ respectively. Hence, by Remark 57 the polynomial constraints are given by

$$\eta_{7a}\rho_1 - \eta_{1a}\rho_7 = 0 \quad \text{for all } a \in \mathcal{A}$$
$$\eta_{4a}\rho_2 - \eta_{2a}\rho_4 = 0 \quad \text{for all } a \in \mathcal{A}$$
$$\eta_{8a}\rho_2 - \eta_{2a}\rho_8 = 0 \quad \text{for all } a \in \mathcal{A}$$
$$\eta_{10a}\rho_2 - \eta_{2a}\rho_{10} = 0 \quad \text{for all } a \in \mathcal{A}$$
$$\eta_{9a}\rho_3 - \eta_{3a}\rho_9 = 0 \quad \text{for all } a \in \mathcal{A}$$
$$\eta_{12a}\rho_6 - \eta_{6a}\rho_{12} = 0 \quad \text{for all } a \in \mathcal{A},$$

where $\rho_s = \sum_a \eta_{sa}$. The linear constraints apart from $\eta \geq 0$ can be computed to be

$$\rho_1 - \gamma(\eta_{12} + \eta_{13} + \eta_{14} + \eta_{22}) = \mu_1$$
$$\rho_2 - \gamma(\eta_{11} + \eta_{23} + \eta_{24} + \eta_{32}) = \mu_2$$
$$\rho_3 - \gamma(\eta_{21} + \eta_{33} + \eta_{42}) = \mu_3$$
$$\rho_4 - \gamma(\eta_{31} + \eta_{43} + \eta_{44} + \eta_{52}) = \mu_4$$
$$\rho_5 - \gamma(\eta_{41} + \eta_{53} + \eta_{54}) = \mu_5$$
$$\rho_6 - \gamma(\eta_{34} + \eta_{61} + \eta_{62} + \eta_{11,3}) = \mu_6$$
$$\rho_7 - \gamma(\rho_1/2 + \eta_{72} + \eta_{73} + \eta_{74} + \eta_{82}) = \mu_7$$
$$\rho_8 - \gamma(\eta_{71} + \eta_{83} + \eta_{84} + \eta_{92}) = \mu_8$$
$$\rho_9 - \gamma(\eta_{81} + \eta_{93} + \eta_{10,2}) = \mu_9$$
$$\rho_{10} - \gamma(\eta_{91} + \eta_{10,3} + \eta_{10,4} + \eta_{11,2}) = \mu_{10}$$
$$\rho_{11} - \gamma(\eta_{10,1} + \eta_{11,1} + \eta_{11,4}) = \mu_{11}$$
$$\rho_{12} - \gamma(\eta_{94} + \eta_{12,1} + \eta_{12,2} + \eta_{13,3}) = \mu_{12}$$
$$\rho_{13} - \gamma(\rho_1/2 + \eta_{12,4} + \eta_{13,1} + \eta_{13,2} + \eta_{13,4}) = \mu_{13}.$$

Further, the objective function is given by

$$\langle r, \eta \rangle_{\mathcal{S} \times \mathcal{A}} = \eta_{11} + \eta_{12} + \eta_{13} + \eta_{14}.$$

Let us now consider the uniform distribution $\mu_s = 1/13$ for $s \in \mathcal{S}$ as an initial distribution and $\gamma = 0.999$ as a discount factor. Like for the toy problem we used the interior point method *Ipopt* implemented in the Julia packages `JuMP` and `Ipopt` to solve this polynomial optimization problem. The solver took around $0.03$s consistently (on a 2 GHz Quad-Core Intel Core i5 processor). The found solution is (rounded to three digits)

$$\eta^* = \begin{array}{c} \\ 1 \\ 2 \\ 3 \\ 4 \\ 5 \\ 6 \\ 7 \\ 8 \\ 9 \\ 10 \\ 11 \\ 12 \\ 13 \end{array} \begin{array}{cccc} R & L & U & D \\ \left( 0.033 \right. & 0.000 & 0.000 & 0.000 \\ 0.044 & 0.033 & 0.000 & 0.000 \\ 0.049 & 0.077 & 0.000 & 0.000 \\ 0.066 & 0.049 & 0.000 & 0.000 \\ 0.000 & 0.066 & 0.000 & 0.000 \\ 0.000 & 0.000 & 0.033 & 0.000 \\ 0.133 & 0.000 & 0.000 & 0.000 \\ 0.075 & 0.117 & 0.000 & 0.000 \\ 0.057 & 0.042 & 0.000 & 0.000 \\ 0.033 & 0.024 & 0.000 & 0.000 \\ 0.000 & 0.000 & 0.033 & 0.000 \\ 0.000 & 0.000 & 0.017 & 0.000 \\ 0.000 & 0.000 & 0.016 & \left. 0.000 \right) \end{array} \in \Delta_{\mathcal{S} \times \mathcal{A}} \subseteq \mathbb{R}^{\mathcal{S} \times \mathcal{A}}$$

and has the objective value $\langle r, \eta^* \rangle_{\mathcal{S} \times \mathcal{A}} = 0.033$. The corresponding optimal state policy $\tau^*$ is obtained simply by conditioning on states, and any pre-image under the observation kernel is an optimal observation policy, in this case simply $\pi^* = \beta^+ \tau^*$ which is (rounded to two digits)

$$\pi^* = \begin{array}{c} \\ o_1 \\ o_2 \\ o_3 \\ o_4 \\ o_5 \\ o_6 \\ o_7 \end{array} \begin{array}{cccc} R & L & U & D \\ \left( 1.00 \right. & 0.00 & 0.00 & 0.00 \\ 0.53 & 0.47 & 0.00 & 0.00 \\ 0.48 & 0.52 & 0.00 & 0.00 \\ 0.00 & 1.00 & 0.00 & 0.00 \\ 0.00 & 0.00 & 0.99 & 0.00 \\ 0.00 & 0.00 & 1.00 & 0.00 \\ 0.00 & 0.00 & 0.99 & \left. 0.00 \right) \end{array} \in \Delta_{\mathcal{A}}^{\mathcal{O}}.$$

This policy

1. moves right on observation 1 corresponding to the states 1 and 7,

2. moves right and left with probability close to $1/2$ on observation 2 corresponding to states $2, 4, 9$ and $10$,

3. moves right and left with probability close to $1/2$ on observation 3 corresponding to the states 3 and 8,

4. moves left on observation 4 corresponding to the state 5,

5. moves up on observation 5 corresponding to the states 6 and 12,

6. moves up on observation 6 corresponding to the states 11,

7. moves up on observation 7 corresponding to the state 13.

The action choices of the policy $\pi^*$ are also shown in Figure 7. Note that the policy $\hat{\pi}$ selects the best action in the states $1, 5, 6, 11, 12$ and 13. Those are the states that are either identifiable from its observation (this is the case for $5, 11$ and 13) or where the optimal actions of all states leading to the same observation agree (this is the case for the pairs $\{1, 7\}$ and $\{6, 12\}$). In the other states, where the corresponding observation is ambiguous, the policy randomizes among the two actions, which are optimal for the compatible states. This is for example the case for the states $2, 4, 9$ and 10, which all lead to observation 2. The optimal MDP policy would move left in state 2 and 4 and move right in the states 9 and 10. The POMDP policy has to randomize between moving left and right, since otherwise the agent could never reach the goal state if starting in 7 or 13. The same consideration applies to the states 3 and 8, which both lead to observation 3. The Julia code is available in the supplements and under `https://github.com/muellerjohannes/geometry-POMDPs-ICLR-2022`.

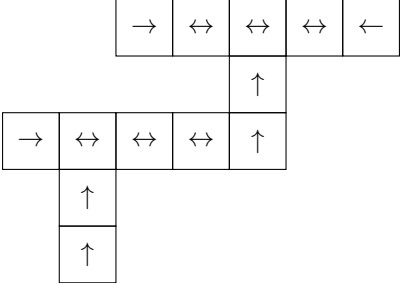

Figure 7: Depiction of the policy $\hat{\pi}$, which is found using the polynomial programming formulation of the POMDP and applying the interior point method *Ipopt* implemented in the Julia libraries `JuMP` and `Ipopt`; an arrow into two direction indicates that the agent moves into those two directions with probability close to $1/2$.

### F.3 A THREE DIMENSIONAL EXAMPLE

Let us now discuss an example where the set of discounted state-action distributions is three-dimensional and not two-dimensional as before. For this, we consider a generalization of the previous example where $|\mathcal{S}| = 3, |\mathcal{A}| = 2$ and $|\mathcal{O}| = 3$ such that

$$\dim(\Delta_{\mathcal{A}}^{\mathcal{S}}) = \dim(\Delta_{\mathcal{A}}^{\mathcal{O}}) = 3 \cdot (2 - 1) = 3.$$

The observation mechanism used is

$$\beta = \begin{pmatrix} 1 & 0 & 0 \\ 1/2 & 1/2 & 0 \\ 1/3 & 1/3 & 1/3 \end{pmatrix}.$$

Further, the action mechanism $\alpha$ and the initial distribution $\mu$ are sampled randomly and the used discount factor is 0.5. Since the initial distribution is generic and $\beta$ is invertible, the set of state-action frequencies $\mathcal{N}_\gamma^\mu$ and the set of feasible state-action frequencies $\mathcal{N}_\gamma^{\mu,\beta}$ are three-dimensional and are in fact combinatorially equivalent to the three-dimensional cube $\Delta_{\mathcal{A}}^{\mathcal{S}} \cong \Delta_{\mathcal{A}}^{\mathcal{O}} \cong [0, 1]^3$ (see Theorem 16). In Figure 8 we plot a random three-dimensional projection of the sets. More precisely, we plot their one-dimensional faces dashed and solid for the MDP and POMDP respectively. The combinatorial equivalence to the three-dimensional cube can be see in this plot.

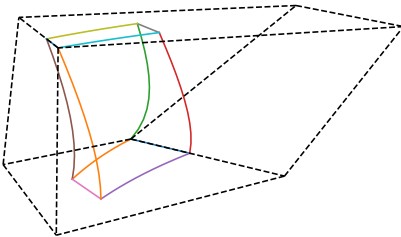

Figure 8: A random three-dimensional projection of the set of discounted state-action frequencies of the MDP is the polytope defined by the dashed straight edges. The same random three-dimensional projection of the set of feasible discounted state-action frequencies is the basic semialgebraic set where the edges are shown by the solid lines. Both of those sets are combinatorially equivalent to a three dimensional cube.

