# OpenReview forum: "The Geometry of Memoryless Stochastic Policy Optimization in Infinite-Horizon POMDPs"
_ICLR.cc/2022/Conference — ICLR 2022 Poster_

### Official Review · Reviewer_4HAL · 2021-11-01

**Correctness:** 4
**Technical Novelty And Significance:** 3
**Empirical Novelty And Significance:** Not applicable
**Recommendation:** 8
**Confidence:** 3

**Main Review:**

The paper presents an interesting set of theoretical properties about the landscape of planning in infinite-horizon POMDPs. It is interesting to know that key quantities as visiting frequencies or expected values of POMDP can be expressed as a fractional function of polynomial equations. Most importantly, degree of polynomials is proportional to a certain measure of observability (the number of possible internal states).

My main criticism is about the implication of results. While several interesting discoveries are presented in order, it does not provide much implications to actually solving the planning problem, or deeper understandings of POMDP. The paper is also quite heavy on notation and terminologies, and thus hard to follow. IMO, applying the results to a few more interesting toy examples would have helped a lot. I liked the blind-controller example.

I wonder if the authors can provide more explicit forms of the optimal policy if we consider a weaker version of partial observability. For instance, one may consider the case when a single observation comes from a single internal state w.h.p (relaxation of cardinality based measures). Another easier partially observability might be the case where the only partial observability is decided at the start of episode and non-changing or slowly-changing.

**Summary Of The Paper:**

This paper studies several properties of the landscape of memoryless planning in infinite-horizon POMDPs. The primary focus of this paper is to provide formal expression of (a) state-action frequencies and (b) cumulative reward functions given a set of memoryless stochastic policies. Specifically, they show that both quantities can be expressed as rational functions of polynomial equations. The degree of polynomials used is proportional to the number of possible internal states given a set of observations (Theorem 4). Then, the authors describes the set of feasible state-action frequencies as the solution set to a system of polynomial equations and inequalities (Theorem 16). Finally, they provide the upper bound for the number of critical points in the landscape of policy optimization (Section 5).

**Summary Of The Review:**

The paper considers an important and challenging problem, with a set of new interesting theoretical discoveries. I think this is a right step to move forward understanding POMDPs, and could be a nice addition to literature, and thus I recommend acceptance.

---

> ### Author Response · Authors · 2021-11-18
> **Response to the initial review**
>
> We thank reviewer 4HAL for the review and the valuable suggestions.  Our responses to the raised points are the following:
>
> * *Comments on implications of the results:*
>
>     * Our explicit formulation of the reward maximization as a polynomial optimization problem makes it possible to approach the policy optimization problem in POMDPs using polynomial system solvers. One possible advantage of these methods is that they lend themselves to certified solutions and to verifying that all solutions are found. As this might not have been highlighted enough, we added a specific remark that outlines this approach (see Remark 19, 58 and Algorithm 1). We demonstrated in the Appendix F that this can offer a computationally feasible approach to the reward maximization problem.
>
>     * The insights on the location and number of critical points of the reward function obtained in Section 5 have implications for gradient based methods, the design of policy models and the design of observation mechanisms and we added according explanations to our manuscript (see Remark 63, 64, 65 as well as Appendix D.5).
>
>     * We added a discussion of the implications of Theorem 20 for MDPs. In this special setting, assuming a generic instantaneous reward function, our results show that every critical point of the reward function corresponds to a deterministic policy (see Corollary 62).
>
>     * Our polynomial programming formulation also sheds light into the algebraic complexity of planning in POMDPs. We added a comment to the main body pointing to the discussion of the algebraic degree of reward maximization in the Appendix.
>
> * *Response concerning our notation:* We acknowledge that mathematical rigor comes with some amount of notation, however, we wanted to make this work more accessible to people from the applied algebraic community. Nevertheless, we tried to use intuitive notations and to repeat most statements in words to not solely rely on those notations.
>
> * *Response concerning examples:* We agree with the reviewer that more examples help to illustrate our findings and therefore, we re-organized Appendix F. It now includes more details on the toy example of Figure 1 as well as a new example of a navigation problem in a grid world. For both problems, we compute closed form expressions of the polynomial constraints describing the set of feasible state-action frequencies. Further, we apply polynomial optimization algorithms to solve the reward maximization problem in both cases.
>
> * *Comparison to finite memory policies:* We agree that the investigation of finite memory policies is an interesting and important problem. As pointed out in the introduction and outlined in the Appendix E, policies with memory can be treated under the same framework of our work, by augmenting the observation space with an external memory. After this, our results transfer directly to the case of optimization of policies with finite memory. We recognize, however, that the specific details and implementation of memory update mechanisms can in principle involve much more structure that deserves a separate investigation. We believe that our results are a very good basis for such an endeavor and hope to see it developed in future work by us or other groups.
>
> * *Response concerning relaxed compatibility conditions:* We thank the reviewer for this interesting suggestion. Montúfar and Rauh (2019) showed that optimal policies are stable under perturbations of the observation kernel $\beta$. This can be used to obtain upper bounds over the number of critical points of the reward function over the faces, which are guaranteed to possess an almost optimizer. In particular, this allows to apply Theorem 20 to the problem of the computation of almost optimal policies in the case of non-invertible observation mechanism. We added a discussion of this in Remark 64. Nevertheless, we believe that this idea deserves further attention.

---

### Official Review · Reviewer_ejas · 2021-11-02

**Correctness:** 3
**Technical Novelty And Significance:** 3
**Empirical Novelty And Significance:** 3
**Recommendation:** 6
**Confidence:** 2

**Main Review:**

The paper is interesting, well written and the ideas introduced appear to be novel. I have the following comments:

The presentation is at times somewhat dry. Commentary on the implications of some of the statements would benefit the paper. In this regard, while theoretical properties are always welcome and of interest. It is not very clear what is the direct application of the author's work is. E.g., what is the practical importance of bounding the number of critical points? Can the resulting number be exploited in the construction of efficient algorithms? Can the author's illustrate this in numerical examples?

**Summary Of The Paper:**

In this work, the authors study the problem and structure of memoryless stochastic policies in POMDPs. The focus is on the discounted infinite-horizon problem over finite state-action spaces. For this problem, the authors show theoretical properties on the the problem critical points and provide methods to compute their number.

**Summary Of The Review:**

The paper appears to be novel and well written. I am somewhat concerned on how interesting it might be to the community at large, due to its focus on theoretical properties with no clear direct application. In this regard, it it not entirely clear to this reviewer that ICLR is really the correct venue for this work.

---

> ### Author Response · Authors · 2021-11-18
> **Response to the initial review**
>
> We thank reviewer ejas for the review and the valuable suggestions. Our responses to the raised points are the following:
>
> * *Response explaining the interest in the number of critical points:*
>     * The reward function of POMDPs is known to be non-convex and to exhibit arbitrarily bad strict local minima. As a result policy gradient methods converge to suboptimal strict local minima in certain cases (see Bhandari and Russo, 2019), which explains the importance of a better structural understanding of the reward optimization problem. One step towards this is to understand the number and location of critical points. For example, knowledge about the location of critical points can be used to tighten the search space for optimal policies and to design policy models, which are guaranteed to include optimal policies (see Remark 65). Since gradient methods converge to critical points under mild conditions, the number of critical points upper bounds the number of potential limit points of gradient based policy optimization.
>
>     * Note that our results not only bound the number of critical points but also describe their location (a fact that we now highlight more prominently by a new title of Section 5) which for example has implications on the required stochasticitiy of optimal policies. For instance, in the special case of MDPs with generic instantaneous reward, our results show that there are no non-deterministic policies which are critical points of the reward function. We highlighted this fact in the updated manuscript below Theorem 20 and in greater detail in Corollary 62 in the appendix.
>
>     * The discussion of the algebraic degree of the reward maximization problem, which underlies the proofs of Section 5, has more implications than the bounds on the number of critical points. Indeed, it describes the algebraic complexity of optimal policies, meaning that it upper bounds the degree of the minimal polynomials of the entries of the optimal policy. This algebraic complexity has previously been connected to the computational complexity and we hope to explore this direction further. We added a pointer to the discussion of the algebraic complexity as presented in the appendix to the main body.
>
> * *Comments on implications of the results:*
>
>     * Our explicit formulation of the reward maximization as a polynomial optimization problem makes it possible to approach the policy optimization problem in POMDPs using polynomial system solvers. One possible advantage of these methods is that they lend themselves to certified solutions and to verifying that all solutions are found. As this might not have been highlighted enough, we added a specific remark that outlines this approach (see Remark 19, 58 and Algorithm 1). We demonstrated in the Appendix F that this can offer a computationally feasible approach to the reward maximization problem.
>
>     * The insights on the location and number of critical points of the reward function obtained in Section 5 have implications for gradient based methods, the design of policy models and the design of observation mechanisms and we added according explenations to our manuscript (see Remark 63, 64, 65 as well as Appendix D.5).
>
>     * We added a discussion of the implications of Theorem 20 for MDPs. In this special setting our result shows that every critical point of the reward function corresponds to a deterministic policy (see Corollary 62).
>
>     * Our polynomial programming formulation also sheds light into the algebraic complexity of planning in POMDPs. We added a comment to the main body pointing to the discussion of the algebraic degree of reward maximization in the Appendix.

---

### Official Review · Reviewer_JQge · 2021-11-03

**Correctness:** 4
**Technical Novelty And Significance:** 3
**Empirical Novelty And Significance:** Not applicable
**Recommendation:** 6
**Confidence:** 3

**Main Review:**

In general, the paper is well-written and I enjoyed reading, and the tools and ideas are novel. To the best of my knowledge, this appears to be the first study of the geometry of memoryless stochastic policy search for POMDPs. It would make the paper stronger if more simulation results, for example, on the geometry around the critical points, and/or compare with the case with finite-memory policies, etc. In other words, though highly nontrivial and I appreciate it, only asserting on the upper bound on the number of critical points seem not super useful. As an initial attempt, it might be also valuable to try some numerical examples to have more insights on the scenarios where the theory cannot cover.

**Summary Of The Paper:**

This paper studies the problem of finding the best memoryless stochastic policy for an infinite-horizon partially observable Markov decision process (POMDP). It is shown that the (discounted) state-action frequencies and the expected cumulative reward are rational functions of the policy, and the degree is determined by the degree of partial observability. The problem is then formulated as a linear optimization with polynomial constraints. The authors then demonstrate how the partial observability constraints can lead to multiple smooth and non-smooth local optimizers and we estimate the number of critical points.

**Summary Of The Review:**

Here are some (relatively minor) comments:

1. Terminology. I think usually in RL we refer to the "expected (discounted) state-action frequency" here as "(discounted) visitation/occupancy measure". Calling it "frequency" sounds a bit non-standard to me.
2. Proposition 22, should the mapping p being from t to $det(A+t B)$ (instead of $det(A+\lambda B)$)?
3. I was wondering how the results in the current paper reconcile with the results in Azizzadenesheli et al., '18? It seems that in that paper, some "Gradient dominance" property can be established, which should lead to some global convergence results. Since it is a closely related paper, a more detailed comparison seems necessary.
4. I was wondering whether the results in Sec. 5 can be extended to the case with $\gamma=1$ (average reward case)? If not, what would be the technical hurdles? Some discussion might be needed.

---

> ### Author Response · Authors · 2021-11-18
> **Response to the initial review I**
>
> We thank reviewer JQge for the review and the valuable suggestions. Our responses to the raised points are the following:
>
> * *Response concerning the added simulation:* We added implementations of the polynomial programming formulation of the reward maximization problem that we obtained in our manuscript. We used this to solve the toy problem of Figure 1 and a navigation problem in a grid world. We collected our findings in the re-organized Appendix F, see also the general response.
>
> * *Concerning the geometry around critical points:* The geometry around the critical points is certainly an important property of the objective function, in particular for gradient based optimization. We want to outline one possible approach to the study of the geometry around critical points that our work opens. Our reformulation of the optimization problem as a polynomially constraint problem with linear objective could shade light into this. In fact, the critical points in the policy polytope correspond to critical points of a linear function over a polynomially constrained set and more precisely to critical points of a linear function over a multi-homogeneous variety. This identification can be made by passing from a policy to its state-action frequency. Closed form expressions of the defining polynomial equalities follow from our Proposition 8 and Proposition 14 (see Appendix D for details of this correspondence). This observation gives an approach to studying the geometry around the critical points by studying the geometry of the associated varieties around the critical points (see Remark 63).
>
>     We should point out that second order criteria can be directly evaluated based on our explicit descriptions of the constraints. For instance the well developed KKT second order conditions for nonlinear constrained optimization correspond to evaluating the definiteness of the Hessian of the constraint functions along the tangent space of the active constraints. Based on this, one obtains necessary and sufficient local optimality conditions for critical points. The general approach that we take in order to bound the number of critical points is based on formulating the classic first order KKT criterion for nonlinear constrained optimization as a polynomial system. In principle, a similar approach could be pursued for other criteria in order to obtain corresponding bounds, although in the case of curvature criteria, we expect that this will lead to systems of inequalities rather than equations, with additional complications.
>
> * *Comparison to finite memory policies:* We agree that the investigation of finite memory policies is an interesting and important problem. As pointed out in the introduction and outlined in the Appendix E, policies with memory can be treated under the same framework of our work, by augmenting the observation space with an external memory. After this, our results transfer directly to the case of optimization of policies with finite memory. We recognize, however, that the specific details and implementation of memory update mechanisms can in principle involve much more structure that deserves a separate investigation. We believe that our results are a very good basis for such an endeavor and hope to see it developed in future work by us or other groups.

---

> > ### Author Response · Authors · 2021-11-18
> > **Response to the initial review II**
> >
> > * *Response explaining the interest in the number of critical points:*
> >      * The reward function of POMDPs is known to be non-convex and to exhibit arbitrarily bad strict local minima. As a result policy gradient methods converge to suboptimal strict local minima in certain cases (see Bhandari and Russo, 2019), which explains the importance of a better structural understanding of the reward optimization problem. One step towards this is to understand the number and location of critical points. For example, knowledge about the location of critical points can be used to tighten the search space for optimal policies and to design policy models, which are guaranteed to include optimal policies (see Remark 65). Since gradient methods converge to critical points under mild conditions, the number of critical points upper bounds the number of potential limit points of gradient based policy optimization.
> >
> >     * Note that our results not only bound the number of critical points but also describe their location (a fact that we now highlight more prominently by a new title of Section 5) which for example has implications on the required stochasticitiy of optimal policies. For instance, in the special case of MDPs with generic instantaneous reward, our results show that there are no non-deterministic policies which are critical points of the reward function. We highlighted this fact in the updated manuscript below Theorem 20 and in greater detail in Corollary 62 in the appendix.
> >
> >     * The discussion of the algebraic degree of the reward maximization problem, which underlies the proofs of Section 5, has more implications than the bounds on the number of critical points. Indeed, it describes the algebraic complexity of optimal policies, meaning that it upper bounds the degree of the minimal polynomials of the entries of the optimal policy. This algebraic complexity has previously been connected to the computational complexity and we hope to explore this direction further. We added a pointer to the discussion of the algebraic complexity as presented in the appendix to the main body.
> >
> > **Response to the specific comments:**
> >
> > 1. It is true that visitation or occupancy measure is a more standard term in the RL literature. We made the choice to call those distributions state-action frequencies since they have been known under this name in the MDP community and the classic works on their geometry of Derman in the 60s and 70s are formulated in this language. However, we added a pointer that they are known as visitation or occupancy measure in the RL literature, which we hope helps improve readability.
> >
> > 2. Thanks for pointing out the typo, we corrected it.
> >
> > 3. The work on policy gradient methods for POMDPs of Azizzadenesheli et al. (2018) indeed yields a gradient dominance theorem. However, a global convergence guarantee does not follow from this result, since policy gradient methods are known to converge to suboptimal strict local minima in certain cases (see Bhandari and Russo, 2019 and Appendix F). In fact, this failure of global convergence guarantees is part of our motivation to study the critical points of the reward function as we elaborated in more detail above.
> >
> > 4. Thanks for pointing out that possible generalization. Indeed, Theorem 20 can be generalized to the mean reward setting, however, the upper bound takes a slightly different and more complicated form. We added a discussion of this to the appendix in Remark 61.

---

### Official Review · Reviewer_TytX · 2021-11-06

**Correctness:** 4
**Technical Novelty And Significance:** 3
**Empirical Novelty And Significance:** Not applicable
**Recommendation:** 8
**Confidence:** 2

**Main Review:**

Strength:
POMDP is a very hard problem, and so is the memoryless policy optimization problem for POMDP. This paper sheds some light on the geometry of this non-convex problem, and I appreciate the effort and the mathematical rigor. I enjoyed reading the paper.

Weakness:

While this is a hard problem, the results in this paper are far from the practice in most senses. I am not sure how the bound on the number of critic points will be helpful for policy optimization algorithms in practice.

For example,  the authors can think about whether the bound in (5) can help identify “easy” cases, i.e. problems with one or relatively few critical points.


**Summary Of The Paper:**

This paper studies the geometric property of memoryless policy optimization problem for POMDP. The main idea is to formulate the state-action visitation frequency constraint as a polynomial constraint. Then, the degrees on the polynomials are bounded, based on which the number of critical points is obtained.


**Summary Of The Review:**

As I said, this is a hard problem and the authors obtained some interesting results. Yet, I am not sure ICLR is the best venue for this paper. Maybe a more theoretic-focused venue is more relevant.

---

> ### Author Response · Authors · 2021-11-18
> **Response to the initial review I**
>
> We thank reviewer TytX for the thoughtful review and valuable suggestions. Our responses to the raised points are the following:
>
> * *In response to your comment about the practical relevance:*
> Our work seeks to characterize the geometric properties of policy optimization in (PO)MDPs. (PO)MDPs constitute one of the most important models for planning under uncertainty (see, e.g., Azizzadenesheli et al., 2016) and naturally numerous heuristics and practical approaches have been proposed in previous works. We observe, however, that some of the key theoretical properties of this problem have not been sufficiently developed in the past. We believe that the theory is interesting on its own right but also that it can facilitate numerous practical developments. Concretely:
>
>     - One of the fruitful contemporary paradigms in machine learning is the shift of perspective from parameter space to function space (think of supervised learning with neural networks). In the context of POMDPs the observation policies and the occupation frequencies play an analogous role. The key aspect here is that the objective function in function space is `easy', in fact linear, in the case of POMDPs. The function space perspective has been the key to numerous recent breakthroughs (convergence guarantees, convergence rates, generalization bounds) in the optimization of nonconvex objective functions in parameter space and we believe that it could serve a similar role in the context of POMDP policy optimization (where we also have a nonconvex optimization problem over the parameters, possibly a parametrized policy model, and a convex objective in function space).
>
>     - With the explicit constraints in function space, we can formulate the problem as a constrained optimization problem. Although this is also possible over the space of observation policies, doing it over the space of frequencies has the advantage that the objective function is linear.
>     From this perspective, we can relax the problem in terms of a Lagrangian and formulate a Lagrange dual optimization problem that gives bounds on the optimum value of the optimization problem even when the primal problem is difficult to solve (see Boyd and Vandenberghe, Chapter 5).
>
>     - Moreover, the polynomial nature of the constraints and our explicit formulation theoreof allows us to exploit dedicated tools for polynomial optimization. One possible advantage of these methods is that they lend themselves to certified solutions and to verifying that all solutions are found. As this might not have been highlighted enough, we added a specific remark that outlines this approach (see Remark 19, 58 and Algorithm 1). We demonstrated in the Appendix F that this can offer a viable approach to the reward maximization problem.
>
>     - Using our description as a polynomial optimization problem we also derive upper bounds on the number of connected components of the superlevel sets of the reward function. Studying the connected components of superlevel sets (or sublevel sets for minimization problems) has been one of the central endeavors in the analysis of optimization and generalization in supervised learning with neural networks (see On Connected Sublevel Sets in Deep Learning, Nguyen 2019). We found it surprising that such results could be obtained in the case of optimization problems with a feedback loop.
>
> * *Identification of easy cases:* We added a discussion of the implications of Theorem 20 for fully observable systems. In this case, Theorem 20 guarantees that every critical point is a deterministic policy (see page 9 and Corollary 62). Further, we evaluated Theorem 20 for the toy problem of Figure 1, and discussed on which faces it is tight and on which not, which can be checked by inspecting the graph of the reward function in Figure 1 (see Appendix F). Nevertheless, we agree with the reviewer that a further investigation of easy cases could be beneficial as this could be used as a guidance to design good observation mechanisms. We added a pointer to this possible future direction in the appendix (see Remark 64).

---

> > ### Author Response · Authors · 2021-11-18
> > **Response to the initial review II**
> >
> > * *Response explaining the interest in the number of critical points:*
> >     * The reward function of POMDPs is known to be non-convex and to exhibit arbitrarily bad strict local minima. As a result policy gradient methods converge to suboptimal strict local minima in certain cases (see Bhandari and Russo, 2019), which explains the importance of a better structural understanding of the reward optimization problem. One step towards this is to understand the number and location of critical points. For example, knowledge about the location of critical points can be used to tighten the search space for optimal policies and to design policy models, which are guaranteed to include optimal policies (see Remark 65). Since gradient methods converge to critical points under mild conditions, the number of critical points upper bounds the number of potential limit points of gradient based policy optimization.
> >
> >      * Note that our results not only bound the number of critical points but also describe their location (a fact that we now highlight more prominently by a new title of Section 5) which for example has implications on the required stochasticitiy of optimal policies. For instance, in the special case of MDPs with generic instantaneous reward, our results show that there are no non-deterministic policies which are critical points of the reward function. We highlighted this fact in the updated manuscript below Theorem 20 and in greater detail in Corollary 62 in the appendix.
> >
> >      * The discussion of the algebraic degree of the reward maximization problem, which underlies the proofs of Section 5, has more implications than the bounds on the number of critical points. Indeed, it describes the algebraic complexity of optimal policies, meaning that it upper bounds the degree of the minimal polynomials of the entries of the optimal policy. This algebraic complexity has previously been connected to the computational complexity and we hope to explore this direction further. We added a pointer to the discussion of the algebraic complexity as presented in the appendix to the main body.

---

> > > ### Comment · Reviewer_TytX · 2021-11-24
> > > **Good response**
> > >
> > > Thank you for your detailed response and the insights. I enjoyed reading it.
> > >
> > > After reading the response and other reviewers' comments, I have raised the score accordingly.

---

> > > > ### Author Response · Authors · 2021-11-25
> > > > **Re: Good response**
> > > >
> > > > Thank you for taking the time to read our reply and thank you again for your comments that helped us improving the manuscript. We're glad to hear that you appreciated our changes.

---

### Author Response · Authors · 2021-11-18
**Main updates of the manuscript**

We want to thank all the reviewers for the careful inspection of our manuscript, positive feedback, and helpful comments. We updated our manuscript accordingly. In the revision the updates are highlighted in green.

* We added a description of the pipeline to obtain the polynomial optimization problem in Algorithm 1. Our results yield closed form expressions of the polynomial constraints describing the set of feasible state-action frequencies. This allows us to apply well-developed polynomial optimization algorithms, as for example interior point methods and sums of squares relaxations, to solve the reward maximization problem. In some cases this is very fast and yields a global optimum where local policy optimization may only lead to a non-global local optimum. For comparison, note that the standard policy gradient method (which also holds for POMDPs) requires that one evaluates (or estimates) the Q function at each update, which is costly even when all transition mechanisms are known.

* We added a computational example for a navigation problem in a grid world and also for the example in Figure 1, where we apply our polynomial programming formulation. We demonstrate how to compute the optimal policy based on the explicit constraints using Julia (Ipopt interior point methods) and Python (SumOfSquares hierarchy of relaxations). In both cases this is computationally feasible (running time in the order of $10^{-2}s$). In the toy example of Figure 1, this method finds the globally optimal policy, where in contrast policy gradient methods converge to either of the two strict local maxima of the reward function, which we also observe in experiments. This shows that our polynomial programming formulation can in some cases offer a computationally viable approach to solving planning problems in POMDPs with possible benefits over standard gradient optimization.

* We added a generalization of Theorem 20 to the mean reward case, that we discuss in more detail in the individual responses.

* We added a corollary about the location of critical points in MDPs derived from our main results.

We think that these updates in response to your comments have substantially strengthened the manuscript and hope they address your comments to your satisfaction. We remain attentive to your feedback!

---

> ### Author Response · Authors · 2021-11-30
> **End of discussion period**
>
> As the discussion period is closing, we want to thank the reviewers and AC once more for the careful consideration of our manuscript and positive feedback. We made thorough efforts to address all comments and questions from the initial reviews and believe this has further strengthened the manuscript. We are glad about TytX's positive feedback in response to this and remain attentive to any remaining comments or questions.

---

### Decision · Program_Chairs · 2022-01-20

**Decision:**

Accept (Poster)

**Comment:**

The paper studies the problem of finding an optimal memory less policy for POMDPs.  This work makes an important theoretical contribution.  The reviewers are unanimous in recommending the acceptance of the paper.  Well done!